# *Caenorhabditis elegans* PIEZO channel coordinates multiple reproductive tissues to govern ovulation

Xiaofei Bai[1], Jeff Bouffard[2], Avery Lord[3], Katherine Brugman[4], Paul W Sternberg[4], Erin J Cram[3], Andy Golden[1]*

[1]National Institute of Diabetes and Digestive and Kidney Diseases, National Institutes of Health, Bethesda, United States; [2]Department of Bioengineering, Northeastern University, Boston, United States; [3]Department of Biology, Northeastern University, Boston, United States; [4]Division of Biology and Biological Engineering, California Institute of Technology, Pasadena, United States

**Abstract** PIEZO1 and PIEZO2 are newly identified mechanosensitive ion channels that exhibit a preference for calcium in response to mechanical stimuli. In this study, we discovered the vital roles of *pezo-1*, the sole *PIEZO* ortholog in *Caenorhabditiselegans,* in regulating reproduction. A number of deletion alleles, as well as a putative gain-of-function mutant, of PEZO-1 caused a severe reduction in brood size. In vivo observations showed that oocytes undergo a variety of transit defects as they enter and exit the spermatheca during ovulation. Post-ovulation oocytes were frequently damaged during spermathecal contraction. However, the calcium signaling was not dramatically changed in the *pezo-1* mutants during ovulation. Loss of PEZO-1 also led to an inability of self-sperm to navigate back to the spermatheca properly after being pushed out of the spermatheca during ovulation. These findings suggest that PEZO-1 acts in different reproductive tissues to promote proper ovulation and fertilization in *C. elegans*.

*For correspondence:
andyg@nih.gov

Competing interests: The authors declare that no competing interests exist.

## Introduction

Mechanotransduction — the sensation and conversion of mechanical stimuli into biological signals — is essential for development. PIEZO1 and PIEZO2 are newly identified excitatory mechanosensitive proteinsthat play important roles in a wide range of developmental and physiological processes in mammals (*Alper, 2017*; *Coste et al., 2010*; *Coste et al., 2012*; *Murthy et al., 2017*; *Wu et al., 2017*). PIEZO1 is a non-selective ion channel that forms homotrimeric complexes at the plasma membrane; however, PIEZO1 exhibits a preference for $Ca^{2+}$ in response to mechanical stimuli (*Coste et al., 2010*; *Gnanasambandam et al., 2015*; *Syeda et al., 2015*). Recent studies have shown that the human and mouse PIEZO1 channels respond to different mechanical stimuli, including static pressure, shear stress and membrane stretch (*Coste et al., 2010*; *Poole et al., 2014*; *Ranade et al., 2014*). PIEZO1 also regulates vascular branching and endothelial cell alignment upon sensing frictional force (shear stress) (*Li et al., 2015*; *Nonomura et al., 2018*). Stem cells also use PIEZO1 to sense mechanical signals and to initiate $Ca^{2+}$ signaling to promote proliferation and differentiation (*Del Mármol et al., 2018*; *He et al., 2018*). PIEZO2 primarily functions as a key mechanotransducer for light touch, proprioception and breathing (*Nonomura et al., 2017*; *Woo et al., 2015*; *Woo et al., 2014*). Mutations in both human *PIEZO1* and human *PIEZO2* have been identified among patients suffering from channelopathy diseases, such as dehydrated hereditary stomatocytosis (DHSt), generalized lymphatic dysplasia (GLD), and distal arthrogryposis type 5 (DA5), in which osmoregulation is disturbed (*Albuisson et al., 2013*; *Andolfo et al., 2013*; *Bae et al., 2013*; *Coste et al., 2013*; *Li et al., 2018*; *Lukacs et al., 2015*; *McMillin et al., 2014*; *Zarychanski et al.,*

*2012*). Loss-of-function mutations in the *PIEZO1* gene cause autosomal recessive congenital lymphatic dysplasia, whereas gain-of-function mutations lead to autosomal dominant stomatocytosis (*Alper, 2017*). However, the cellular and molecular mechanisms of PIEZO dysfunction in these diseases are not well understood.

*Caenorhabditis elegans* is an attractive model system for the study of mechanotransduction in vivo. *C. elegans* contains multiple tubular tissues, including the reproductive system, that experience mechanical stimulation (*Cram, 2014*; *Cram, 2015*; *Voglis and Tavernarakis, 2005*). The *C. elegans* reproductive system consists of two U-shaped gonad arms, each ending with a spermatheca and joined in the center by a shared uterus. *C. elegans* hermaphrodites produce sperm during the L4 larval stage and then shift to produce oocytes during the adult stage. About 150 sperm are stored in each spermatheca, whereas the oocytes form in the oviduct in each gonad arm. The oocyte adjacent to the spermatheca undergoes oocyte maturation ~25 min before being ovulated into the spermatheca (*Greenstein, 2005*). Oocyte maturation is triggered by sperm-derived polypeptides known as major sperm proteins (MSPs), which activate the oocyte mitogen-activated protein kinase (MPK-1) (*Miller, 2001*; *Yang et al., 2010*). Once the oocyte matures, five pairs of contractile myoepithelial cells that make up the somatic gonad and that encase the germline, named sheath cells, push the matured oocyte into the spermatheca for fertilization. The spermatheca is an accordion-like multicellular tube, consisting of two spermathecal valves, the distal valve (closest to the oviduct) and the spermathecal-uterine (sp-ut) valve, and a bag-like chamber between the two valves (*Kimble and Hirsh, 1979*; *McCarter et al., 1999*).

The two spermathecal valves are spatiotemporally coordinated to allow oocyte entry during ovulation and exit after fertilization, through acto-myosin contractions (*Kelley and Cram, 2019*). Ovulation is triggered by signaling between oocytes, sheath cells, and sperm through increasing cytosolic inositol 1,4,5-trisphosphate ($IP_3$) and $Ca^{2+}$ concentrations (*Bui and Sternberg, 2002*; *Clandinin et al., 1998*; *Han et al., 2010*). The ovulated oocyte spends 3–5 min in the dilated spermatheca with both valves closed to allow the oocyte and sperm to complete fertilization and to initiate eggshell formation (*Johnston et al., 2010*). The constriction of the spermathecal bag cells and the opening of the spermathecal-uterine valve cells expel the fertilized egg into the uterus. Meanwhile, the sperm that are swept out of the spermatheca during oocyte exit crawl back to the constricted spermatheca. The navigation of the sperm back to the spermatheca is regulated by the chemoattractant prostaglandin, which is secreted by the oocytes and sheath cells (*Kubagawa et al., 2006*). Despite the probable role of mechanical stimuli (such as stretch of oocyte entry or the contraction of the spermatheca) during this whole process, the mechanisms underlying the mechanosensitive channels in ovulation and fertilization remain largely unknown.

In this study, we hypothesized that a mechanosensitive protein such as PEZO-1, the sole PIEZO-like protein in *C. elegans,* is involved in processes that include cellular movements, such as those observed in ovulation where oocytes must transit into and out of the spermatheca. Multiple deletion mutations, as well as a putative gain-of-function mutation, caused severe reproductive deficiencies, such as reduced brood sizes and defects in ovulation and sperm navigation. Somewhat surprisingly, normal calcium release was observed in the spermatheca during early ovulations of *pezo-1* mutants. Sperm that were readily washed out of the spermatheca during ovulation failed to migrate back to the spermatheca, thus depleting the spermatheca of sperm early in the reproductive lifecycle. Supplementing male sperm through mating significantly repopulated the spermatheca with cross-sperm and rescued the extremely low ovulation rate and reduced brood size of *pezo-1* mutants. Using an auxin-inducible degradation (AID) system, we depleted PEZO-1 in somatic tissues and the germline. Reduced brood sizes were observed in each tissue-specific degradation strain, suggesting that PEZO-1 from many tissues has multiple inputs in regulating reproduction. Thus, our analysis of numerous *pezo-1* mutants suggests that PEZO-1 has a complex role in a number of tissues that are required for reproduction.

## Results

### PEZO-1 is expressed in multiple tissues throughout development

The *C. elegans* genome encodes a single *PIEZO* ortholog, *pezo-1*, of which there are 14 mRNA isoformsas the result of differential splicing and transcriptional start sites (*Figure 1—figure supplement*

*1A*; *Harris et al., 2019*); these 14 isoforms code for 12 different PEZO-1 proteins. All isoforms share a common C-terminus. To visualize the expression pattern of *pezo-1* in vivo accurately, we directly knocked-in different fluorescent reporter genes into both the N-terminus and C-terminus of the *pezo-1* endogenous locus using CRISPR/Cas9. The C-terminal knock-in reporters should tag all *pezo-1* isoforms, whereas the N-terminal knock-in reporters should only tag the eight longest *pezo-1* isoforms (*Figure 1A*, *Figure 1—figure supplement 1A*). Both GFP and mScarlet were used as reporters to generate N- and C-terminal fusions proteins. GFP::PEZO-1, mScarlet::PEZO-1, and PEZO-1::mScarlet were widely expressed from embryonic stages through adulthood (*Figure 1B–E, G–J*, *Figure 1—figure supplement 1B–G*). The genome-edited animals behaved normally, suggesting that tagging PEZO-1 with these fluorescent reporter genes causes no functional disruption. Notably, PEZO-1 is strongly expressed in several tubular tissues, including the pharyngeal-intestinal and spermathecal-uterine valves, which is consistent with our hypothesis that *pezo-1* may be responsible for mechanoperception in these tissues (*Figure 1B*, *Figure 1—figure supplement 1B,C*). Under higher magnification, we observed PEZO-1 on the plasma membranes of oocytes and embryonic cells during a variety of embryonic stages, suggesting that PEZO-1 is a transmembrane protein (*Figure 1C–E*). PEZO-1 is expressed in multiple reproductive tissues, including the germline, somatic oviduct, and spermatheca (*Figure 1F–J*). Higher magnification imaging of the spermatheca revealed that PEZO-1 is also expressed on sperm membranes (*Figure 1J*). Consistent with the hypothesis that reproductive tissues are regulated by mechanosensitive stimuli in *C. elegans*, expression of PEZO-1 probably functions to sense physical strain or contractility during ovulation and fertilization. Live imaging and detailed analysis of PEZO-1 expression patterns during reproduction revealed that GFP::PEZO-1 is expressed in sheath cells, sperm, both spermathecal valves and the spermathecal bag cells (*Figure 1K–O*, *Video 1*). The fluorescent signal of GFP::PEZO-1 is observed in both spermathecal valves, suggesting that PEZO-1 may function to sense the mechanical stimuli at the valves during ovulation (*Figure 1K,M,N*, *Video 1*). As the fertilized oocyte is pushed into the uterus, GFP::PEZO-1-labeled sperm crawl back into the constricting spermatheca after each ovulation (*Figure 1O*, *Video 1*). Collectively, these data indicate that PEZO-1 is expressed in the somatic gonadal cells and germline cells.

## Deletion of *pezo-1* causes a decrease in brood size

To investigate the function of *pezo-1*, the phenotypes of *pezo-1* knockout (*pezo-1$^{KO}$*) animals were analyzed. Three candidate null alleles were generated by CRISPR/Cas9 genome editing; one allele was a deletion of exons 1–13 (*pezo-1 NΔ*), a second had a deletion of the last seven exons, 27–33 (*pezo-1 CΔ*) (*Figure 2—figure supplement 1A,B*), and a third had a full-length deletion of the entire *pezo-1* coding sequence (*pezo-1* full deletion). Two other alleles were generated by CRISPR/Cas9: *pezo-1(sy1398)*, which has a deletion of an exon unique to the two shortest isoforms, i and j, and a putative null allele, *pezo-1(sy1199)*, which has a 'STOP-IN' mutation in exon 27 that should interfere with translation of the C-termini of all isoforms (*Figure 2—figure supplement 1B*). Although GFP::PEZO-1 and PEZO-1::mScarlet are expressed widely in adult worms, we did not observe obvious morphological differences between homozygous *pezo-1$^{KO}$* mutants and control animals. However, in all tested *pezo-1* mutants, the number of F$_1$ progeny was significantly lower than in the wild type (*Figure 2A*, *Figure 2—figure supplement 1C*). The decrease in brood size was enhanced as animals aged (36–60 hr post mid-L4, *Figure 2—figure supplement 1C*) or when grown at a higher temperature (25˚C, *Figure 2—figure supplement 1D*). In addition, about 5–25% of F$_1$ embryos failed to hatch from *pezo-1 CΔ* homozygous mutants (*Figure 2B*). To mimic a gain-of-function phenotype in *pezo-1*, we fed wildtype animals with Yoda1, a PIEZO1-specific chemical agonist that keeps the channel open (*Syeda et al., 2015*). Reduced brood sizes were observed when wildtype animals were exposed to 20 μM Yoda1 (*Figure 2C*). This phenotype did not worsen when *pezo-1$^{KO}$* animals were also treated with Yoda (*Figure 2C*). These data suggest that either deletion or overactivation of PEZO-1 is sufficient to disrupt brood size.

## Severe ovulation defects were observed in the *pezo-1* mutants

Using differential interference contrast (DIC) and confocal microscopy, we analyzed the defects associated with the observed reduction in brood size. Although embryos fill the uterus in wildtype mothers (*Figure 2D*), a mass of ooplasm in the uteri of both *pezo-1$^{KO}$* and STOP-IN mutants was

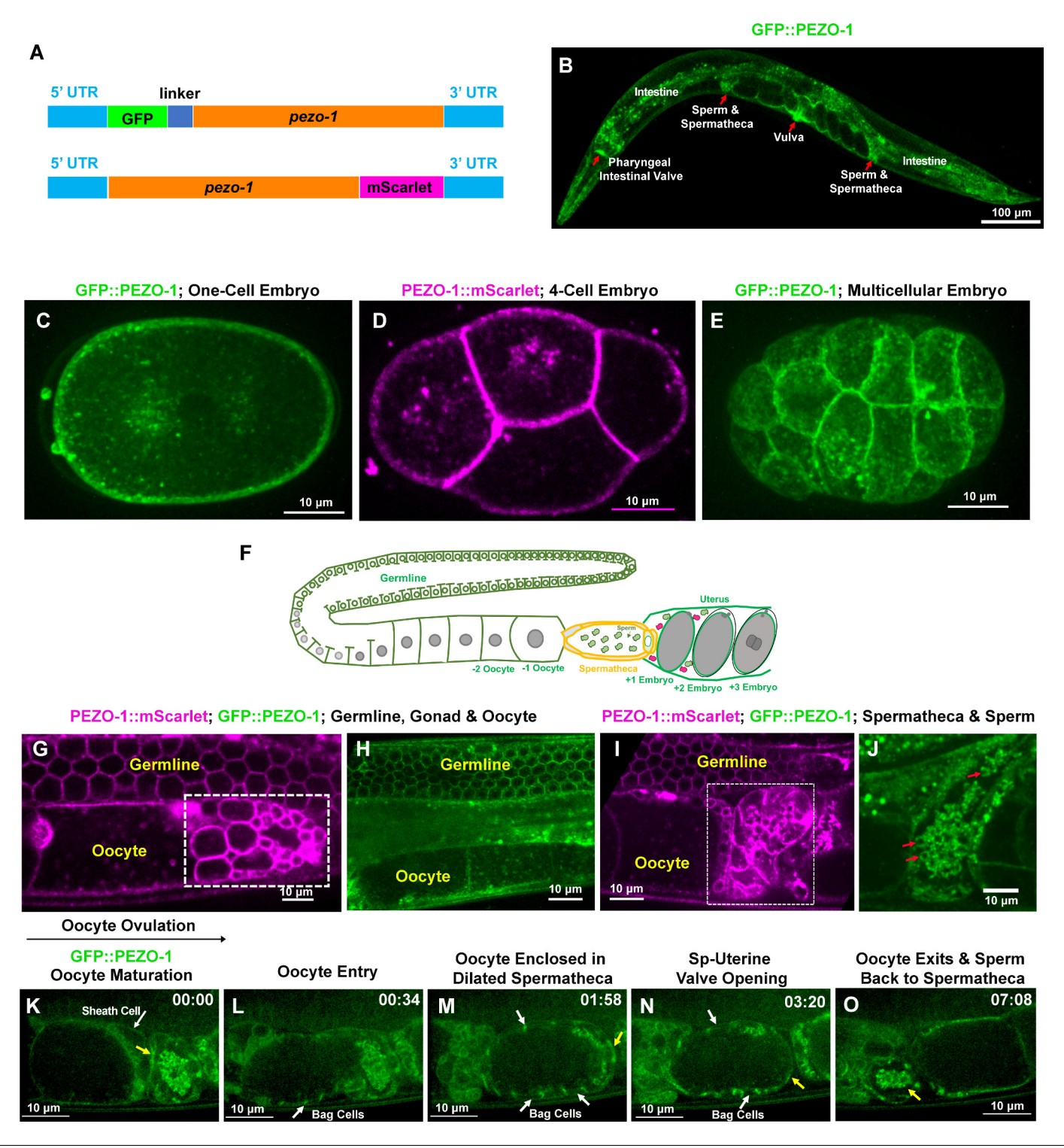

**Figure 1.** *pezo*-1 is widely expressed in *C. elegans*. (**A**) Two fluorescent reporter genes were knocked-in to both N-terminus and C-terminus of *pezo-1*. (**B**) GFP::PEZO-1 is strongly expressed in multiple mechanosensitive tissues, such as the pharyngeal-intestinal valve, spermatheca, and vulva (red arrows). (**C, E**) GFP::PEZO-1 (green) is expressed in the plasma membrane of different-staged embryos. (**D**) PEZO-1::mScarlet (magenta) also localizes to the plasma membranes of embryos. (**F**) A schematic of the *C. elegans* gonad. (**G–J**) Both PEZO-1::mScarlet (magenta) and GFP::PEZO-1 (green) localize to reproductive tissues, such as the plasma membranes of the germline cells (**G–I**), somatic gonad (**G–J**), spermatheca (I; in white box), and sperm (J; red arrows). PEZO-1::mScarlet (magenta) also labels the spermatids that have not yet migrated into the spermatheca (small circles, white box in panel [G]) and the residual bodies not yet engulfed by the sheath cells (bigger circles, white box in panel [G]) (*Huang et al., 2012*). (**K–O**)

*Figure 1 continued on next page*

*Figure 1 continued*

Representative images of PEZO-1 localization during ovulation and fertilization. GFP::PEZO-1 (green) localizes to the sheath cell (white arrow) and the spermathecal distal valve (yellow arrow (**K**), which remains closed before ovulation. The oocyte is ovulated, enters into the spermatheca (**L**) and remains enclosed in the spermatheca until fertilization is completed (**M**). During fertilization, GFP::PEZO-1 remained on the spermathecal-uterine (sp-ut) valve as indicated by a yellow arrow (**M, N**). The bag cells of the spermatheca also express GFP::PEZO-1 at this time (representative bag cells are marked by white arrows in panels (**L–N**). After fertilization, the sp-ut valve opened (**N**, yellow arrow) and allowed the newly fertilized zygote to exit the constricting spermatheca (**N, O**). Constriction of the spermatheca pushes the fertilized zygote into the uterus; sperm can be seen in the constricted spermatheca (**O**, yellow arrow). The black arrow above panel (**K**) shows the direction of embryo travelthrough the spermatheca from left to right. The timing of each step is labeled on the top right in minutes and seconds. Scale bars are indicated in each panel.

The online version of this article includes the following source data and figure supplement(s) for figure 1:

**Source data 1.** Number of independent samples were collected for *pezo-1* expression pattern in *C. elegans*.

**Figure supplement 1.** PEZO-1 is expressed in multiple tissues throughout development.

observed (*Figure 2E*, *Figure 2—figure supplement 1E*). Occasionally, a few fertilized embryos were observed inside this mass of ooplasm (data not shown). *pezo-1 CΔ* and STOP-IN mutants displayed the most severe defects, with 100% of animals having a uterus filled with ooplasm at 60 hr post L4 (*Figure 2F*, *Figure 2—figure supplement 1E*). Staining with DAPI in *pezo-1^KO* uteri revealed chromosome structures that were indicative of diakinesis-staged oocytes (*Figure 2H*). Sperm chromatin was not clearly observed, so we cannot state for certain that these crushed oocytes were not fertilized. By contrast, only mitotic chromatin of variably aged embryos were detected in control animals (*Figure 2G*). Consistent with this observation, only unfertilized oocytes and newly fertilized embryos without intact eggshells were stained with the lipophilic dye, BODIPY, in wildtype animals (*Figure 2I*). BODIPY staining revealed widespread penetration of the entire ooplasmic mass in the uteri of *pezo-1 CΔ* animals (*Figure 2J*). These data suggest that some oocytes are not fertilized upon transit through

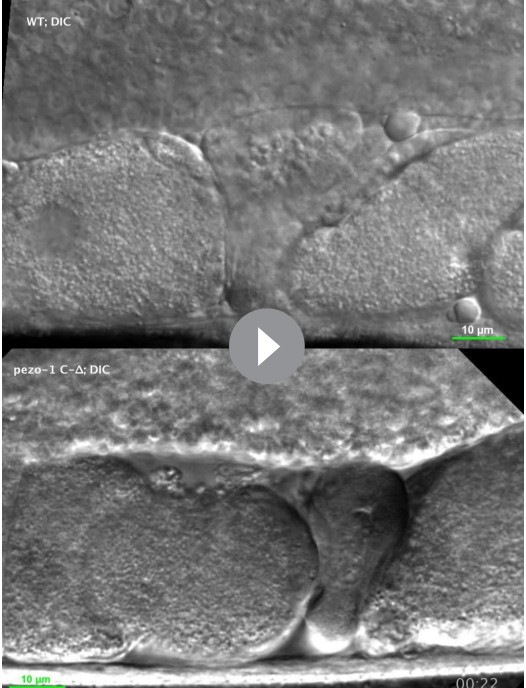

**Video 2.** Crushed oocyte phenotype frequently occurs in the *pezo-1 CΔ* mutant. Time-lapse video recording showing a wildtype oocyte (top panel) entering into the spermatheca and completing fertilization in 5 min. The constricted spermatheca smoothly expels the oocyte into the uterus. White arrows in the top panel indicate an opening spermathecal valve. In the bottom panel, the *pezo-1 CΔ* oocyte successfully enters the spermatheca, but the oocyte is crushed by the sp-ut valve and the ooplasmic debris is observed in the uterus. Yellow arrows in the bottom panel indicate the spermathecal valve. Images are single z planes taken every 2 s. Timing is indicated in lower right. Playback rate is 15 frames/second. Scale bars are indicated in each panel.

https://elifesciences.org/articles/53603#video2

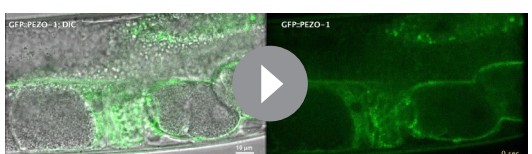

**Video 1.** PEZO-1 expression pattern during ovulation. Ovulation imaged in the genome-edited animals expressing GFP::PEZO-1 (green). The yellow arrow in the right panel indicates GFP::PEZO-1 expression on the spermathecal valves. White arrows in the right panel indicate GFP::PEZO-1 expression on the bag cells. After fertilization, GFP::PEZO-1-labeled sperm crawled back to the spermatheca. The left panel shows the merged channel of DIC (grey) with GFP (green). The right panel indicates the GFP (green) channel only. Images are single z planes taken every 2 s. Timing is indicated in the lower right panel. Playback rate is 15 frames/second. A scale bar is shown in the left panel.

https://elifesciences.org/articles/53603#video1

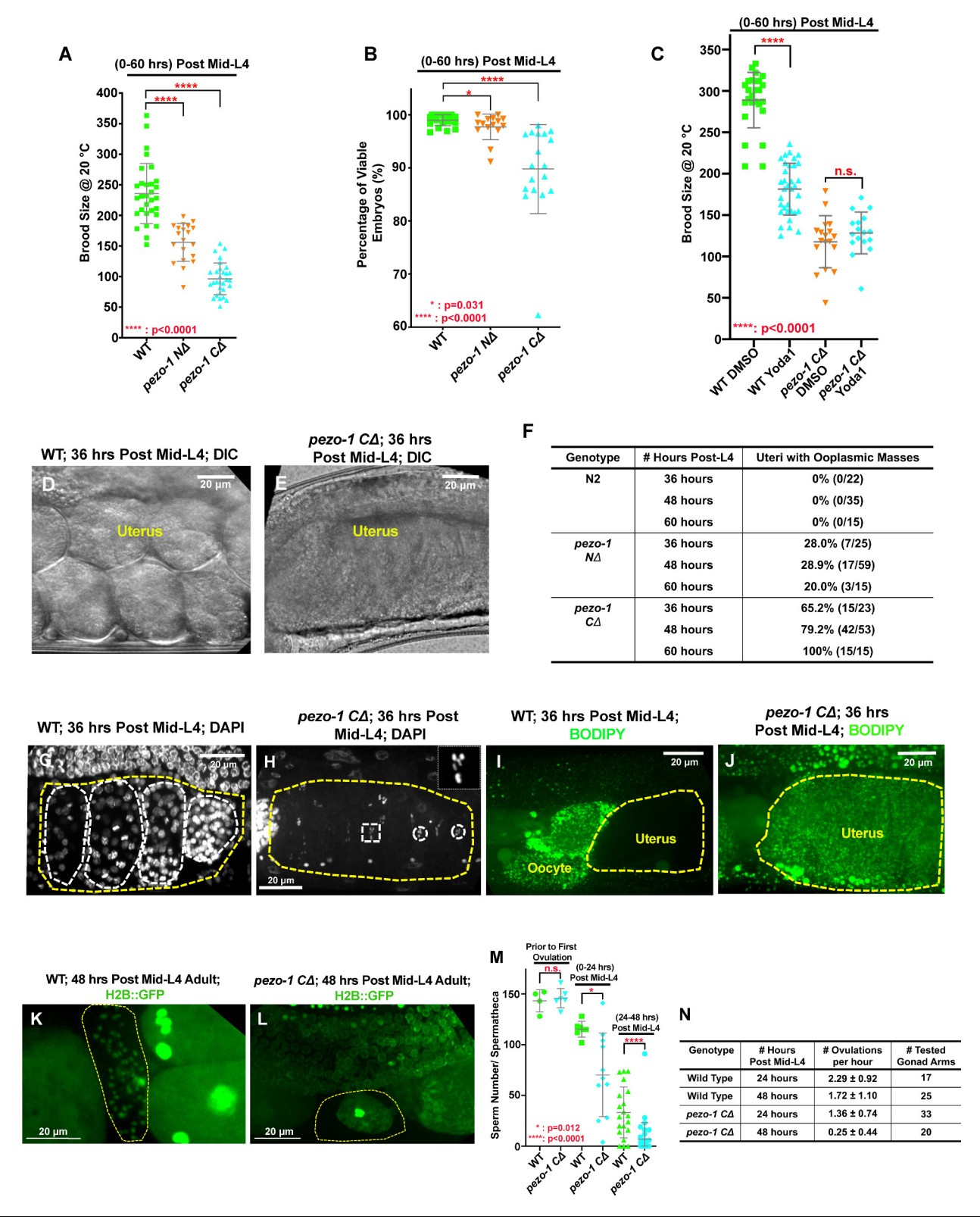

**Figure 2.** Deletions of the *pezo-1* gene cause a reduction in brood size. (**A**) Brood size was significantly reduced in both *pezo-1 NΔ* and *pezo-1 CΔ* animals when compared with wildtype, and this reduction was most evident in older adult animals. (**B**) The percentage of viable embryos was reduced in the *pezo-1 CΔ* animals. (**C**) Dietary supplementation of a PIEZO1-channel-specific activator Yoda1 in wildtype animals significantly reduced the brood size compared with control treatment, but brood size was not further reduced in *pezo-1 CΔ* when treated with Yoda1. (**D, E**) DIC images of the uteri of *Figure 2 continued on next page*

*Figure 2 continued*

gravid adult animals. Wildtype animals had young embryos in their uteri (D), whereas only a large ooplasmic mass was observed in *pezo-1 C*Δ mutant uteri (E). (F) Quantification of the percentage of uteri with ooplasmic masses in wildtype and *pezo-1* deletion mutants. N2 is the wildtype strain. (G, H) DAPI staining demonstrated that multicellular embryos (white circles in panel [G]) were present in the uteri of wildtype animals, whereas only oocyte meiotic chromosomes (white circles and rectangle) were observed in the uteri of *pezo-1 C*Δ mutants (panel [H]; inset in the top right white box shows an amplified image of the meiotic chromatin marked with a white rectangle). The yellow dotted lines indicate the boundaries of the uteri in panels (G) and (H). (I, J) Only unfertilized oocytes and newly fertilized zygotes are permeable to BODIPY (green) in wildtype (WT) animals (I), whereas staining was observed throughout the entire uterine mass (yellow circle in panel [J]) of *pezo-1 C*Δ animals. (K, L) An H2B::GFP transgene was crossed into our strains to visualize oocyte and sperm chromatin. (K) Sperm labeled by H2B::GFP (green cells in yellow circle) reside in the spermatheca (yellow circle) of Day 2 adults (48 hr post mid-L4). (L) Only oocyte debris (yellow circle) is left in the spermatheca of an age-matched *pezo-1 C*Δ mutant. (M) Quantification of sperm counts in both wildtype and *pezo-1 C*Δ hermaphrodites at different time windows. (N) Quantification of the oocyte ovulation rate of wildtype and *pezo-1 C*Δ adults at different ages. The oocyte ovulation rate was significantly reduced in the older *pezo-1 C*Δ mutant adults. P-values: *, p=0.031 (B); *, p=0.012(M); ****, p<0.0001 (*t*-test).

The online version of this article includes the following source data and figure supplement(s) for figure 2:

**Source data 1.** Quantification data describing brood size, the percentage of viable embryos and sperm counts of *pezo-1* mutants compared with wildtype.

**Figure supplement 1.** Verification of CRISPR/Cas9-generated deletions in *pezo-1* knockout animals.

**Figure supplement 1—source data 1.** Quantification data describing brood size and the percentage of viable embryos of *pezo-1* mutants compared with wild-type.

the spermatheca and that these unfertilized oocytes may be crushed when they pass through the spermathecal valves. Although these crushed oocyte phenotypes are reminiscent of those observed in animals depleted of some eggshell components (*Johnston et al., 2010*), there are notable differences. The *pezo-1* mutant oocytes are not fertilized and do not make an eggshell. The lack of fertilization or eggshell synthesis is not likely to be responsible for the crushed oocyte phenotype, because the oocytes in *spe* mutants survive spermatheca transit and are often laid after passing through the uterus. A more detailed characterization of the ovulation defects is provided below.

In addition to these apparently crushed oocytes, reduced numbers of sperm resident in the spermatheca were observed in Day 1 *pezo-1* adults (0–24 hr post mid-L4) and even fewer were observed in the spermathecae in Day 2–3 adults (24–48 hr post mid-L4) compared with wild type (*Figure 2K–M*). Normal numbers of sperm were present in these mutant hermaphrodites prior to the first ovulation, suggesting that the ability of the sperm to return to the spermatheca after each ovulation was disrupted (*Figure 2M*). Sperm loss could also contribute to the low brood sizes observed in our *pezo*-1 mutants.

Ovulation rates were significantly reduced in *pezo-1 C*Δ Day 2 (post mid-L4 48 hr) animals (*Figure 2N*), which is consistent with the reduced brood sizes that worsen in Day 2 animals. As the presence of sperm in the spermatheca is known to stimulate ovulation (*McCarter et al., 1999*; *Miller, 2001*), the reduction in sperm number could be responsible for this reduction in ovulation rate. Overall, the reduced brood size in *pezo-1* mutants is probably due to a combination of defects in multiple tissues, resulting in defective ovulations, crushed oocytes, and defects in the ability of sperm to navigate back into the spermatheca after each ovulation.

To characterize the transit of oocytes through the spermatheca carefully, we performed live imaging to record the ovulation and fertilization process in both wildtype and *pezo-1*^KO^ animals (*Figure 3A–E'*, *Videos 2* and *3*). The imaging began with the mature oocyte entering the spermatheca, labeled by the apical junction marker DLG-1::GFP (*Figure 3A,B*). In wildtype animals, the contracting sheath cells push the oocyte into the spermatheca, and simultaneously pull the open spermatheca over the oocyte (*Videos 2* and *3*). Once the oocyte enters the spermatheca, both spermatheca valves remain closed during fertilization (*Figure 3C*). Opening of the sp-ut valve allows the fertilized oocyte to be expelled into the uterus (*Figure 3D,E*). In *pezo*-1 mutants, many of the oocytes that did successfully enter the spermatheca were crushed when they exited through the sp-ut valve (*Figure 3A'–E'*, *Videos 2* and *3*). We observed that the sp-ut valve, labeled by DLG-1::GFP, did not completely open when the oocyte attempted to exit the spermatheca, which may lead to crushing the oocyte (*Figure 3C'–E'*, *Video 3*). The ooplasm from the crushed oocytes accumulated in the uterus (*Figure 3E'*, *Video 3*) as a large ooplasmic mass (as shown in *Figure 2E*). During our analysis of the *pezo-1* mutants, we frequently observed that oocytes partially entered the

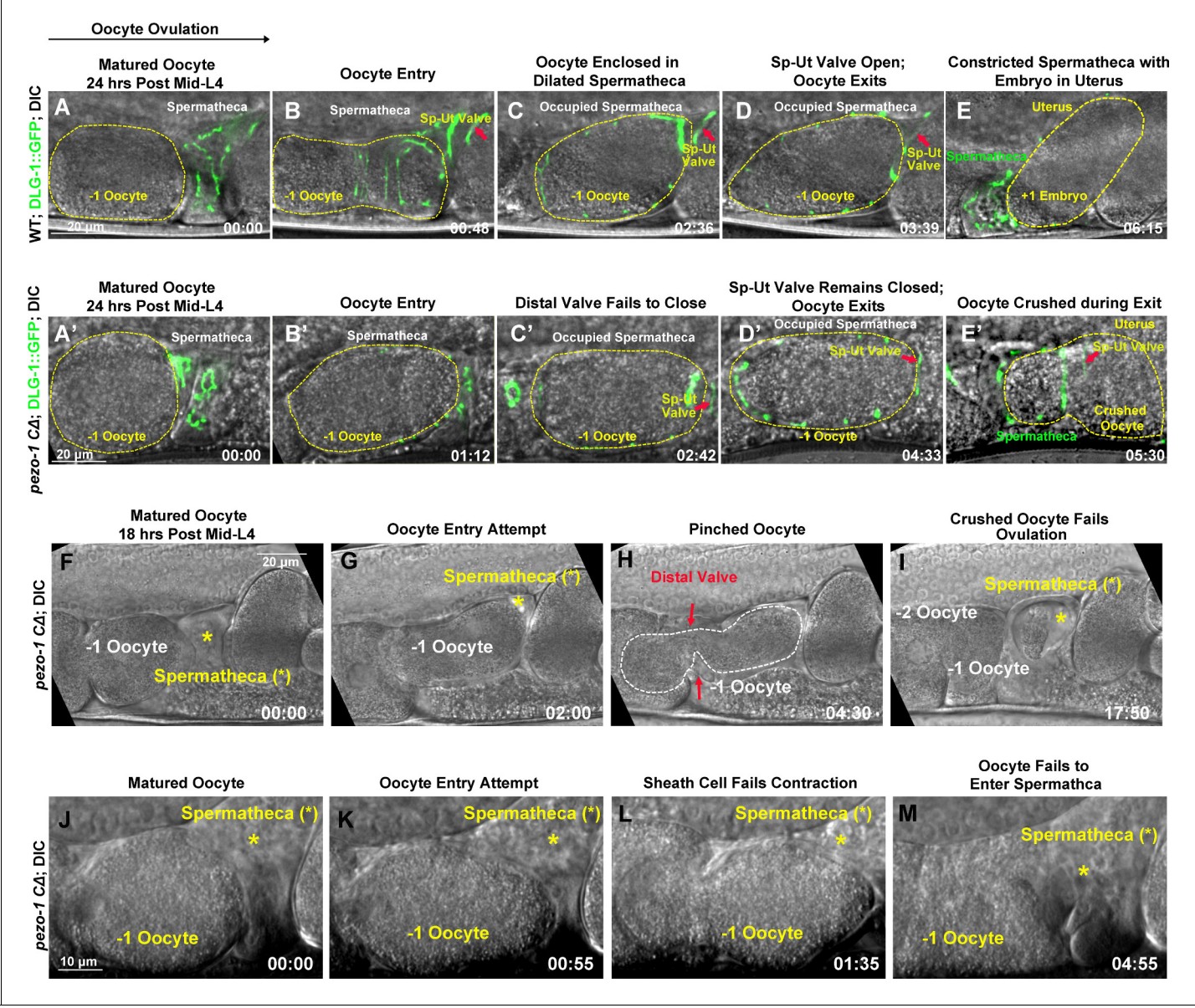

**Figure 3.** PEZO-1 mutants exhibit severe ovulation defects. (A–E) Ovulation in wildtype animals. (A, B) Ovulation is initiated by oocyte (yellow dotted circle) entry into the spermatheca, which was labelled by the apical junctional marker DLG-1::GFP (green). (C) Fertilization occurs in the occupied spermatheca (yellow dotted circle). (D, E) After fertilization, the sp-ut valve (red arrows) opened immediately to allow the newly fertilized zygote (yellow dotted circle) to exit the spermatheca and enter the uterus. (A'–E') Abnormal ovulation was observed in *pezo-1 CΔ* animals. Control of the spermathecal valves was aberrant (C'–E') during ovulation and the DLG-1::GFP labelled sp-ut valve (red arrow) never fully opened; the oocyte was crushed as it was expelled (E'). (F–M) Two examples of ovulation defects observed in the *pezo-1 CΔ* mutants. (F–I) The ovulating oocyte (white dotted circle) was pinched off by the spermathecal distal valve (red arrows in panel [H]). This oocyte never exited into the uterus. (J–M) *pezo-1 CΔ* oocytes frequently failed to enter the spermatheca and were retained in the oviduct (M). The black arrow above panel (A) shows the direction of embryo travel through the spermatheca from left to right. All four image time series follow this same left to right orientation. The timing of each step is labeled on the bottom right in minutes and seconds. Scale bars are shown in each panel.

The online version of this article includes the following source data for figure 3:

**Source data 1.** Number of independent samples were collected for imaging ovulation defects in *pezo-1* mutants.

spermatheca but were then pinched off and broken into two pieces, one of which remained trapped in the oviduct (proximal gonad; *Figure 3F–I*, *Video 4*). Moreover, some oocytes failed to enter the spermatheca and slid back into the oviduct (*Figure 3J–M*, *Video 5*). The defective ovulation is probably due to incomplete constriction of the sheath cells. Overall, disrupted ovulation and oocyte

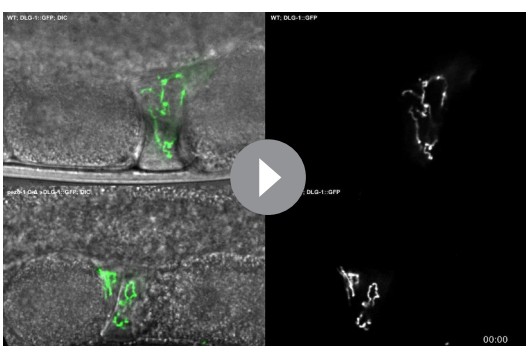 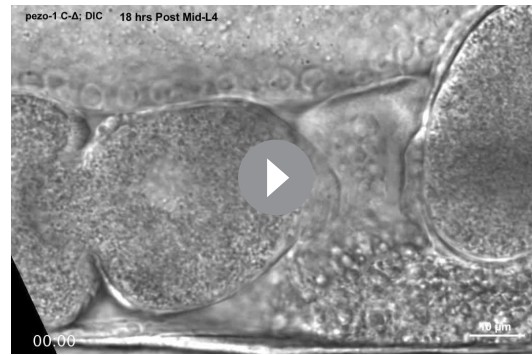

**Video 3.** The sp-ut valve fails to open during spermathecal contraction. Time-lapse recordings on the left are of DIC and GFP. Recordings on right are of GFP alone. Oocyte entry occurs from the left at the 15 s mark. The spermatheca was labelled by the apical junctional marker DLG-1::GFP. In the wild type (top panels), the sp-ut valve (white arrow) opened immediately to allow the oocyte to be expelled into the uterus (on the right). In *pezo-1 C∆* (bottom panels), however, the DLG-1::GFP labelled sp-ut valve (white arrow) never fully opened, the oocyte was crushed as it was expelled, and ooplasmic debris was pushed out into the uterus. Images are single z planes taken every 3 s. Timing is indicated in the bottom right corner. Playback rate is 15 frames/second. Scale bars are shown in each DIC panel.

https://elifesciences.org/articles/53603#video3

**Video 4.** Spermatheca dilation is defective in *pezo-1* mutants. Time-lapse video recording (DIC). Oocyte entry occurs from the left at the 35-s mark. The distal valve was not able to close completely and the oocyte was pinched. One portion of the broken oocyte was left in the spermatheca, the other portion remains in the oviduct (white arrows, left panel). Images are single z planes taken every 2 s. Timing is indicated in the bottom left corner. Playback rate is 15 frames/second. A scale bar is shown in the bottom right corner.

https://elifesciences.org/articles/53603#video4

transit defects were observed in *pezo-1* mutants, consistent with the decreased brood size observed in all of our *pezo-1* mutants.

## PEZO-1 mutants are affected upon depletion of cytosolic Ca²⁺

Given that PEZO-1 is the ortholog of mammalian mechanosensitive calcium channels and that Ca²⁺

### regulators

Given that PEZO-1 is the ortholog of mammalian signaling is a major regulator of *C. elegans* spermathecal contractility, we tested whether there was suppression or enhancement when *pezo-1* mutants were combined with the depletion of several important cytosolic Ca²⁺ regulators. To manipulate potential calcium signaling, an ER Ca²⁺ release channel, ITR-1, and an inositol-1,4,5-triphosphate (IP₃) kinase, LFE-2, were depleted by RNAi in both wildtype and *pezo-1* mutants. IP₃ binding to ITR-1 releases Ca²⁺ from the ER, which activates myosin for spermathecal contractility (*Bouffard et al., 2019*; *Clandinin et al., 1998*; *Kovacevic et al., 2013*). Therefore, we hypothesized that combining *pezo-1* mutants with *itr-1* RNAi would greatly enhance the reduction in brood size if they were both critical to ovulation and fertilization. We carefully calibrated *itr-1* RNAi treatment and determined that feeding L4 animals for 36–60 hr produced optimal intermediate conditions that caused minimal developmental defects and normal brood sizes in wildtype animals. Consistent with our hypothesis, feeding *itr-1* RNAi resulted

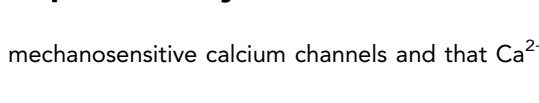
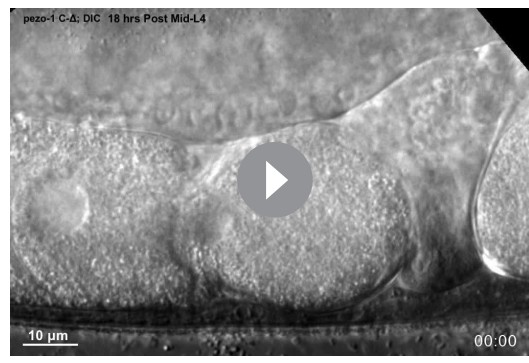

**Video 5.** Sheath cell contraction is defective in *pezo-1* mutants. Time-lapse video recording (DIC). An oocyte fails to enter the spermatheca after a few attempts. Sheath cells fail to contract and push the oocyte into the spermatheca (on the right) and the oocyte moves left, back into the oviduct. Images are single z planes taken every 2 s. Timing is indicated in the bottom right corner. Playback rate is 15 frames/second. A scale bar is shown in the bottom left corner.

https://elifesciences.org/articles/53603#video5

in even smaller broods than those observed in *pezo-1* mutants alone (*Figure 4A*). By contrast, feeding *lfe-2* RNAi, which should elevate cytosolic $Ca^{2+}$, partially rescued the reduced brood size (*Figure 4B*). Therefore, *pezo-1^{KO}* mutants were further compromised with *itr-1* (RNAi), yet partially rescued when combined with *lfe-2* (RNAi). Similarly, depletion of the plasma membrane $Ca^{2+}$ channel *orai-1*, which is activated to replenish $Ca^{2+}$ in the cytosol from an extracellular source (*Lorin-Nebel et al., 2007*), led to nearly zero brood size in *pezo-1 CΔ* mutant but only a 40% reduction in brood size in wild type (*Figure 4C*). Furthermore, disruption of ER $Ca^{2+}$ stores with sarcoplasmic/ER $Ca^{2+}$ ATPase (SERCA) *sca-1* (RNAi) (*Yan et al., 2006*) also caused an extremely low brood size in *pezo-1 CΔ* (*Figure 4C*), whereas *sca-1* (RNAi) slightly increased the brood size in wild type (*Figure 4C*). Therefore, these observations are consistent with the hypothesis that *pezo-1* may function in cytosolic and ER $Ca^{2+}$ homeostasis, which is crucial for proper spermathecal contractility and dilation. *pezo-1* mutants show normal calcium signaling in spermatheca cells during ovulation.

Owing to the permeability of PIEZO channels to $Ca^{2+}$ and the importance of calcium signaling in regulating spermathecal contractility, we tested whether the deletion of *pezo-1* disrupted cytosolic $Ca^{2+}$ homeostasis. We imaged oocyte passage through the spermathecae of both wild type and *pezo-1* mutants expressing the $Ca^{2+}$ indicator GCaMP3, which was driven by a spermatheca-specific *fln-1* promoter (*Bouffard et al., 2019*; *Kovacevic et al., 2013*). Co-localization of the GCaMP3 transgene with mScarlet::PEZO-1 in the spermatheca suggested that this transgene would be useful for the analysis of *pezo-1* function in spermathecal calcium signaling (*Figure 5A–E*, *Video 6*). To determine whether calcium signaling was altered in our *pezo-1* mutants, a set of high-speed GCaMP imaging data from different animals was generated and the average pixel intensity of each frame was quantified (*Figure 5F–J'*, *Figure 5—figure supplement 1A–D*, *Video 6*). We defined the initial time frame as the time just before the oocyte entered the spermatheca. In wildtype animals, the

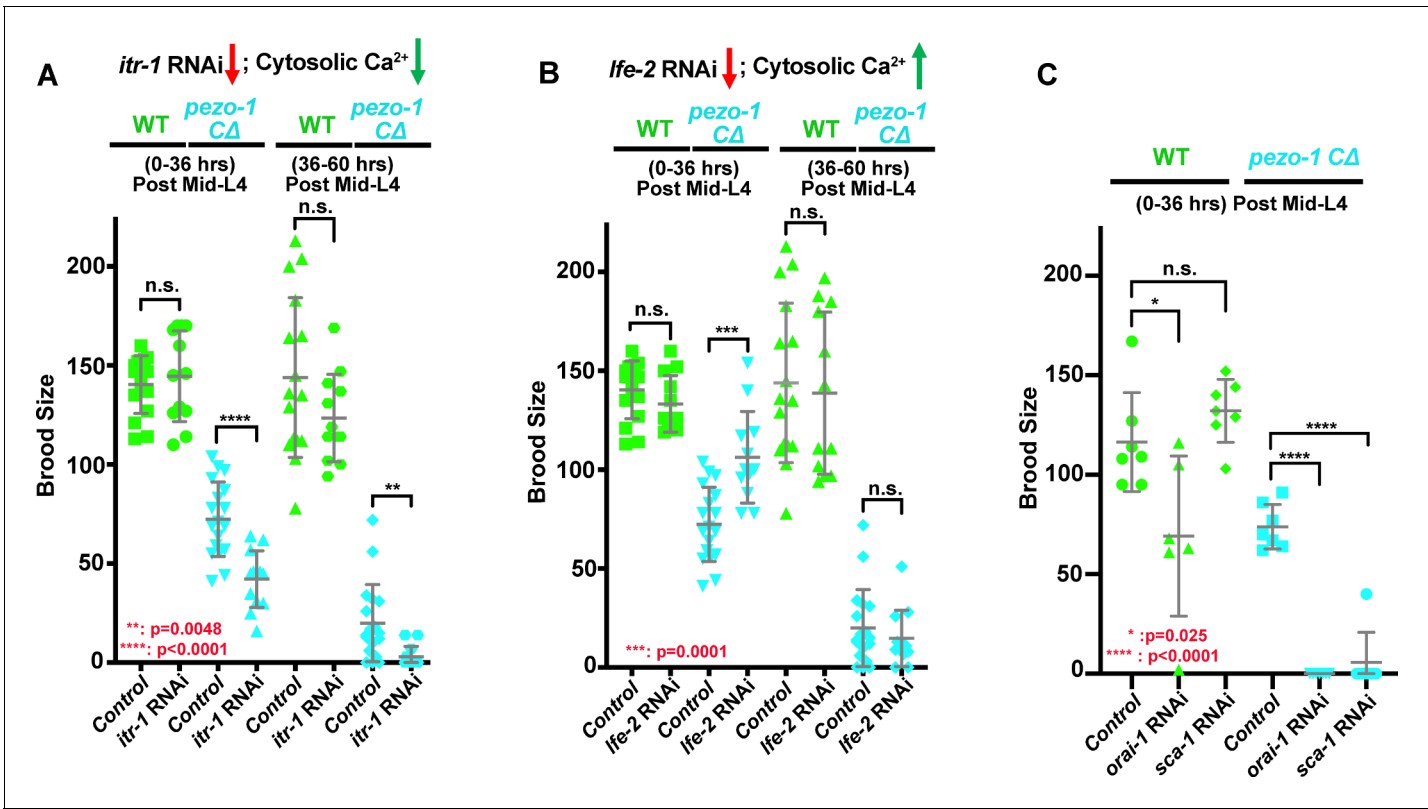

**Figure 4.** *pezo-1* mutants show genetic interactions with cytosolic $Ca^{2+}$ regulators. (**A**) *itr-1* (RNAi) reduced the brood size in *pezo-1 CΔ* animals. (**B**) By contrast, *lfe-2* (RNAi) slightly rescued the smaller brood size in *pezo-1 CΔ* animals. (**C**) Depletion of both *orai-1* and *sca-1* by RNAi also enhanced the brood size reduction of *pezo-1 CΔ* mutants. P-values: *, p=0.025 (C); **, p=0.0048 (A); ***, p=0.0001 (B); ****, p<0.0001 (t-test).
The online version of this article includes the following source data for figure 4:

**Source data 1.** Quantification of brood size for genetic interaction of *pezo-1* mutants with RNAi depletion of calcium regulators.

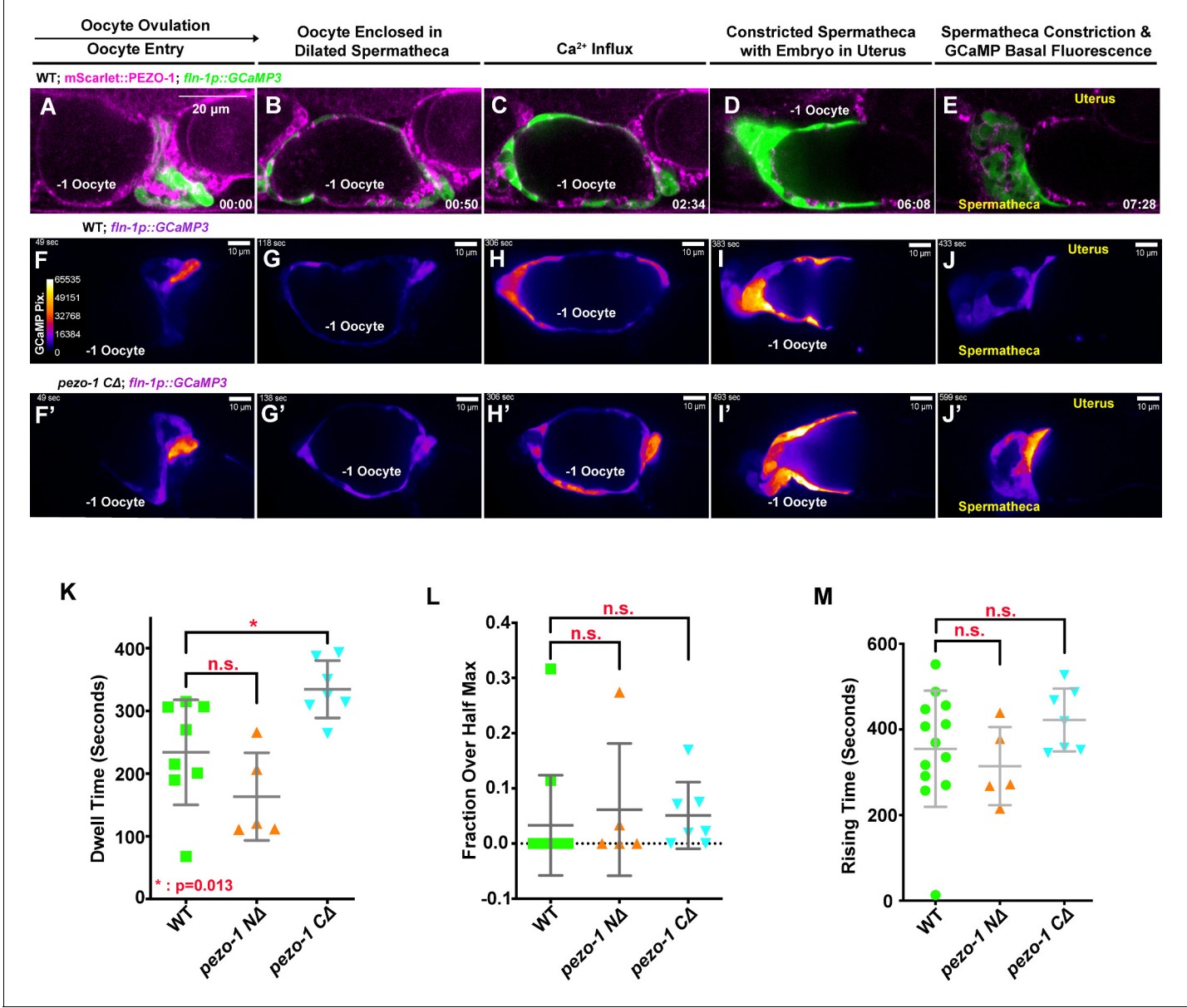

**Figure 5.** PEZO-1 mutants show normal GCaMP3 fluorescence during ovulation. (**A–E**) mScarlet::PEZO-1 colocalizes with GCaMP3, which is driven by a spermatheca-specific promoter. These images represent the third ovulation for this spermatheca. (**F–J'**) Time series frames from GCaMP3 recordings in the third ovulation of both wildtype animals (**F–J**) and *pezo-1 CΔ* animals (**F'–J'**). Ca$^{2+}$ influx was quantified during ovulation and fertilization, as indicated by the intensity of GCaMP3 pixels (colored bar in panel [F]). (**F, F'**) Oocyte entry into the spermatheca in wildtype and *pezo-1 CΔ*. (**G, G'**) Oocytes in the spermatheca, (**H, H'**) Ca$^{2+}$ influx during fertilization, (**I, I'**) intense Ca$^{2+}$ influx as the sp-ut valve closes to push newly fertilized zygote into the uterus, and (**J, J'**) the return to basal levels as the spermatheca prepares for the next ovulation. (**K**) Dwell time is a tissue function metric calculated as the time the oocyte resides in the spermatheca from the closing of the distal valve to the opening of the sp-ut valve. (**L, M**) Calcium signaling metrics: fraction over half max (**L**) and rising time (**M**) in *pezo-1* mutants showed normal calcium levels during ovulation compared with wild type (*Bouffard et al., 2019*). The black arrow above panel (**A**) shows the direction of embryo travel through the spermatheca from left to right. All three image time series follow this same left to right orientation. The timing of each step is labeled in the bottom right in minutes and seconds (**A–E**), or on the top left in seconds (**F–J'**). Scale bars are shown in each panel.

The online version of this article includes the following source data and figure supplement(s) for figure 5:

**Source data 1.** Quantification of calcium metrics in *pezo-1* mutants and wild-type.

**Figure supplement 1.** Normal calcium signaling was observed in the spermathecal cells in *pezo-1* mutants.

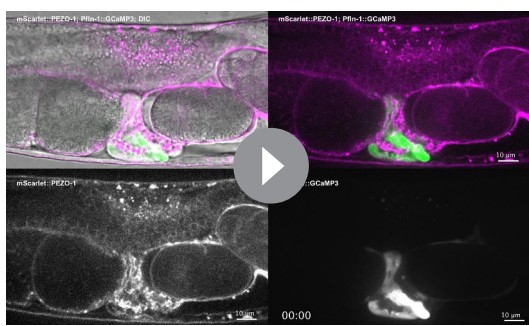

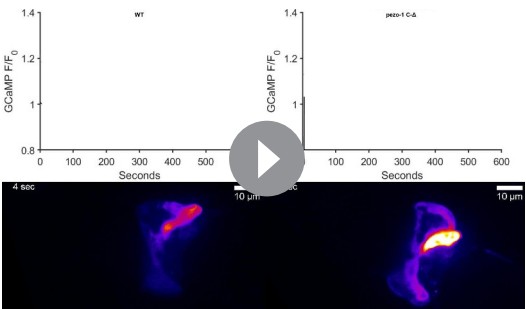

**Video 6.** mScarlet::PEZO-1 colocalizes with spermathecal-specific GCaMP3. Example of the colocalization of mScarlet::PEZO-1 (magenta) with the Pfln-1::GCaMP3 transgene (green) in the spermathecal cells in a wildtype animal. The top left recording shows the merged channel of DIC (grey), mScarlet::PEZO-1 (magenta) and the *Pfln-1::GCaMP3* transgene (green). The top right panel lacks the DIC channel. The bottom left recording shows just the mScarlet::PEZO-1 expression pattern during ovulation. The bottom right video indicates that Pfln-1::GCaMP3 only displays the changes in GCaMP3 intensity, which are indicative of calcium influx. Images were acquired in a single z plane every 2 s. Timing is indicated in the lower right panel. Playback rate is 30 frames/second. Scale bars are shown in each panel.

https://elifesciences.org/articles/53603#video6

**Video 7.** Normal GCaMP3 influx was observed in *pezo-1* mutants. Examples of GCaMP3 recordings of embryo transits in wildtype (left panels) and *pezo-1 CΔ* (right panels) animals. Recordings were temporally aligned to the start of oocyte entry at 50 s. GCaMP3 normalized average pixel intensity ($F/F_0$, top, Y-axis) versus GCaMP3 time (top, X-axis) generated from GCaMP3 recordings, with highlighted metrics shown on the top of the tracings. Dwell time is a tissue function metric that represents the duration from the closing of the distal valve to the opening of the sp-ut valve, rising time is a calcium signaling metric measuring the time from the opening of the distal valve to the first time point at which the time series reaches half maximum of GCaMP3 intensity, and fraction over half max is a calcium signaling metric, which measures the duration of the dwell time over the GCaMP3 half-maximal value divided by the total dwell time. Images were acquired in a single z plane every 1 s. Timing is indicated in the top left corners of the two lowerhe panels. Playback rate is 30 frames/second. Scale bars are shown in these panels.

https://elifesciences.org/articles/53603#video7

fluorescent intensity of GCaMP3 at the sp-ut valve immediately increased when the oocyte entered the spermatheca (*Figure 5A,F and F'*, *Videos 6* and *7*). During fertilization, an increase in intensity of GCaMP3 was frequently observed in the bag cells and the sp-ut valve until the oocyte exited the spermatheca (*Figure 5B–D,G–I and G'–I'*, *Videos 6* and *7*). The GCaMP3 signal decreased to basal intensity after the fertilized oocyte was expelled into the uterus (*Figure 5E,J and J'*, *Videos 6* and *7*). To quantify statistically and to analyze the oocyte transit, we defined a series of parameters, including the dwell time and two calcium signaling metrics from the GCaMP3 time series (*Bouffard et al., 2019*). A spermathecal tissue function metric, dwell time, is defined as the time from spermathecal distal valve closure to sp-ut valve opening, which represents the time during which the oocyte resides in the enclosed spermatheca. The calcium signaling metric, fraction over half max, is defined as the duration of the dwell time over the GCaMP3 half-maximal value divided by the total dwell time. The fraction over half max allows us to capture the relative level of calcium throughout the time during which the embryo passes through the spermatheca. Rising time indicates the time from the opening of the distal valve to the first time point at which the GCaMP fluorescent intensity reaches half maximum (*Bouffard et al., 2019*). In *pezo-1 CΔ* mutants, longer transit times of the oocyte through the spermatheca resulted in elongated dwell times (*Figure 5K*, *Video 7*), suggesting that deletion of *pezo-1* resulted in disrupted tissue function. Surprisingly, GCaMP3 fluorescence in *pezo-1* was not significantly different from that in the wildtype (*Figure 5L,M*, *Video 7*; see 'Materials and methods'). GCaMP3 time series (*Figure 5—figure supplement 1A,B*, *Video 7*), heat maps (*Figure 5—figure supplement 1C*), and kymograms (*Figure 5—figure supplement 1D,E*) also displayed normal $Ca^{2+}$ levels during oocyte passage through the spermatheca in *pezo-1* mutants. It should be noted that we only imaged the GCaMP3 reporter during the very first three ovulations in young adult animals to avoid $Ca^{2+}$ signaling interference from a distorted gonad morphology and mechanical pressure from a gravid uterus. Furthermore, it is difficult to monitor older *pezo-1*

hermaphrodites as they do not ovulate on microscope slides. As only mild defects were observed in the *pezo-1* mutants during these early ovulations and oocyte transit defects increased in severity over time (*Figure 2F*), our data does not exclude the possibility that Ca$^{2+}$ signaling may be more severely disrupted as the animal goes through more ovulation cycles. Alternatively, the live imaging assay may not be sensitive enough to detect subtle variations in calcium signaling.

## Sperm from matings rescues the low brood size phenotype in *pezo-1* mutants

In *C. elegans*, successful ovulation and fertilization requires signal coordination between sperm, oocytes, and sheath cells (*Han et al., 2010*). Given that PEZO-1 is expressed in these tissues, it is plausible that oocyte transit defects and reduced brood sizes are the result of impaired inter-tissue signaling, which may be mediated by PEZO-1. To investigate how this may occur, bidirectional signaling between sperm and oocytes was first tested. To test for the ability of sperm to fertilize oocytes, both wildtype and *pezo-1* mutant males were mated with *fem-1(hc17)* hermaphrodites, which do not produce any sperm or self-progeny (*Doniach and Hodgkin, 1984*) and are essentially females. The *fem-1(hc17)* animals produced cross-progeny after mating with *pezo-1* mutant males, indicating that the *pezo-1* mutant males are fertile and that their sperm can crawl through the uterus to the spermatheca upon mating (*Figure 6A*). As *pezo-1* mutant hermaphrodites do not produce any self-progeny after Day 3 (60 hours post mid-L4) (*Figure 6B*), we tested whether mating with either wildtype or mutant males would result in any cross progeny in the aged *pezo-1* mutants. *pezo-1* mutant hermaphrodites resumed ovulation and fertilization upon mating once the male's sperm (from either wildtype or *pezo-1* males) reached the spermatheca (*Figure 6B–D*). To test whether sperm signaling was defective in inducing ovulation in *pezo-1* mutants, we mated both *spe-9(hc52ts)* and control *him-8(e1489)* males with both wildtype and *pezo-1* mutant hermaphrodites. *spe-9(hc52ts)* male sperm can physically contact the oocytes but fail to fertilize them, although the sperm signaling is apparently normal and triggers ovulation (*Singson et al., 1998*). Interestingly, the low ovulation rate in older *pezo-1 CΔ* animals was significantly rescued by *spe-9(hc52ts)* sperm (*Figure 6E*), although the ovulated oocytes were not fertilized. An additional experiment was performed to test the ability of the sheath to respond to the sperm signal that triggers ovulation. Even though our data in *Figure 6E* suggest that just the presence of sperm can trigger ovulation, we went on to show that purified MSP-fluorescein can also trigger ovulation in older *pezo-1 CΔ* hermaphrodites that are depleted of sperm and are no longer ovulating (*Figure 6F–H*). Overall, these data suggest that the absence of self-sperm contributes to a profound reduction of oocyte maturation, ovulation rate, and self-fertility in the aged *pezo-1* mutants.

## Sperm guidance and navigation is disrupted in *pezo-1* mutants

In wildtype hermaphrodites, the sperm are constantly being pushed out of the spermatheca each time the sp-ut valve opens to expel the fertilized oocyte into the uterus. These sperm, however, are fully capable of crawling back to the spermatheca to induce high levels of oocyte maturation and ovulation (*Miller, 2001*; *Miller et al., 2003*). This is a very efficient mechanism, such that almost every self-sperm in a hermaphrodite is used to fertilize an oocyte. It is sperm number that defines brood size; oocytes are in excess. Oocytes secrete F-series prostaglandins derived from polyunsaturated fatty acids (PUFAs) to guide sperm to the spermatheca (*Han et al., 2010*; *Kubagawa et al., 2006*). To test whether *pezo-1* hermaphrodites fail to attract the sperm back to the spermatheca, male sperm navigational performance was assessed in vivo by staining males with a vital fluorescent dye, MitoTracker CMXRos, which efficiently stains sperm in live animals (*Whitten and Miller, 2007*). Both wildtype and *pezo-1 CΔ* stained males were mated to non-labeled wildtype hermaphrodites for 30 min. The sperm distribution was assessed and quantified by dividing the uterus into three zones (*Figure 7A*) and counting the number of fluorescent sperm in each zone (*McKnight et al., 2014*) one hour after males were removed from the mating plates. In wildtype hermaphrodites, most fluorescent sperm from both wildtype and *pezo-1 CΔ* males navigated through the uterus and accumulated in the spermatheca (*Figure 7B,C,F,G*). However, fewer fluorescent male sperm reached the spermatheca in Day 3 adult *pezo-1 CΔ* hermaphrodites, and most sperm remained throughout zones 1 and zone 2, the zones furthest from the spermatheca (*Figure 7D,E,H,I*). This was observed for both wildtype and *pezo-1* mutant male sperm in mating with *pezo-1 CΔ* hermaphrodites

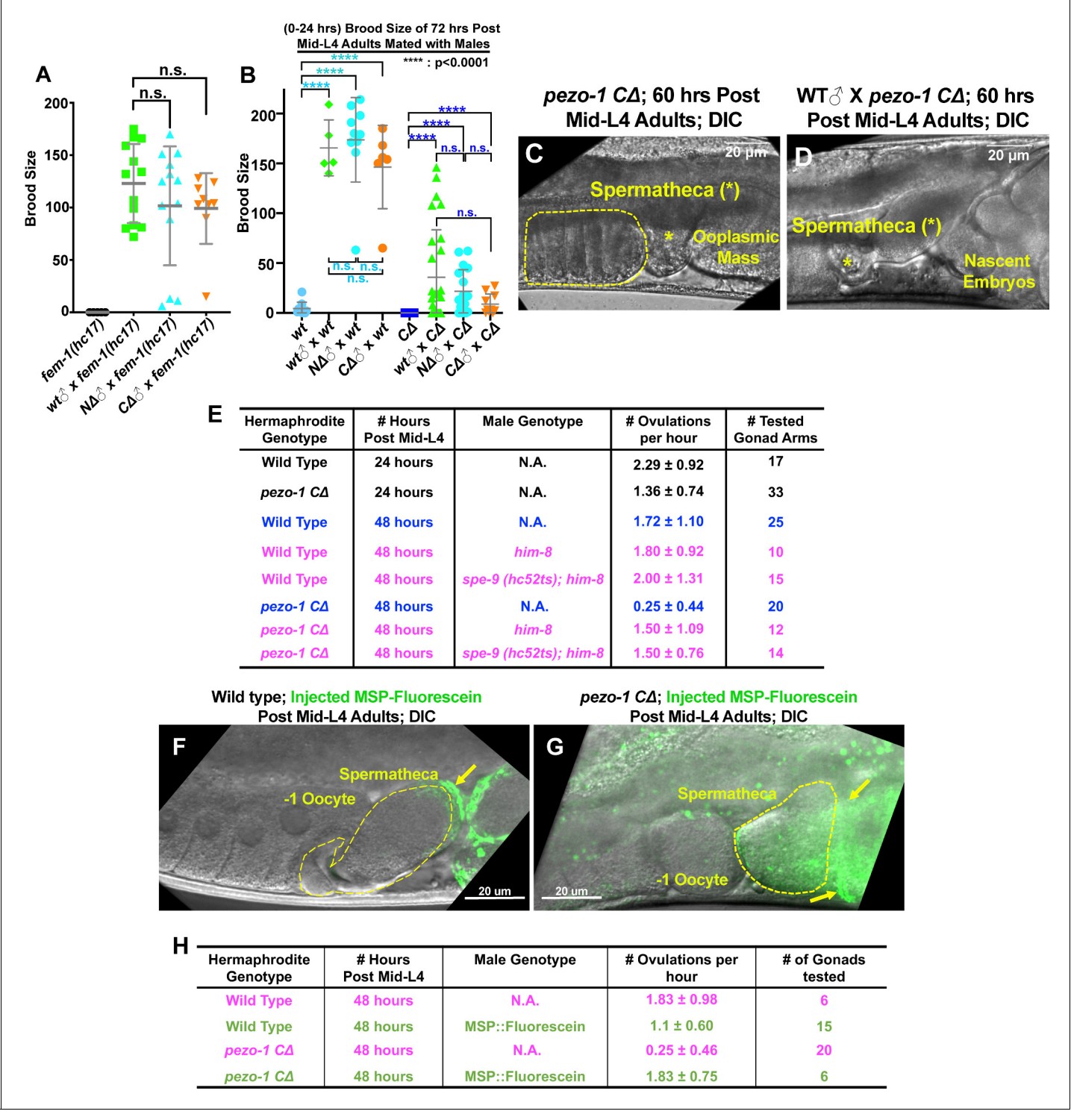

Figure 6. Male sperm rescue the ovulation defects in *pezo-1* mutants. (A) Both *pezo-1 CΔ* and *NΔ* males are fertile and sire progeny when mated with *fem-1(hc17ts)* mutants (essentially female animals). (B) Mating with male sperm rescued fertility in Day 3 *pezo-1 CΔ* adults (72 hr post mid-L4). (C) The oocyte maturation and ovulation rate are very low in Day 3 *pezo-1 CΔ* mutant adults, and oocytes accumulate in the proximal gonad arm (yellow dashed circle). (D) By contrast, the ovulation rates are recovered to high levels after mating with wildtype male sperm. Newly fertilized embryos pushed the ooplasmic mass out of the uterus. Yellow asterisks indicate the spermatheca (C, D). (E) Quantification of the oocyte ovulation rate of wildtype and *pezo-1 CΔ* adults at different ages. *him-8(e1489)* and *spe-9 (hc52ts)* sperm significantly rescue ovulation rates in *pezo-1 CΔ* hermaphrodites, even though they do not fertilize oocytes. (F, G) Injection of purified fluorescein-tagged MSP in the uteri of both wildtype and *pezo-1 CΔ* aged adults. Fluorescein-tagged MSP moved through the entire uterus to localize next to the spermatheca. The yellow dotted circle represents the spermatheca.

*Figure 6 continued on next page*

*Figure 6 continued*

The yellow arrows indicate the fluorescein-tagged MSP (green) localized next to the spermatheca. (**H**) Quantification of the oocyte ovulation rate of wildtype and *pezo-1 C∆* adults without or without injections of fluorescein-tagged MSP. P-values: ****, p<0.0001 (*t*-test). Scale bars are shown in panels (C, D, F, G).

The online version of this article includes the following source data and figure supplement(s) for figure 6:

**Source data 1.** Quantification of sire progeny in different mating assays.

**Figure supplement 1.** Male sperm rescue the fecundity in *pezo-1 C∆* female.

**Figure supplement 1—source data 1.** Quantification of sire progeny and sperm count in different mating assays.

(***Figure 7J***). These observations suggest that in the reproductive tracts of wildtype hermaphrodites, *pezo-1* mutant male sperm are motile and display normal navigational behavior. However, in *pezo-1* mutant hermaphrodite reproductive tracts, both wildtype and *pezo*-1 mutant sperm were compromised in their navigational behavior over the time frame of this experiment. Although it remains possible that the ooplasmic masses that accumulate in the uterus of *pezo-1* mutant hermaphrodites could physically interfere with the migration of wildtype and *pezo-1* mutant sperm back to the spermatheca, our labeled sperm experiments with female *pezo-1* mutants (see below) suggest that this is not a likely explanation.

To test whether the defective ovulation and sperm attraction were just self-sperm problems, we generated the same *pezo-1 C∆* (used throughout this study) in temperature-sensitive *fem-1(hc17ts)* females. In *pezo-1 C∆* female mutants, the number of $F_1$ progeny was significantly reduced compared with that in control *fem-1(hc17ts)* at the permissive temperature of 15°C, which allows for the production of self-sperm (***Figure 6—figure supplement 1A***). We then mated these Day 2 (36 hours post mid-L4) females with both wildtype and mutant males and scored for cross progeny at the nonpermissive temperature of 25°C. The male sperm were labeled by MitoTracker CMXRos before mating. We carefully quantified the number of male sperm in the reproductive tract of the *pezo-1 C∆* females after mating for 30 min (***Figure 6—figure supplement 1B***). All tested female animals sired crossed progeny but at greatly reduced levels in *pezo-1 C∆* females (***Figure 6—figure supplement 1C,D***). This suggests that the attractive signal from the oocytes or sheath cells are defective in their ability to attract male sperm to the spermatheca. Thus, the defect in the ability to attract sperm to the spermatheca is not just a self-sperm problem; cross sperm from males also fail to migrate to the spermatheca.

The data shown in ***Figure 6A and B*** suggest that mutant sperm, when mated with WT hermaphrodites or *fem-1* females, can migrate to the spermatheca and fertilize a large number of oocytes. However, when mated into the *pezo-1 C∆* hermaphrodites, these mutant sperm do sire cross progeny but at greatly reduced levels compared to wildtype male sperm (***Figure 6B***, right side). This result supports the conclusion that an attractive signal from the oocytes or sheath cells is missing or reduced in *pezo-1* hermaphrodites. Thus, we believe that there is no problem with the ability of sperm to crawl and fertilize oocytes.

## Tissue-specific degradation of PEZO-1 reveals multiple roles of PEZO-1 in both somatic tissues and germline cells

Our study aims to reveal the role of PEZO-1 in regulating reproduction and coordinating inter-tissue signaling. To dissect PEZO-1 function in distinct tissues, we utilized an auxin-inducible degradation system (AID) to degrade PEZO-1 in the soma and the germ line (***Zhang et al., 2015***). We knocked-in the degron coding sequence at the *pezo-1* C-terminus using CRISPR/Cas9, so that all isoforms would be targeted (***Figure 8A***). To activate the AID system, this line was then crossed with the strains expressing the degron interactor transgene *tir-1::mRuby* driven by the following promoters: $P_{eft-3}$, $P_{pie-1}$ and $P_{sun-1}$ (***Zhang et al., 2015***). $P_{eft-3}$::*tir-1::mRuby* was expressed in most or all somatic tissues, including the spermatheca and the sheath cells (***Figure 8B***), whereas $P_{pie-1}$::*tir-1::mRuby* and $P_{sun-1}$::*tir −1::mRuby* were expressed in the germ line (***Figure 8C,D***). Weak TIR1-1::mRuby expression was observed in the sperm and oocytes of the germline strains (***Figure 8C,D***, ***Figure 8—figure supplement 1A–C***).

To assess the efficacy of PEZO-1 degradation in different reproductive tissues, we generated a strain in which the *pezo-1* gene was tagged at its N-terminus with GFP and at its C-terminus with

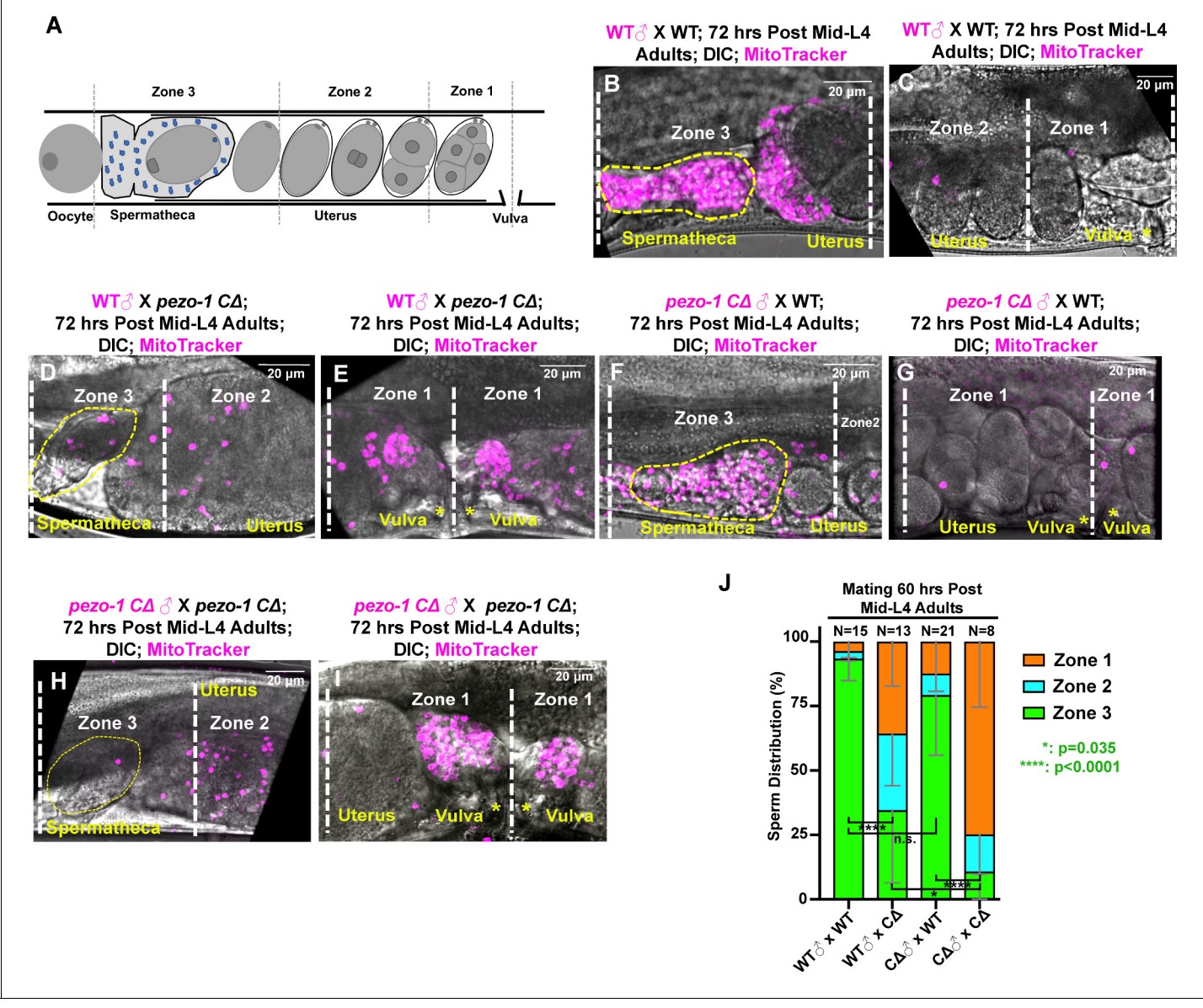

**Figure 7.** Sperm guidance and navigation is disrupted in *pezo-1* mutants. (**A**) To quantify sperm migration, this illustration indicates the three zones that were scored for sperm distribution. Zone 3 is the spermatheca region and the space containing the +1 fertilized embryo (yellow dotted circles in panels (B, D, F, H), whereas Zone 1 is the area closest to the vulva. Sperm distribution is measured 1 hr after males were removed from the mating plate. (**B–I**) The distribution of fluorescent male sperm labeled with MitoTracker in the three zones in both wildtype and *pezo-1* mutants 1 hr after the males were removed. Yellow asterisks indicate the vulva (C, E, G, I). Scale bars are indicated in each panel. (**J**) Quantification of sperm distribution values. The numbers of the scored uteri are shown above each of the bars. P-values: ****, p<0.0001 (*t*-test).

The online version of this article includes the following source data for figure 7:

**Source data 1.** Quantification of sperm count in sperm distribution assays.

the degron (GFP::PEZO-1::Degron). This strain was crossed with the strains expressing tir-1::mRuby driven by the three different promoters described above (*Figure 8—figure supplement 2B–B''*; D–D'', F–F''). GFP::PEZO-1::Degron strongly expresses at the plasma membrane of germline cells, oocytes, sperm, somatic sheath cells, and spermathecal cells (*Figure 8—figure supplement 2A–A''*, *B–B''*, *D–D''*, *F–F'*). The animals were exposed to either 0.25% ethanol as control or 2 mM auxin (indole-3-acetic acid, or IAA) for one generation, and the GFP fluorescent intensity in their $F_1$ progeny was analyzed. The strain expressing the degron interactor transgene $P_{eft-3}::tir-1::mRuby$ had a significant reduction of the fluorescent intensity of GFP::PEZO-1::Degron at the sheath and

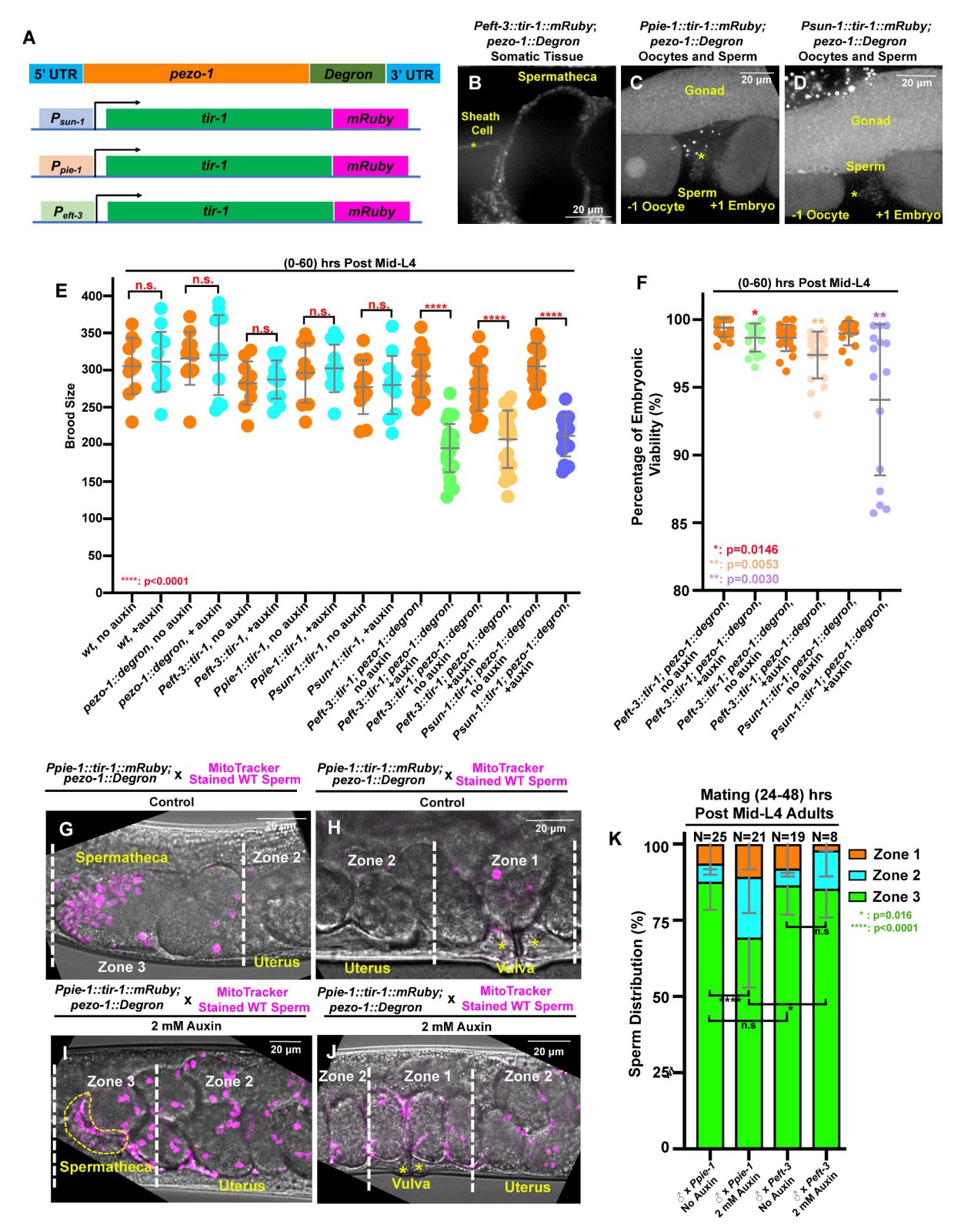

**Figure 8.** Tissue-specific degradation of PEZO-1 causes a reduced brood size and sperm navigational defects. (**A**) Schematic of the auxin-inducible degradation (AID) system. A degron tag was inserted at the 3′ end of the *pezo-1* coding sequence using CRISPR/Cas9-mediated editing. (**B**) The *eft-3* promoter was used to drive TIR-1 expression in most or all somatic tissues, including the spermatheca and the sheath cells. TIR-1::mRuby driven by the germline-specific promoters *sun-1* and *pie-1* is strongly expressed in the germline and oocytes (**C, D**), and weakly expressed in the sperm (asterisks in

*Figure 8 continued on next page*

Figure 8 continued

panels [C, D]). (E, F) Brood size and embryonic viability were reduced in all degron strains when animals were treated with 2 mM auxin. Data are presented as the mean ± standard error from at least two independent experiments. (G–J) Sperm distribution 1 hr after the removal of males from mating plates. The germline-specific PEZO-1::Degron hermaphrodites were mated with wildtype males for 30 min. The representative images show that *pezo-1* degradation in the germ line influences sperm distribution from the vulva (zone 1) to the spermatheca (zone 3). (K) Quantification of sperm distribution in the PEZO-1::Degron strains grown on plates with (+) or without (–) 2 mM auxin. P-values: *, p=0.0146 (F); *. p=0.016 (K); **, p=0.0030 (F); **, p=0.0053 (F); ****, p<0.0001 (E, K) (*t*-test). Scale bars are shown in each micrograph.

The online version of this article includes the following source data and figure supplement(s) for figure 8:

**Source data 1.** Quantification of brood size and sperm counts in each AID strain.
**Figure supplement 1.** Expression pattern of *tir-1::mRuby* in reproductive tissues.
**Figure supplement 2.** Tissue-specific degradation of PEZO-1 displays a reduced GFP::PEZO-1 fluorescence in each tissue expressing *tir-1::mRuby*.
**Figure supplement 2—source data 1.** Quantification of the fluorescent intensity of GFP-PEZO-1::Degron at different conditions.
**Figure supplement 3.** Somatic-tissue specific degradation of PEZO-1 causes severe ovulation defects.

---

spermathecal cells (*Figure 8—figure supplement 2C–C''*). GFP fluorescence intensities in the germline and on oocytes in the germline-specific GFP::PEZO-1::Degron animals were 2–3 fold lower when the animals were exposed to auxin, but the intensities were not affected in the somatic tissues (*Figure 8—figure supplement 2E–E'', G–G'', H, I*). Therefore, auxin-inducible degradation of GFP::PEZO-1::Degron in the different tissues is consistent with the TIR-1::mRuby expression pattern.

To characterize further the defects associated with the degradation of PEZO-1 in these different tissues, L4 animals were exposed to either 0.25% ethanol as control or 2 mM auxin, and brood sizes were determined 0–60 hr post L4 (Day 1–3). Interestingly, the brood sizes were significantly reduced in each of the PEZO-1::Degron strains compared with control, regardless of the promoter used. However, the reduction in brood size was less severe than that observed in the *pezo-1^{ko}* mutants (*Figures 8E,F* and *2A*). To ensure efficient degradation, we exposed animals to auxin for one generation and analyzed the brood size of their $F_1$ progeny. This longer auxin exposure did not significantly enhance the reduction in brood size (data not shown).

Depletion of PEZO-1 in the somatic tissues, including spermathecal and sheath cells, led to a variety of ovulation defects (*Figure 8—figure supplement 3A–I*). Pinched oocytes were frequently observed during ovulation (N = 9/27, *Figure 8—figure supplement 3I*). A fraction of the pinched oocytes entered the spermatheca, whereas the rest were left in the oviduct (*Figure 8—figure supplement 3C,D,I*). Surprisingly, most of the pinched oocytes were successfully expelled into the uterus and underwent embryogenesis as smaller embryos (data not shown). In addition, the process of oocyte entry into the spermatheca was frequently delayed or blocked (*Figure 8—figure supplement 3E–I*), suggesting that the distal spermathecal valve remained closed. In experiments in which wildtype sperm were in vivo labeled as described earlier, and mated into control and somatic-specific PEZO-1::Degron hermaphrodites, nearly 90% of the labeled sperm reached the spermatheca (zone 3) and only a few labeled sperm were observed in the uterus (*Figure 8G,H,K*). Notably, the ooplasmic uterine masses that we observed in our *pezo-1^{ko}* mutants were rarely observed in the somatic-specific degron strain.

Consistent with our male mating experiments, only 69% of the MitoTracker-labelled wildtype sperm accumulated at the spermatheca (zone 3) in the germline-specific PEZO-1::Degron animals exposed to auxin (*Figure 8I–K*). The remaining sperm were observed throughout the whole uterus (zones 1 and 2) after one hour of mating (*Figure 8I,J*). Crushed oocytes were rarely observed in the uterus of the germline-specific PEZO-1::Degron animals, in which the sperm distribution assay was performed. Therefore, the degradation of PEZO-1 in the germ line did not cause the severe uterine ooplasmic masses as we have observed for our *pezo-1^{ko}* mutants but it did interfere with sperm navigation to the spermatheca, suggesting impaired attractant signaling. This is a more likely explanation as uterine ooplasmic masses are not a physical impediment that could account for the defects in sperm migration.

## Modeling human PIEZO genetic diseases in *C. elegans*

*PIEZO* patient-specific alleles, which are known to disrupt the normal physiological functioning of the cardiovascular, musculoskeletal, and blood systems in humans, were the motivation for examining the role of *pezo-1* in the tubular structures of *C. elegans*. Our studies with null alleles of *pezo-1*

provide strong evidence that *pezo-1* is essential for normal *C. elegans* reproduction. It is therefore reasonable to model human monogenic diseases that are associated with *PIEZO1 and PIEZO2* mutations using the *C. elegans* reproductive system as a read-out of function. Individuals diagnosed with Dehydrated Hereditary Stomatocytosis (DHSt) were found to have a missense mutation in a conserved arginine residue (R2488Q) of PIEZO1. The orthologous residue (R2718L/P) was also mutated in PIEZO2 in individuals with Distal Arthrogryposis type 5 (DA5) (*Andolfo et al., 2013*; *Coste et al., 2013*; *Li et al., 2018*; *McMillin et al., 2014*).

Previous studies have shown that these arginine changes are functioning as gain-of-function mutations in their respective PIEZO protein (*Albuisson et al., 2013*; *Coste et al., 2013*; *Li et al., 2018*; *McMillin et al., 2014*). Sequence alignment indicated that R2405 in *C. elegans* PEZO-1 is the arginine residue homologous to both R2488 in human PEIZO1 and R2718 in human PIEZO2 (*Figure 9A*). Using CRISPR/Cas9, we generated the patient-specific *PIEZO2* allele (p.R2718P) in *C. elegans*, named *pezo-1(R2405P)*. To compare this patient-specific allele with that of our null alleles, and to determine the phenotypic consequences of a patient-specific allele, homozygous animals carrying the *pezo-1(R2405P)* mutation were created. Such homozygotes displayed reproductive defects similar to the *pezo-1^{ko}* mutants, including reduced ovulation rates, ooplasmic uterine masses (*Figure 9B*), and reduced brood sizes (*Figure 9C*). In addition, the phenotypes of *pezo-1(R2405P)* homozygotes were mildly enhanced in combination with *itr-1* RNAi and suppressed with *lfe-2* RNAi, consistent with our findings with *pezo-1^{ko}* mutants (*Figure 9D*). Interestingly, similar to the rescue assay in *pezo-1 CΔ*, the reduced ovulation rate in *pezo-1(R2405P)* was also significantly rescued by *spe-9(hc52ts)* sperm, suggesting that this variant of *pezo-1* may similarly disrupt ovulation and sperm-to-sheath signaling, leading to self-sterility (*Figure 9E*). Overall, these observations support the idea that *C. elegans* is an appropriate model system for the study of *PIEZO* diseases. Future suppressor screens with this and other *pezo-1* patient-specific alleles should help to identify other genetic interactors.

## Discussion

The PIEZO proteins are responsible for sensing mechanical stimuli during physiological processes. Most studies of PIEZOs have focused on electrophysiological assays in cultured cells. To take advantage of an in vivo system to investigate the developmental roles of the PIEZO channel in mechanotransduction, we generated deletion alleles as well as a patient-specific allele in the sole *C. elegans pezo-1* gene. The *C. elegans* reproductive system is an tubular system that is attractive for studies of *PIEZO* function and for mimicking the *PIEZO* patient-specific alleles, which are known to disrupt the normal physiological functioning of the cardiovascular, musculoskeletal, and blood systems in humans (*Albuisson et al., 2013*; *Alper, 2017*; *Andolfo et al., 2013*; *Bae et al., 2013*). Although the PEZO-1 protein is broadly expressed throughout the animal, we focused on the reproductive system because of its striking phenotypes. Utilizing different *pezo-1* mutants and the tissue-specific degradation of PEZO-1, our data indicate that dysfunction of *pezo-1* led to a significantly reduced brood size. This reduced brood size phenotype worsens with age. In *C. elegans*, the reproductive process incorporates a series of sequential events, including proper ovulation, fertilization, expulsion of the fertilized zygote into the uterus, and sperm navigation back to the spermatheca after each fertilization event, all of which are regulated by multiple inter-tissue signaling pathways.

### PEZO-1 channel regulates ovulation and expulsion of the fertilized zygote possibly by maintaining cytosolic Ca$^{2+}$ homeostasis

Ovulation is driven by the rhythmic and coordinated contraction of the gonadal sheath cells and the opening of the spermathecal distal valve (*McCarter et al., 1999*). Similarly, expulsion of the fertilized zygote into the uterus is achieved by the contraction of the spermatheca and the opening of the spermathecal-uterine valve. Mutations in the *pezo-1* gene cause dramatic effects on this entire process. We observed sheath cell defects such that the mature oocyte was not properly pushed into the spermatheca. In addition, spermathecal valve defects either inhibited proper entry of the oocyte into the spermatheca, or proper exit. In many cases, the oocyte was crushed as it progressed through the spermatheca, resulting in accumulation of ooplasm in the uterus. Genetic interactions between *pezo-1* mutants and *itr-1* or *lfe-2* RNAi support the idea that *pezo-1* may play a role in maintaining Ca$^{2+}$ homeostasis during ovulation and zygote expulsion from the spermatheca. This is

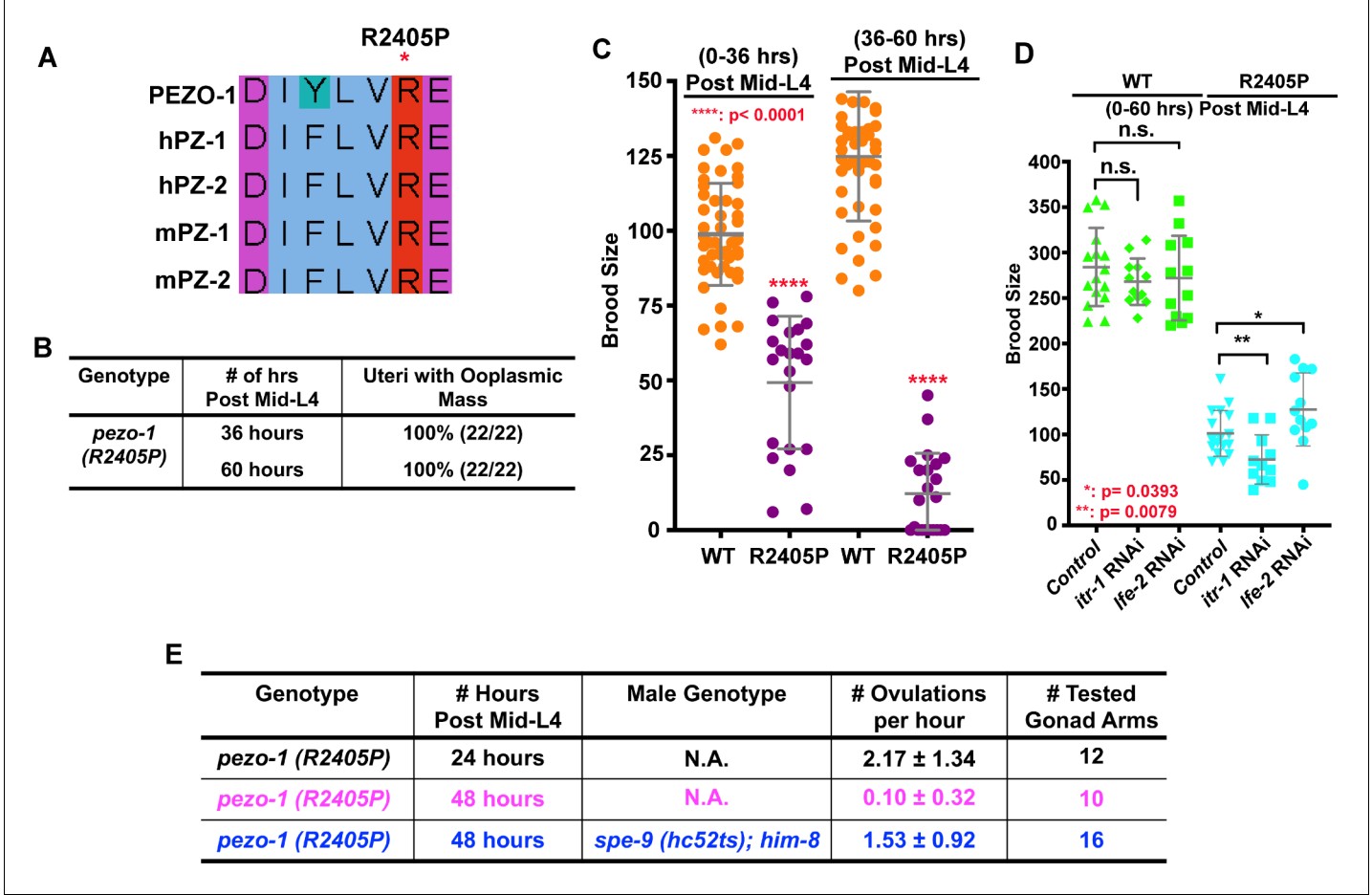

**Figure 9.** The *PIEZO1* disease allele causes severe brood size reduction in *C. elegans*. (A) Sequence alignment showing arginine 2405 (R2405) in *C. elegans* PEZO-1 is highly conserved with human and mouse PIEZO1 and PIEZO2. (B) A conserved patient-specific allele, *pezo-1(R2405P)*, was generated and causes uterine ooplasmic masses and (C) a severe reduction in brood size. (D) *itr-1(RNAi)* enhanced the brood size reduction of *pezo-1(R2405P)* mutants, while *lfe-2(RNAi)* slightly rescued the reduced brood size. (E) *spe-9(hc52ts)* sperm rescued the very low ovulation rate in *pezo-1(R2405P)* hermaphrodites. P-values: *, p=0.0393 (D); **, p=0.0079 (D); ****, p<0.0001 (C) (*t*-test).

The online version of this article includes the following source data for figure 9:

**Source data 1.** Quantification of brood size in mutants for the patient-specific allele *pezo-1(R2405P)* and the genetic interaction of *pezo-1(R2405P)* with the RNAi depletion of calcium regulators.

consistent with previous studies showing PIEZO1 responses to mechanical stimuli through $Ca^{2+}$ signaling (*He et al., 2018*; *Li et al., 2014*).

On the basis of the present studies, we hypothesize a few possible pathways for a $Ca^{2+}$-mediated response to mechanical stimuli to which PEZO-1 may contribute. One possibility is that PEZO-1 may detect when cytosolic $Ca^{2+}$ levels are extremely low and might replenish the cell with extracellular $Ca^{2+}$, in a manner similar to that involving the CRAC channel ORAI-1 (*Lorin-Nebel et al., 2007*). Consistent with this idea, our genetic data revealed an enhancement of the *pezo-1* phenotype upon CRAC channel *orai-1* RNAi, which is responsible for replenishing cytosolic $Ca^{2+}$ (*Figure 4C*). This suggests that PEZO-1 and ORAI-1 act in parallel pathways to replenish cytosolic $Ca^{2+}$.

Previous studies identified the ER $Ca^{2+}$ pump sarco/endoplasmic reticulum $Ca^{2+}$ ATPase (SERCA) as an interacting partner of PIEZO1, which suppresses PIEZO1 activation (*Zhang et al., 2017*). SERCA is essential for recycling $Ca^{2+}$ into SR/ER $Ca^{2+}$ stores, which is an important process for maintaining $Ca^{2+}$ homeostasis during tissue contractility (*Periasamy and Huke, 2001*; *Zwaal et al., 2001*). PIEZO1 has been reported to be involved in integrin activation to recruit the small GTPase R-Ras to the ER, which promotes $Ca^{2+}$ release from an intracellular store to the cytosol

(*McHugh et al., 2010*). These observations suggest that PEZO-1 may act as an ER $Ca^{2+}$ channel to regulate ER $Ca^{2+}$ homeostasis.

Last, normal spermathecal GCaMP fluorescence was observed during the first three ovulations in *pezo-1* mutants, suggesting that other $Ca^{2+}$ or mechanosensitive channels may perform redundant functions during $Ca^{2+}$ influx. One alternative model could be that PEZO-1 acts in parallel to these $Ca^{2+}$ regulators and yet does not have a direct role in calcium homeostasis itself. Future studies will be required to resolve the precise molecular effect of PEZO-1 on $Ca^{2+}$ and to understand how PEZO-1 regulates inter/intra cellular communication with/without $Ca^{2+}$ and potentially how other interacting partners coordinate during these processes.

## PEZO-1 channel is required for sperm navigation

*C. elegans* employs multiple peptide and lipophilic hormones to coordinate different tissues during reproduction. Ovulation is initiated by MSP (major sperm proteins) signaling derived from sperm to trigger oocyte maturation and sheath cell contraction (*Kuwabara, 2003*; *McCarter et al., 1999*; *Miller, 2001*). After each fertilization event, oocytes secrete F-series prostaglandins (F-PGs) into the extracellular environment of the reproductive tract and stimulate sperm attraction back to the spermatheca (*Hoang et al., 2013*). Our observations revealed a strong expression of PEZO-1 on the plasma membranes of both oocytes and sperm. Dysfunction of *pezo-1* causes a severe reduction of the ovulation rate and defective sperm navigation back to the spermatheca in aged animals. Male mating significantly rescued the very low ovulation rate in *pezo-1* mutants, as did the injection of purified fluorescently tagged MSP. Furthermore, the sperm navigation defects were observed in the germline-specific degradation of PEZO-1 animals, which showed fewer sperm successfully navigating back to the spermatheca. Collectively, depletion of PEZO-1 disrupted the ability of sperm to navigate back to the spermatheca, which may contribute to the reduced ovulation rate and defective sheath cell contraction.

## Working model

Our study supports the working model that PEZO-1 functions to promote the sheath cell contractions that push the oocyte into the spermatheca as the first step in ovulation (*Figure 10*, step 1). Simultaneously, PEZO-1 may play a role in sensing the sheath cell contractions and in triggering the spermathecal distal valve to open to allow oocyte entry into the spermatheca. During fertilization, the distal and spermathecal-uterine valves have to remain closed, which is probably influenced by PEZO-1 (*Figure 10*, step 2). After fertilization, PEZO-1 regulates the spermathecal tissues and controls the sp-ut valve to trigger a series of events to expel the fertilized oocyte into the uterus. Last, PEZO-1 appears to function in the attraction of the sperm back into the spermatheca after being pushed out by the exiting of the newly fertilized oocyte (*Figure 10*, step 3). Thus, dysfunction of PEZO-1 may contribute to multiple defects in all of these steps, including failure of oocyte entry into the spermatheca, the crushing of oocytes as they transit through the ovary and spermatheca, and defective sheath-to-sperm signaling that perturbs the ability of sperm to crawl back into the spermatheca after each ovulation (*Figure 10*). Future studies are underway to determine the PEZO-1 function in each tissue (sheath, spermatheca, oocyte, and sperm) more precisely using even more cell-specific promoters in the AID degradation system.

## Modeling PIEZO diseases in the *C. elegans* reproductive system

Clinical reports indicate that either gain-of-function or loss-of-function mutations in human *PIEZO1* and *PIEZO2* cause a variety of physiological disorders (*Alper, 2017*). Interestingly, both gain-of-function and loss-of-function missense mutations were identified in the same PIEZO disease, such as hydrops fetalis and lymphatic dysplasia. However, the molecular mechanism underlying both extremes of PIEZO channel dysfunction remains unclear (*Alper, 2017*). Complete knockout of *PIEZO1 and PIEZO2* in mammalian models results in embryonic lethality and fetal cardiac defects, suggesting an important role of PIEZO1/2 in embryonic and cardiac development (*Ranade et al., 2014*; *Zhang et al., 2019*). However, the lack of surviving homozygous *PIEZO1/2* mutants in mammalian models makes it challenging to investigate the PIEZO function during embryogenesis and development.

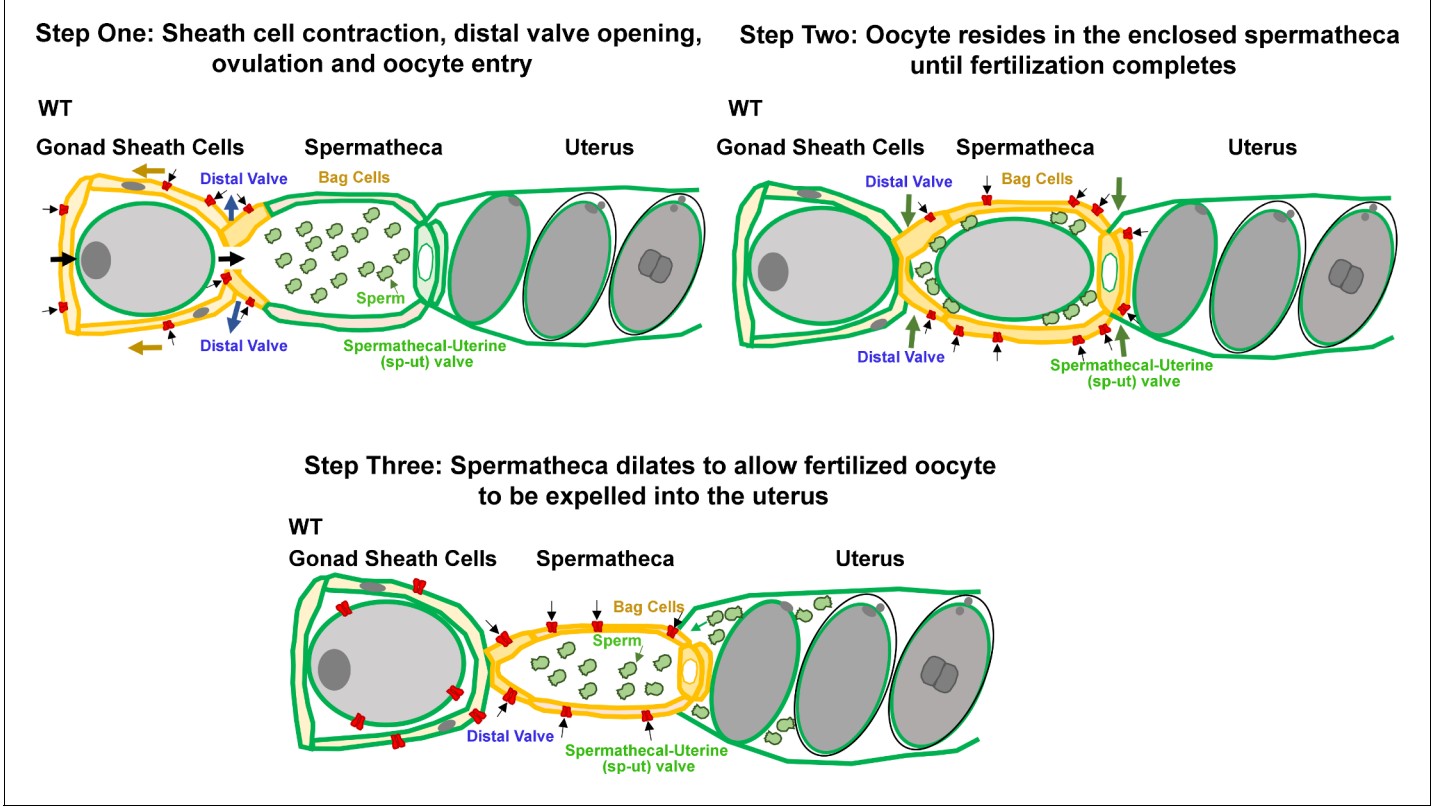

**Figure 10.** Working model for PEZO-1 during ovulation. Step One: PEZO-1 regulates somatic sheath cells and the spermathecal distal valve to push the oocyte into the spermatheca. Once a matured oocyte is ready for ovulation, PEZO-1 (red trapezoids) on the somatic sheath cells (yellow) triggers the contraction of the sheath to push the oocyte into the dilating spermatheca, through the distal valve. Meanwhile, the activated PEZO-1 (red trapezoids) on the distal valve (yellow) keeps the valve open and allows oocyte entry the spermatheca (green). Step Two: during fertilization, the PEZO-1 (red trapezoids) coordinates both distal (yellow) and spermathecal-uterine valves (yellow) to remain closed for 3–5 min. Step Three: After fertilization, PEZO-1 (red trapezoids) is activated on the spermathecal bag cells (yellow) and the sp-ut valve (yellow) to trigger a series of mechanical events (including spermathecal contractions and sp-ut valve opening) to expel the fertilized oocyte into the uterus (green). After oocyte entry into the uterus, we speculate that the PEZO-1 (red trapezoids) on the oocyte (far left) also functions to attract the sperm (green cells) back to the spermatheca. The precise mechanism of how PEZO-1 regulates sperm attraction remains unknown. Dysfunction of PEZO-1 causes the oocytes to be crushed as they are pushed into (Step 1) and expelled from the spermatheca (Step 3). The yellow represents the tissues that are under mechanical tension at each step during ovulation. PEZO-1 probably functions at the plasma membrane to sense the mechanical stimuli and to trigger intracellular signaling. The black arrows indicate the direction of extracellular cation influx when PEZO-1 channels are activated.

A DA5 patient-specific allele in the *C. elegans pezo-1* gene displayed reproductive phenotypes that were identical to those of our *pezo-1* deletion mutants, suggesting that this allele must be loss-of-function. The observation that our *pezo-1* deletion strains and a putative patient-specific gain-of-function mutation both lead to reproductive defects suggests that either hypomorphic or hypermorphic PEZO-1 channel activity is harmful. Therefore, our study demonstrates the usefulness of *C. elegans* as a model system to investigate PIEZO-derived human diseases.

The phenotypes described here in *C. elegans* do not exactly resemble those of the PIEZO-derived human diseases, but there are similarities at the cellular level that may be relevant to the human diseases. Stretch-sensitive channels from the Piezo family are important for vascular development and lymphatic valve formation. In zebrafish, Piezo channels sense fluid flow to regulate both endothelial and smooth muscle cell maturation and heart valve development (*Duchemin et al., 2019*). In mice, PIEZO1 is required for the formation of lymphatic valves, a key structure for proper lymphatic circulation in the body (*Nonomura et al., 2018*). However, both the mechanisms by which Piezo proteins operate and the proteins with which they interact remain unclear. In our study, we introduce a facile in vivo system for the study of PEZO-1 in the reproductive tract of *C. elegans*, a tubular tissue (spermatheca) with valves (spermatheca-uterine valve and distal valve) that must sense

the incoming and exiting oocyte during ovulation and fertilization. The formation and function of these structures are probably conserved between humans and *C. elegans.*

The dramatic reduction in brood size that we observed in all of our *pezo*-1 mutants will allow us to screen plausible chemical antagonists and agonists for PIEZO1 and PIEZO2 patient-specific alleles in vivo. In summary, we have demonstrated that the *C. elegans PIEZO1/2* ortholog *pezo-1* is required for efficient reproduction, and demonstrate the utility of *C. elegans* for the study of PIEZO functions. Future studies will determine whether other patient-specific alleles disrupt ovulation and sperm navigational signaling. Using promoters with more restricted expression patterns, the tissue-specific degradation system used in this report will also allow us to further dissect the cells or tissues that influence each of the phenotypes that we observed in this study. Future genetic and FDA-approved drugs screens will be used to identify putative suppressors in *pezo-1* mutants. These screens may provide insightful approaches for future clinical therapy.

# Materials and methods

### *C. elegans* strains used in this study

*C. elegans* strains were maintained with standard protocols. Strain information is listed in *Table 1*. AG493, AG494 and AG495 were created by crossing AG487 (*pezo-1::Degron*) males with hermaphrodites containing *ieSi65 [Psun-1::tir1::sun-1 3'UTR + Cbr-unc-119(+)] II*, *ieSi57 [Peft-3::tir1::mRuby:: unc-54 3'UTR + Cbr-unc-119(+)] II*, and *fxIs1[Ppie-1::tir1::mRuby] I*, respectively. We screened the $F_3$ adults for the presence of the *tir-1::mRuby* transgene by microscopy and genotyped for the *pezo-1:: Degron* by PCR. AG532 was created by crossing *pezo-1(av146 [gfp::pezo-1]) IV* males with the *unc-119(ed3); pwIs98 [YP170::tdimer2 + unc-119(+)] III* hermaphrodites containing YP170::tdimer2. $F_3$ adults with YP170::tdimer2 were genotyped by PCR screening for the *pezo-1$^{KO}$* allele.

### RNAi treatment

The RNAi-feeding constructs were obtained from the Ahringer and Vidal libraries (*Fraser et al., 2000*; *Rual et al., 2004*). RNAi bacteria were grown until log phase was reached and spread on MYOB plates containing 1 mM IPTG and 25 µg/ml carbenicillin and incubated overnight. To silence the target genes *itr-1* and *lfe-2*, mid-L4 hermaphrodites were picked onto plates with the IPTG-induced bacteria. Animals were grown on RNAi plates at 20°C for 36–60 hr. In order to improve the RNAi penetrance of *orai-1* and *sca-1*, L1 hermaphrodites were picked for RNAi-feeding assays. Alternatively, mid-L4 hermaphrodites were incubated on the *orai-1* or *sca-1* RNAi plates for one generation, and $F_1$ mid-L4 hermaphrodites were moved to fresh RNAi plates for brood size assays.

### Brood size determinations and embryonic viability assays

Single mid-L4 hermaphrodites were picked onto 35 mm MYOB plates seeded with 10 µl of OP50 bacteria and allowed to lay eggs for 36 hr (plate one contains the brood size from 0 to 36 hr post mid-L4). The same hermaphrodite was moved to a new 35 mm MYOB plate to lay eggs for another 24 hr and then were removed from the plate (this plate contains the brood size from 36 to 60 hr post mid-L4). Twenty-four hours after removing the mothers, only fertilized embryos and larvae were counted to determine brood size. Brood sizes were determined at 36 hr and 60 hr. Percentage of embryonic viability = (the number of hatched larva/the total brood size) *100%.

### BODIPY 493/503 staining

BODIPY 493/503 (Invitrogen # D3922) was dissolved in 100% DMSO to 1 mg/ml. BODIPY stock was diluted by M9 to 6.7 µg/ml BODIPY (final concentration of DMSO was 0.8%) as the working stock. Hermaphrodites were washed in M9 three times and incubated in 6.7 µg/ml BODIPY for 20 min and then washed again in M9 at least three times. All washes and incubations were performed in a con-cavity slide (ThermoFisher, # S99369). The stained hermaphrodites were anesthetized with 0.1% tricaine and 0.01% tetramisole in M9 buffer for 15–30 min. The anesthetized animals were then transferred to a 5% agarose pad for imaging. Image acquisition was captured using a Nikon 60 × 1.2 NA water objective with 1 µm z-step size.

**Table 1.** *C. elegans* strains list in the study.

| | Strain | Genotype |
|---|---|---|
| *Figure 1* | AG404 | *pezo-1(av142[mScarlet::pezo-1]) IV*, CRISPR/Cas9 edit |
| | AG408 | *pezo-1(av146 [gfp::pezo-1]) IV*, CRISPR/Cas9 edit |
| | AG483 | *pezo-1(av182 [pezo-1::mScarlet]) IV*, CRISPR/Cas9 edit |
| *Figure 2* | N2 | Bristol (wild-type) |
| | AG406 | *pezo-1(av144[N-Δ]) IV*, CRISPR/Cas9 edit, deletion of exon 1–13 and introns |
| | AG416 | *pezo-1(av149[C-Δ]) IV*, CRISPR/Cas9 edit, deletion of exon 27–33 and introns |
| | AG530 | *pezo-1(av149[C-Δ]) IV; ruIs32 [pie-1p::GFP::H2B + unc-119(+)] III* |
| | AZ212 | *ruIs32 [pie-1p::GFP::H2B + unc-119(+)] III* |
| *Figure 3* | N2 | Bristol (wild-type) |
| | AG416 | *pezo-1(av149) IV*, CRISPR/Cas9 edit, deletion of exon 27–33 and introns |
| | LP598 | *dlg-1(cp301[dlg-1::mNG-C1∃xFlag]) X*, CRISPR/Cas9 edit |
| | AG491 | *pezo-1(av149) IV; dlg-1(cp301[dlg-1::mNG-C1∃xFlag]) X* |
| *Figure 4* | N2 | Bristol (wild-type) |
| | AG416 | *pezo-1(av149) IV*, CRISPR/Cas9 edit, deletion of exon 27–33 and introns |
| *Figure 5* | UN1108 | *xbIs1101 [fln-1p::GCaMP3; pRF4(rol-6$^D$(su1006))] II* |
| | AG414 | *pezo-1(av144) IV; xbIs1101 [fln-1p::GCaMP3; pRF4(rol-6$^D$(su1006))] II* |
| | AG415 | *pezo-1(av149) IV; xbIs1101 [fln-1p::GCaMP3; pRF4(rol-6$^D$(su1006))] II* |
| | AG448 | *pezo-1(av142 [mScarlet::pezo-1]) IV; xbIs1101 [fln-1p::GCaMP3; pRF4(rol-6$^D$(su1006))] II* |
| *Figure 6* | N2 | Bristol (wild-type) |
| | AG406 | *pezo-1(av144) IV*, CRISPR/Cas9 edit, deletion of exon 1–13 and introns |
| | AG416 | *pezo-1(av149) IV*, CRISPR/Cas9 edit, deletion of exon 27–33 and introns |
| | AG531 | *spe-9(hc52ts) I; him-8(e1489) IV* |
| | BA17 | *fem-1(hc17ts) IV* |
| | CB1489 | *him-8(e1489) IV* |
| *Figure 7* | N2 | Bristol (wild-type) |
| | AG416 | *pezo-1(av149) IV*, CRISPR/Cas9 edit, deletion of exon 27–33 and introns |
| *Figure 8* | N2 | Bristol (wild-type) |
| | AG487 | *pezo-1(av190 [pezo-1::degron]) IV*, CRISPR/Cas9 edit |
| | AG493 | *pezo-1(av190 [pezo-1::degron]) IV; ieSi65 [sun-1p::TIR1::sun-1 3'UTR + Cbr-unc-119(+)] II; unc-119(ed3) III* |
| | AG494 | *pezo-1(av190 [pezo-1::degron]) IV; ieSi57 [eft-3p::TIR1::mRuby::unc-54 3'UTR + Cbr-unc-119(+)] II* |
| | AG495 | *pezo-1(av190[pezo-1::degron]) IV; fxIs1[pie-1p::TIR1::mRuby] I* |
| | AG564 | *fxIs1[pie-1p::TIR1::mRuby] I* |
| | AG565 | *ieSi65 [sun-1p::TIR1::sun-1 3'UTR + Cbr-unc-119(+)] II; unc-119(ed3) III.* |
| | AG566 | *ieSi57 [eft-3p::TIR1::mRuby::unc-54 3'UTR + Cbr-unc-119(+)] II* |
| *Figure 9* | N2 | Bristol (wild-type) |
| | AG437 | *pezo-1(av165[R2405P]) IV*, CRISPR/Cas9 edit. |
| | AG531 | *spe-9(hc52ts) I; him-8(e1489) IV* |
| *Figure 1—figure supplement 1* | AG404 | *pezo-1(av142 [mScarlet::pezo-1]) IV*, CRISPR/Cas9 edit |
| | AG408 | *pezo-1(av146 [gfp::pezo-1]) IV*, CRISPR/Cas9 edit |
| | AG483 | *pezo-1(av182 [pezo-1::mScarlet]) IV*, CRISPR/Cas9 edit |

*Table 1 continued on next page*

*Table 1 continued*

| | Strain | Genotype |
|---|---|---|
| *Figure 2—figure supplement 1* | N2 | Bristol (wild-type) |
| | AG406 | *pezo-1(av144) IV*, CRISPR/Cas9 edit, deletion of exon 1–13 and introns |
| | AG416 | *pezo-1(av149) IV*, CRISPR/Cas9 edit, deletion of exon 27–33 and introns |
| | PS8111 | *pezo-1(sy1199) IV*, CRISPR/Cas9 edit, Stop-cassette |
| | PS8546 | *pezo-1(sy1398) IV*, CRISPR/Cas9 edit, deletion of the first exon of isoforms i and j |
| | AG570 | *pezo-1(av240) IV*, CRISPR/Cas9 edit, deletion of full length of *pezo-1* |
| *Figure 5—figure supplement 1* | UN1108 | *xbIs1101 [fln-1p::GCaMP3; pRF4(rol-6$^D$(su1006))] II* |
| | AG414 | *pezo-1(av144) IV; xbIs1101 [fln-1p::GCaMP3; pRF4(rol-6$^D$(su1006))] II* |
| | AG415 | *pezo-1(av149) IV; xbIs1101 [fln-1p::GCaMP3; pRF4(rol-6$^D$(su1006))] II* |
| *Figure 6—figure supplement 1* | AG494 | *pezo-1(av190 [pezo-1::degron]) IV; ieSi57 [eft-3p::TIR1::mRuby::unc-54 3'UTR + Cbr-unc-119(+)] II* |
| | AG416 | *pezo-1(av149) IV*, CRISPR/Cas9 edit, deletion of exon 27–33 and introns |
| | BA17 | *fem-1(hc17ts) IV* |
| | AG571 | *pezo-1(av149) IV; fem-1(hc17ts) IV* |
| *Figure 8—figure supplement 1* | AG493 | *pezo-1(av190 [pezo-1::degron]) IV; ieSi65 [sun-1p::TIR1::sun-1 3'UTR + Cbr-unc-119(+)] II; unc-119(ed3) III* |
| | AG494 | *pezo-1(av190 [pezo-1::degron]) IV; ieSi57 [eft-3p::TIR1::mRuby::unc-54 3'UTR + Cbr-unc-119(+)] II* |
| | AG495 | *pezo-1(av190[pezo-1::degron]) IV; fxIs1[pie-1p::TIR1::mRuby] I* |
| *Figure 8—figure supplement 2* | AG582 | *pezo-1(av241 [gfp::pezo-1::degron]) IV*, CRISPR/Cas9 edit |
| | AG567 | *pezo-1(av241 [gfp::pezo-1::degron]) IV; ieSi57 [eft-3p::TIR1::mRuby::unc-54 3'UTR + Cbr-unc-119(+)] II* |
| | AG568 | *pezo-1(av241 [gfp::pezo-1::degron]) IV; fxIs1[pie-1p::TIR1::mRuby] I* |
| | AG569 | *pezo-1(av241 [gfp::pezo-1::degron]) IV; ieSi65 [sun-1p::TIR1::sun-1 3'UTR + Cbr-unc-119(+)] II; unc-119(ed3) III* |
| *Figure 8—figure supplement 3* | AG494 | *pezo-1(av190 [pezo-1::degron]) IV; ieSi57 [eft-3p::TIR1::mRuby::unc-54 3'UTR + Cbr-unc-119(+)] II* |
| *Video 1* | AG408 | *pezo-1(av146 [gfp::pezo-1]) IV*, CRISPR/Cas9 edit |
| *Video 2* | N2 | Bristol (wild-type) |
| | AG406 | *pezo-1(av149)] IV*, CRISPR/Cas9 edit, deletion of exon 27–33 and introns |
| *Video 3* | LP598 | *dlg-1(cp301[dlg-1::mNG-C13xFlag]) X*, CRISPR/Cas9 edit |
| | AG491 | *pezo-1(av149) IV; dlg-1(cp301[dlg-1::mNG-C13xFlag]) X* |
| *Video 4* | AG406 | *pezo-1(av149) IV*, CRISPR/Cas9 edit, deletion of exon 27–33 and introns |
| *Video 5* | AG448 | *pezo-1(av142 [mScarlet::pezo-1]) IV; xbIs1101 [fln-1p::GCaMP3; pRF4(rol-6$^D$(su1006))] II* |
| *Video 6* | UN1108 | *xbIs1101 [fln-1p::GCaMP3; pRF4(rol-6$^D$(su1006))] II* |
| | AG415 | *pezo-1(av149) IV; xbIs1101 [fln-1p::GCaMP3; pRF4(rol-6$^D$(su1006))] II* |

## Whole-animal DAPI staining

Animals were washed in M9 in a concavity slide, and then transferred to 1 µl of egg white/M9/azide on SuperFrost slides (Daigger # EF15978Z). Alternatively, animals were directly picked from plates into egg white/M9/azide, trying not to carry over too much bacteria. With an eyelash, buffer around animals was spread out to a very thin layer, until the animals were almost desiccated onto the slide. Slides were immersed in a Coplin jar containing Carnoy's fixative and fixed for a minimum of 1.5 hr or for as long as one week at room temperature or 4°C. Sequential ethanol (EtOH) rehydration was carried out in coplin jars containing about 50 ml of the following solutions for 2 min each: 90% EtOH in water, 70% EtOH in water, 50% EtOH in PBS, 25% EtOH in PBS, and PBS alone. Slides were then immersed in coplin jars containing DAPI stain (1 µg/ml) in PBS for 10 min. Slides were rinsed three times, 5 min each, in PBS. A drop of Vectashield mounting medium (#H-1500–10) was added, as was a coverslip, followed by nail polish to seal the coverslip. Image acquisition was captured by a Nikon 60 × 1.2 NA water objective with 1 µm z-step size.

## Yoda-1 dietary supplementation

Yoda1 (Tocris # 5586) was dissolved in DMSO to a stock concentration of 2.5 mM. This stock was added to 100 ml MYOB medium to a final concentration of 20 µM. Single mid-L4 hermaphrodites were picked onto 35 mm Yoda1-supplemented MYOB plates and control DMSO-only MYOB plates, each seeded with 10 µl of OP50 bacteria and allowed to lay eggs for 36 hr (plate one contains the brood from 0 to 36 hr post mid-L4). Each hermaphrodite was moved to a new 35 mm MYOB plate (with or without Yoda1) to lay eggs for another 24 hr and then was removed from the plate (this plate contains the brood from 36 to 60 hr post mid-L4). Twenty-four hours after removing the mothers, only fertilized embryos and larvae were counted to determine the brood size. Brood sizes were determined at 60 hr. Percentage of embryonic viability = (the number of hatched larva/the total number of hatched and unhatched animals) *100%.

## Live imaging to determine ovulation rates

For imaging ovulation, animals were immobilized on 4% agar pads with anesthetic (0.1% tricaine and 0.01% tetramisole in M9 buffer). DIC image acquisition was captured by a Nikon 60 × 1.2 NA water objective with 1–2 µm z-step size; 10–15 z planes were captured. Time interval for ovulation imaging was every 45–60 s, and duration of imaging was 60–90 min. Ovulation rate = (number of successfully ovulated oocytes)/total image duration.

## CRISPR design

We used the Bristol N2 strain as the wild type for CRISPR/Cas9 editing. The gene-specific 20-nucleotide sequences for crRNA synthesis were selected with the help of a guide RNA design checker from Integrated DNA Technologies (IDT) (https://www.idtdna.com) and were ordered as 20 nmol or 4 nmol products from Dharmacon (https://dharmacon.horizondiscovery.com), along with tracrRNA. Repair template design followed the standard protocols (*Paix et al., 2015*; *Vicencio et al., 2019*). Approximately 30 young gravid animals were injected with the prepared CRISPR/Cas9 injection mix, as described in the literature (*Paix et al., 2015*). *pezo-1 NΔ* and *pezo-1 CΔ* mutants were generated by CRISPR/Cas9 mixes that contained two guide RNAs at flanking regions of *pezo-1* coding regions. Heterozygous *pezo-1* deletion animals were first screened by PCR and then homozygosed in subsequent generations. mScarlet insertions at the *pezo-1* C-terminus were performed by Nested CRISPR (*Vicencio et al., 2019*). Homozygous *nest-1* edited animals were confirmed by PCR and restriction enzyme digestion and selected for the secondary CRISPR/Cas9 editing. Full-length mScarlet insertion animals were screened by PCR and visualized by fluorescence microscopy. All homozygous animals edited by CRISPR/Cas9 were confirmed by Sanger sequencing (Eurofins). The detailed sequence information for the repair template and guide RNAs are listed in *Table 2*.

The short isoform deletion, *pezo-1(sy1398)*, was generated using Cas9 expressed from a plasmid (*Friedland et al., 2013*) and four guides (GAGAACTTGAATTCAATGG, AAGCTTCTTCCGTCTCCGG, GCAGTATTTGACCAACTGG, ATAAAACAAGGCAACCAGG) along with a *dpy-10* guide and repair oligo. These reagents were injected into young adult N2 animals, and successful injections were identified by the presence of roller or dumpy progeny on the plate. Roller progeny were singled out and screened via PCR for the deletion mutation. The deletion was verified by Sanger sequencing using two external primers (CTCTCGCCTATCCACTTGAGCTTA and GGAAACAATTGAGCCGAGAATGGA) to amplify the region. This deletion should only disrupt the expression of isoforms i and j (*Figure 2—figure supplement 1B*). The CRISPR-Cas9 STOP-IN mutant, *pezo-1(sy1199)*, was generated using purified Cas9 protein at 10 µg/µl concentration, a purified guide RNA near the mutation location (CCAGAAGCTCGTAAGCCAGG), and a single-stranded DNA repair oligo containing three stop codons, one in every reading frame (underlined, cttatcgctgtttctgaaccagaagctcgtaagccGG-GAAGTTTGTCCAGAGCAGAGG<u>TGAC</u><u>TAAG</u><u>TGA</u>TAAgctagcaggaggcactgaagaaacggatggtgatgaag). These reagents were injected into N2 young adults along with a *dpy-10* guide and repair oligo. Successful injections were identified by the presence of dumpy and roller progeny. Thirty roller progeny were singled out from 'jackpot' plates (plates with a high incidence of dumpy and roller progeny) and screened via PCR (GACAGGACTTTCCCGCCAACTTAA and ATCATTCGCCGATTGCACAAGTTG) and the presence of a NheI restriction site that was included in the repair oligo.

**Table 2.** List of the sequence for the CRISPR design.

| Strain | Genotype | Description | Sequence name | Sequence 5′–3′ | PAM |
|---|---|---|---|---|---|
| AG406 | *pezo-1 (av144)* IV | Deletion of exons 1–13 and introns of *pezo-1* | crRNA N-terminus | **ACACAGCAACAACAGAATGA** | CGG |
| | | | crRNA C-terminus | TGGGGGTGTTGCAGTGGCTA | AGG |
| | | | Repair template | atctgaatcggtggtcgtaacacagc aacaacaga**g**tttgacacattttccg ttgagacttgaaaaatag | |
| | | | Genotyping F$_1$ | GCGGTAAATCTGAATCGGTGG | |
| | | | Genotyping R$_1$ | TTGGAAAAGCAGGCACAACC | |
| | | | Genotyping internal | CGATCCAGCGTGGATGAACT | |
| AG416 | *pezo-1 (av149)* IV | Deletion of exons 27–33 and introns of *pezo-1* | crRNA N-terminus | CGGTGGCAGCGTACATTATC | TGG |
| | | | crRNA C-terminus | CACCAGCGACACTCATCGAA | TGG |
| | | | Repair template | tccagtctcccatatttatttttttctgttccag <u>TA</u>G<u>A</u><u>TAA</u>G<u>TAA</u>GAGCAAAAAGAAGCAAGAATAA | |
| | | | Genotyping F$_1$ | AATCTGACTTGTGCCCTCCG | |
| | | | Genotyping R$_1$ | AATCAGGCGAGCAGTGAGAG | |
| | | | Genotyping internal | TCCACAGTCAATTCCTGCGT | |
| AG404 | *pezo-1(av142 [mScarlet::pezo-1])* IV | Knock in mScarlet at N-terminus of *pezo-1*, *mScarlet* was amplified from plasmid pMS050 | crRNA | ACACAGCAACAACAGAATGA | CGG |
| | | | Repair template F$_1$ | tgaatcggtggtcgtaacacagcaacaacaga ATG CTTGTAGAGCTCGTCCATTCC (mScarlet) | |
| | | | Repair template R$_1$ | AATTTGACGACGCACGA TTTTAAAAGCGGCGGGAC**T**GT CTTGTAGAGCTCGTCCATTCC (mScarlet) | |
| AG408 | *pezo-1(av146 [gfp::pezo-1])* IV | Knock in GFP at N-terminus of *pezo-1*, GFP was amplified from plasmid pDD282 | crRNA | ACACAGCAACAACAGAATGA | CGG |
| | | | Repair template F$_1$ | tgaatcggtggtcgtaacacagcaacaacagaATG agtaaaggagaagaattgttc (GFP) | |
| | | | Repair template R$_1$ | AATTTGACGACGCACGA TTTTAAAAGCGGCGGGAC**T**GT CTTGTAGAGCTCGTCCATTC (GFP) | |
| AG483 | *pezo-1(av182 [pezo-1::mScarlet])* IV. | Knock in mScarlet at C-terminus of *pezo-1*, mScarlet was amplified from plasmid pMS050 | NEST1 crRNA | CACCAGCGACACTCATCGAA | TGG |
| | | | Repair template | AATATTCCTGTTCCGATCACCAGCGACACT CATCGAA**TGG**AC**T**CG**T**ATGAG T**AA**GAA**A**AA**A**CA**G**GA**G** GTCTCCAAGGGAGAGGCCGTCATCAAGGAGTTCA TGC GTTTCAAGGTCCAAGCG**C**TCCGAGGGACGTCAC T**CCA** CCGGAGGAATGGACGAGCTCTACAAGTAAatttaaa- ta tttcactgtcaaatattctgcga (mScarlet) | |
| | | | Genotyping F$_1$ | TGGTTCGAGAAGCGAAGGAC | |
| | | | Genotyping R$_1$ | aatcaggcgagcagtgagag | |
| | | | NEST2 crRNA | TTCAAGGTCCAAGCGCTCCG | AGG |
| | | | Repair template F$_1$ | GCCGTCATCAAGGAGTTCATGCGT <u>TTCAAGGTCCACATGGAGGGATCCATGAACG</u> | |
| | | | Repair template R$_1$ | TAGAGCTCGTCCATTCCTCCGGTGGAGT GACGTCC**T**TC**T**GA**A**CGCTCGTATTGC TCGACGACGGTG | |

*Table 2 continued on next page*

*Table 2 continued*

| Strain | Genotype | Description | Sequence name | Sequence 5′–3′ | PAM |
|---|---|---|---|---|---|
| AG487 | *pezo-1(av190 [pezo-1::degron])* IV | Knock in Degron sequence at C-terminus of *pezo-1*, Degron was amplified from plasmid pK0132 | crRNA | CACCAGCGACACTCATCGAA | TGG |
| | | | Repair template F1 | AATATTCCTGTTCCGATCACCAGCGACACTCATCGAA**T** **GG**ACTCG**T**ATGAGT**AA**G**AAAAAA**CA**G**GA**G** ggagcatc gggagcctcaggagcatcg (linker) GACTACAAAGACCATGACGGTG (Degron) | |
| | | | Repair template R₁ | tcgcagaatatttgacagtgaaatatttaaat TTACTTCACGAACGCCGCC (Degron) | |
| AG437 | *pezo-1(av165[R2405P])* IV | Generate a point mutation R2405P in *pezo-1* | crRNA | CTATTTGGTTCGAGAAGCGA | AGG |
| | | | Repair template | CATCTTCTCAAAATTTGTCTCGACATCTATTTGG TACCAGAAGCGAAAGACTTCATGTTGGAGCAG gtaattatttagtttta | |
| AG570 | *pezo-1(av240)* IV | Deletion of full length of *pezo-1* | crRNA1 | ACACAGCAACAACAGAATGA | CGG |
| | | | crRNA2 | CACCAGCGACACTCATCGAA | TGG |
| | | | Repair template | ctgaatcggtggtcgtaacacagcaacaacagaATG T**A**GATAAGTAAGAGCAAAAAGAAGCAAGAATAA atttaaatatttc | |
| AG571 | *pezo-1(av242)* IV | Deletion of exons 27–33 and introns of *pezo-1* in *fem-1(hc17)* | crRNA1 | CGGTGGCAGCGTACATTATC | TGG |
| | | | crRNA2 | CACCAGCGACACTCATCGAA | TGG |
| | | | Repair template | tccagtctcccatatttattttttttctgttccag T**A**GA**T**A**A**G**TAA**GAGCAAAAAGAAGCAAGAATAA | |
| AG582 | *pezo-1(av241)* IV | Knock in Degron sequence at C-terminus of *pezo-1* in AG404, Degron was amplified from plasmid pK0132 | crRNA | CACCAGCGACACTCATCGAA | TGG |
| | | | Repair template F₁ | AATATTCCTGTTCCGATCACCAGCGACACTCATCG AAT**GG**ACTCG**T**ATGAGTAAG**AAAAAA**CA**G**GA**G** gga gcatcgggagcctcaggagcatcg (linker)GACTACAA AGACCATGACGGTG (Degron) | |
| | | | Repair template R₁ | tcgcagaatatttgacagtgaaatatttaaat TTACTTCACGAACGCCGCC (Degron) | |
| PS8111 | *pezo-1(sy1199)* IV | Knock in a stop cassette at C-terminus of *pezo-1* | crRNA | CCAGAAGCTCGTAAGCCAGG | AGG |
| | | | Repair template | cttatcgctgtttctgaaccagaagctcgtaagccGG GAAGTTTGTCCAGAGCAGAGGTGACTAAGT GATAAgctagcaggaggcactgaagaaacggatggtgatgaag | |
| | | | Genotyping F1 | GACAGGACTTTCCCGCCAACTTAA | |
| | | | Genotyping R₁ | ATCATTCGCCGATTGCACAAGTTG | |
| PS8546 | *pezo-1(sy1398)* IV | Deletion of the first exon of *pezo-1* isoforms i and j | crRNA1 | gagaacttgaattcaatgg | AGG |
| | | | crRNA2 | aagcttcttccgtctccgg | CGG |
| | | | crRNA3 | gcagtatttgaccaactgg | TGG |
| | | | crRNA4 | ataaaacaaggcaaccagg | GGG |
| | | | Genotyping F₁ | CTCTCGCCTATCCACTTGAGCTTA | |
| | | | Genotyping R₁ | GGAAACAATTGAGCCGAGAATGGA | |

Note: Capital letters represent the ORF or exon sequence, small letters indicate the intron sequence. Bolded letters indicate the optimized bases needed for the CRISPR design.

## Male mating assay with Day 3 hermaphrodites

25–30 mid-L4 wildtype or *pezo-1* mutant hermaphrodites were isolated to a fresh growth plate for 60 hr (such animals should be Day 3 adults at this time). To ensure mating success, ~30 adult males and 10–15 Day 3 hermaphrodites were transferred onto a 35 mm MYOB plate seeded with 10–20 μl of OP50 bacteria and allowed to mate for 12 hr. The other 10–15 Day 3 hermaphrodites were singled and transferred to 35 mm MYOB plates seeded with 10 μl of OP50 bacteria as the controls.

After the group mating, single mated hermaphrodites (72 hr post mid-L4) and 3–5 adult males were then transferred to a fresh 35 mm growth plate where mating could continue for another 24 hr. After 24 hr, the hermaphrodites (96 hr post mid-L4) and males were removed. The brood size (those embryos laid between 72–96 hr post mid-L4) and embryonic viability were determined 24 hr later after removal of all adults. Meanwhile, the broods from 60 to 96 hr post-mid L4 were also determined for the other 10–15 unmated Day 3 hermaphrodites that were kept on single plates as controls.

## Mating assay with the *fem-1* mutant

10–15 mid-L4 BA17 *fem-1(hc17ts)* hermaphrodites raised from embryos at the non-permissive temperature of 25°C were picked to mate with ~30 adult males for 12 hr at 25°C. Single mated hermaphrodites and 3–5 males were then transferred to a fresh 35 mm growth plate and allowed to mate for another 24 hr at 25°C before all adults were removed from the plates. As control, 10–15 unmated BA17 hermaphrodites grown at 25°C were kept on single plates. The brood sizes and embryonic viability were determined 24 hr later. Alternatively, 10–15 L1 BA17 *fem-1(hc17ts)* hermaphrodites were isolated on a fresh growth plate and incubated at 25°C for 48 hr (young adult hermaphrodites). Approximately 30 adult males and 10–15 BA17 young hermaphrodites were then transferred onto a 35 mm MYOB plate seeded with 10–20 ul of OP50 bacteria and allowed to mate for 12 hr at 25°C. Single mated hermaphrodites and 3–5 males were then transferred to a fresh 35 mm growth plate. After laying embryos for 24 hr, the hermaphrodites and males were removed. Meanwhile, the other same-age 10–15 unmated Day 3 hermaphrodites were kept on single plates as the control. The brood size and embryonic viability were counted 24 hr later after removal of all adults. All of the animals were incubated at 25°C during mating and propagation to ensure the penetration of the *fem-1* *(hc17ts)* phenotype.

## Mating assay with the *spe-9* mutant

10–15 hermaphrodites were picked to mate with ~30 AG521 [*spe-9(hc52ts)*] adult males for 12 hr at 25°C. Mated hermaphrodites were immobilized on 4% agar pads with anesthetic (0.1% tricaine and 0.01% tetramisole in M9 buffer) for ovulation rate assays. The acquisition of DIC images was performed by confocal imaging system (see below) with a Nikon 60 × 1.2 N with 1–2 μm z-step size and 10–15 z planes. Time interval for ovulation imaging is every 45–60 s, and the duration of imaging is 60–90 min. Ovulation rate = (number of successfully ovulated oocytes)/total image duration.

## Sperm distribution assay and mating assay

MitoTracker Red CMXRos (MT) (Invitrogen # M7512) was used to label male sperm following the protocol adapted from previous studies (*Hoang et al., 2013*; *Kubagawa et al., 2006*). MT was dissolved in 100% DMSO to 1 mM. About 100 males were transferred to a concavity slide (ThermoFisher, # S99369) with 150 μl 10 μM MT solution (diluted in M9 buffer). Males were incubated in the MT buffer for 2 hr and then transferred to fresh growth plates to recover overnight. The plates were covered by foil to prevent light exposure. About 30 males were placed with 10 anesthetized hermaphrodites (0.1% tricaine and 0.01% tetramisole in M9 buffer) on MYOB plates seeded with a 50–100 μl OP50 bacteria. After 30 min of mating, hermaphrodites were then isolated and allowed to rest on food for one hour. The mated hermaphrodites were then mounted for microscopy on 5% agarose pads with the anesthetic. Image acquisition was captured by a Nikon 60 × 1.2 NA water objective with 1 um z-step size. Quantification of sperm distribution in the uterus starts at the vulva and extends up to and includes the spermatheca. The sperm counted were throughout the gonad at a focal depth of about 30 μm. The whole uterus was divided into three zones. Zone 1 contains the vulva region, and Zone 3 contains the spermatheca. The number of sperm was manually counted within each zone. The distribution percentage = (the number in each zone) / (the total labeled sperm observed) * 100%. The quantified data contains at least 30 total stained sperm in the entire uterus. At least 3–7 mated hermaphrodites were counted in each mating assay, and experiments were repeated at least 3 times.

## Auxin-inducible treatment in the degron strains

Animals were grown on bacteria-seeded MYOB plates containing auxin. The natural auxin indole-3-acetic acid (IAA) was purchased from Alfa Aesar (#A10556). IAA was dissolved in ethanol as a 400 mM stock solution. Auxin was added to autoclaved MYOB agar when it cooled to about 50–60°C before pouring. MYOB plates containing the final concentration of auxin (1 or 2 mM) were used to test the degron-edited worms.

To degrade the target protein efficiently, L1 or L2 hermaphrodites were picked onto auxin plates. Animals were grown on the plates at 20°C for 36–60 hr for the brood size assay. Alternatively, mid-L4 hermaphrodites were incubated on the auxin plate for one generation, and $F_1$ mid-L4 hermaphrodites were picked to a fresh auxin plate for the brood size assay or for phenotypic imaging.

## The microinjection of fluorescein-labeled MSP into aged *pezo-1 C*Δ

The microinjection of 101.6 μM NHS-Fluorescein-labeled MSP-142 into both aged (day 2, 48 hr post mid-L4) wildtype and *pezo-1 C*Δ hermaphrodites was performed as previously described (*Miller, 2001*). The injected worms recovered for 4 hr on MYOB plates with OP50 food before imaging. The acquisition of GFP and DIC images was performed by our confocal imaging system (see below) with 1–2 μm z-step size and 10–15 z planes. Time interval for ovulation imaging was every 45–60 s, and duration of imaging was 60–90 min. Ovulation rate = number of successfully ovulated oocytes / total duration of imaging.

## Microscopy

Live imaging was performed on a spinning disk confocal system that uses a Nikon $60 \times 1.2$ NA water objective, a Photometrics Prime 95B EMCCD camera, and a Yokogawa CSU-X1 confocal scanner unit. Images were acquired and analyzed by Nikon's NIS imaging software and ImageJ/FIJI Bio-formats plugin (National Institutes of Health) (*Linkert et al., 2010*; *Schindelin et al., 2012*). GCaMP3 images were also acquired by a $60 \times /1.40$ NA oil-immersion objective on a Nikon Eclipse 80i microscope equipped with a SPOT RT39M5 sCMOS camera (Diagnostic Instruments, Sterling Heights, MI, USA) with a 0.63x wide field adapter, controlled by SPOT Advanced imaging software (v. 5.0) with Peripheral Devices and Quantitative Imaging modules. Images were acquired at $2448 \times 2048$ pixels, using the full camera chip, and saved as 8-bit TIFF files. Fluorescence excitation was provided by a Nikon Intensilight C-HGFI 130 W mercury lamp and shuttered with a Lambda 10-B SmartShutter (Sutter Instruments, Novato, CA), also controlled through the SPOT software. Single-channel GCaMP time-lapse movies were acquired using a GFP filter set (470/40 × 495 lpxr 525/50 m) (Chroma Technologies, Bellows Falls, VT) at 1 frame per second, with an exposure time of 40–60 ms, gain of 8, and neutral density of 16.

## GCaMP3 imaging acquisition and data processing

For all GCaMP3 imaging data, animals were immobilized on 7.5% agarose pads with 0.05 μm polystyrene beads and imaged using confocal microscopy as described above. Images were acquired every 1 s and saved as 16-bit TIFF files. DIC images were acquired every 3 s. Only successful embryo transits (embryos that were expelled through the sp-ut valve) were analyzed for this GCaMP3 study. The GCaMP3 metrics, including rising time and fraction over half max data, as well as the GCaMP3 intensity heat map were processed by the custom Fiji and Matlab coded platform (*Bouffard et al., 2019*). GCaMP3 kymograms were generated by custom Fiji code using the commands Image > Stacks > Reslice followed by Image > Stacks > Z Project (Average Intensity) (*Bouffard et al., 2019*). Only the very first three ovulations were imaged for each animal. Detailed processing and analysis of the GCaMP time series was performed exactly as described in *Bouffard et al., 2019*.

## Statistics

Statistical significance was determined by p-value from an unpaired two-tailed t-test. P-values: ns, not significant; *, <0.05; **, <0.01; ***, <0.001; ****, <0.0001. Both the Shapiro-Wilk and the Kolmogorov-Smirnov Normality test indicated that all data follow normal distributions.

## Acknowledgements

We thank the *Caenorhabditis* Genetics Center, which is funded by the National Institutes of Health Office of Research Infrastructure Programs (P40OD010440), for providing strains for this study. We thank Dr Orna Cohen-Fix for generously sharing the SP-12::GFP strain, and Dr Harold Smith for sharing the BA17 *fem-1(hc17ts)* strain. We also thank Dr David Greenstein for sharing fluorescein-tagged MSP and discussion about mating assays. We are grateful to the members of the Golden laboratory, Dr Peter Kropp, Dr Tao Cai, Rosie Bauer, Isabella Zafra, and Carina Graham for productive discussions and preparing reagents. We thank our summer intern Kyle Wilson for manuscript editing. We especially thank Dr Harold Smith, Dr Orna Cohen-Fix, Dr Kevin O'Connell, Dr Katherine McJunkin and Dan Konzman for critical inputs on the project and feedback on the manuscript. We thank all members of the Baltimore Worm Club for providing feedback and suggestions to our investigations.

## Additional information

### Funding

| Funder | Grant reference number | Author |
| --- | --- | --- |
| National Institute of General Medical Sciences | GM110268 | Erin J Cram |
| National Institute of Neurological Disorders and Stroke | R01 NS113119 | Paul W Sternberg |
| NIH Clinical Center | R24 0D023041 | Paul W Sternberg |

The funders had no role in study design, data collection and interpretation, or the decision to submit the work for publication.

### Author contributions

Xiaofei Bai, Conceptualization, Data curation, Formal analysis, Investigation, Methodology, Writing - original draft; Jeff Bouffard, Data curation, Formal analysis, Investigation, Methodology; Avery Lord, Data curation, Investigation; Katherine Brugman, Formal analysis, Methodology; Paul W Sternberg, Conceptualization, Supervision, Funding acquisition; Erin J Cram, Conceptualization, Resources, Supervision, Funding acquisition, Investigation, Methodology, Writing - review and editing; Andy Golden, Conceptualization, Supervision, Methodology, Writing - review and editing

### Author ORCIDs

Xiaofei Bai https://orcid.org/0000-0001-8179-8162
Paul W Sternberg http://orcid.org/0000-0002-7699-0173
Andy Golden https://orcid.org/0000-0002-8599-2031

### Decision letter and Author response

Decision letter https://doi.org/10.7554/eLife.53603.sa1
Author response https://doi.org/10.7554/eLife.53603.sa2

## Additional files

### Supplementary files

• Transparent reporting form

### Data availability

All data generated or analysed during this study are included in the manuscript and supporting files. Source data files have been provided for all Figures and figure supplements.

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
