## [Decision Letter]

**Acceptance summary:**

This work using *C. elegans* represents one of the first descriptions of PIEZO proteins in the regulation of germ cell production and function. This careful analysis shows how the *C. elegans* PIEZO-1 protein may modulate this complex process at multiple points and serves as a basis for other studies investigating PIEZO function in mechanotransduction and how model organisms can be used to reveal fundamental information on proteins that are tied to specific diseases.

**Decision letter after peer review:**

Thank you for submitting your article "*Caenorhabditis elegans* PIEZO channel coordinates multiple reproductive tissues to govern ovulation" for consideration by *eLife*. Your article has been reviewed by three peer reviewers, including Diana S Chu as the Reviewing Editor and Reviewer #1, and the evaluation has been overseen by Didier Stainier as the Senior Editor.

Essential revisions:

1) Because the overall set of data do not directly support that PIEZO in *C. elegans* is a calcium regulator, the authors should revise the manuscript, particularly many of the conclusions reached, to less strongly advocate that their data supports a role for this *C. elegans* homolog in calcium signaling. The reviewers agree this, in fact, does not detract from the interest of the paper but instead reveals important information about the potential function of these proteins in different developmental contexts.

2) The experiments and explanations of roles in sperm signaling (in particular comments from reviewer 3 on spe-9 experiments) and experiments and conclusions about sperm navigation were either incomplete or confusing. It is not clear why the defects may not arise as secondary effects from the oocyte production defects. The authors should address all reviewer comments in order to re-evaluate the conclusions that can be reached regarding these experiments in order to make claims about such roles.

3) There should be more controls (or better explanation that address the need for controls) as requested by reviewers for some of the experiments presented, particularly the AID, spe-9, and Yoda1 experiments.

4) Evidence for inter-tissue signaling is not well supported and thus, unless the authors have additional, stronger data, should be removed from the manuscript.

Overall, we would like authors to 1) revise the manuscript to be careful about claims that PEZO-1 functions as a calcium regulator and to consider other roles or functions that may be revealed by their data , 2) revise the presentation and provide controls (or potentially add new data if available to strengthen their points) about claims for PEZO-1 function in sperm signaling or navigation. 3) provide controls for other experiments requested in individual reviewer comments. Reviewers felt that these issues should not warrant significant new experimental analysis.

Reviewer #1:

The manuscript describes the characterization of the function of PEZO-1 in *C. elegans*. PEZO-1 is the homolog of PIEZO1 and PIEZO2 in humans, which are mechano-sensitive channel proteins. The authors generate GFP-tagged versions of PEZO-1 to show that it is expressed broadly in many cell types, including reproductive tissues. They examine knock-out mutants of pezo-1 and find severe defects in fertility. They find that loss of pezo-1 causes oocytes to be crushed during the processes of ovulation, fertilization, and expulsion to the uterus. They further show that defective sperm navigation may also contribute to fertility defects. The authors have evidence that PEZO-1 works through calcium signaling – mutation in pezo-1 shows synergistic defects when combined with reduction of calcium signaling regulators, but calcium signaling itself was not different in their assays. Both tissue-specific knock-down of *C. elegans* pezo-1 and introduction of PIEZO patient-specific alleles in *C. elegans* also lead to fertility defects.

The work is interesting and novel. This is the first characterization of PIEZO function in *C. elegans*, where it is possible to use a variety of approaches to assess PIEZO function in a model organism. The high-quality data obtained through a combination of genetic knock-downs and AID degradation, in vivo imaging of tagged proteins, fertility assays, and calcium imaging is a strength of the work. Though PIEZO proteins are known to be important for mechanosensory transduction in several contexts in other systems, including humans and *Drosophila*, this is also the first demonstration that PEZO-1 is important for fertility, in particular, the production of germ cells that requires an orchestration of movements. In particular, the 'crushed oocyte' phenotype is quite striking. Their demonstration that PIEZO patient-specific alleles also result in fertility defects links the *C. elegans* work to understanding human disease and show that there is potential to use this system in the future to further dissect PIEZO function in developmental contexts in a tractable system. Because of these numerous strengths, I would recommend publication in *eLife*.

There are a few revisions that are required. One modest concern which is that the role of PEZO-1 in mechanical signal transduction is still not described in a concrete way. Of course, this is partially due to the obviously complicated nature of the fertility process that is affected. There are multiple points at which the mechano-sensation and transduction may be affected that result in the major phenotype of oocyte crushing. This could be improved by revising Figure 10. Instead of highlighting 'PEZO-1 Dysfunction' it would be clearer for the authors to highlight where they think PEZO-1 IS functioning. This could be done by either by adding a clear point at each step that describes that function or retitling each step to better highlight PEZO-1. The model as is now (in the text and figure) is more a reiteration of results and not so much a model for how and where PEZO-1 may be 'sensing' or 'responding' to mechanical stress. It is actually interesting that the phenotype is complicated because PIEZO proteins in other organisms may also have likewise complex roles.

Reviewer #2:

This paper explores for the first time the role of the piezo-1/2 ortholog in *C. elegans*, pezo-1, and it identifies a critical function in retaining sperm in the spermatheca. In the absence of pezo-1 sperm are quickly depleted from the spermatheca and as a consequence embryo production is halted and oocyte crushing begins on both ends of the spermatheca. The quality of the work is high and its results are novel and of broad interest. However, the paper suffers from several problems that must be rectified before publication. Likely because piezo-1/2 are known to be mechanosensing calcium channels, the authors probably started this project with the hypothesis that pezo-1 functions as a mechanosensitive calcium channel. However, nature being as it is, in the end they don't have any evidence for such a function. That in itself is not a problem. What is problematic in my view is that in writing the paper they nevertheless continue to push a narrative of pezo-1 regulating calcium in the reproductive system, despite there not being any evidence for this. The second problem I have as a reviewer with the current paper is that the interpretation of many of the experiments is taken too far and it is done in the Results section, which could lead a reader to confuse speculations (or wishful thinking) for actual results. The defect in pezo-1 mutants that can explain all of the observed phenotypes is the rapid depletion of sperm from the spermatheca because they fail to return after being washed out with embryos exiting into the uterus. The mating experiments and somatic or germline specific AID experiments were supposed to clarify whether the sperm navigation defect is due to lack of pezo-1 activity in the sperm or in the somatic gonad. Unfortunately, the data presented is contradictory (or perhaps only not explained well) because after reading and rereading the paper I still can't figure out if pezo-1 is required in the sperm or in the gonad or both. Once the authors clarify this point, remove experiments that don't lead to a conclusion (inx-14/22, YP170::tdimer2), remove overinterpretation statements from the Results section, and add missing controls for some experiments (Yoda1, AID) as detailed in my comments below, I believe the paper will be worthy of publication in e*Life*.

1) In multiple places the authors jump to conclusions based on associative evidence such as localization or genetic interaction. For example: "Notably, PEZO-1 is strongly expressed in several tubular tissues, including the pharyngeal-intestinal and spermathecal-uterine valves, which is consistent with our hypothesis that pezo-1 may be responsible for mechanoperception in these tissues." And: "Consistent with the hypothesis that reproductive tissues are regulated by mechanosensitive stimuli in *C. elegans*, expression of PEZO-1 likely functions to sense physical strain or contractility during ovulation and fertilization." – such sentences push a narrative of pezo-1 function in mechanoperception without any support in the data. Speculations and interpretations belong in the discussion. The result section should present the results without biasing the reader to a single interpretation.

2) "Partial co-localization of PEZO-1 with the ER marker SP12::GFP suggested that PEZO-1 may be processed in the ER and transported to the plasma membrane (Figure 1E)." – based on the image in Figure 1E the localization of SP12 and PEZO-1 is complementary and primarily non-overlapping. If the authors have reason to believe there is substantial co-localization they need to show and quantify it. In any case, it isn't clear why this is an important point to focus on. If the authors think the location of function of PEZO-1 is the plasma membrane then why not just focus on that localization?

3) "The genome-edited animals behaved normally, suggesting no functional disruption of tagging PEZO-1 with these fluorescent reporter genes." – This statement is questionable in light of what is visible in Figure 1F, where there is an accumulation of small oocyte pieces or vesicles or something else that looks abnormal between the -1 oocyte and spermatheca.

4) "The fluorescent signal of GFP::PEZO-1 is observed in both spermathecal valves until the valves open, suggesting that PEZO-1 may function to sense the mechanical stimuli at the valves during ovulation" – since the fluorescent protein is at the membrane it is expected to give a higher signal when the tissue is contracted compared to when it is dilated. The change in fluorescence intensity observed in the GFP::PEZO-1 movie on its own is not an indication of any mechanosensing. If the authors wish to claim enrichment in the valve at a certain time point they would need to do a ratiometric comparison with a membrane marker such as PH::mCherry.

5) "To mimic a gain-of-function phenotype in pezo-1, we fed wildtype animals with Yoda1, a PIEZO1 specific chemical agonist, which keeps the channel open" – No control was performed to show that Yoda1 doesn't have side effects in worms. There are plenty of chemicals that are supposedly specific but actually are not. A simple experiment that could at least show Yoda1's phenotype is PEZO-1-dependent is to perform the treatment on pezo-1KO, which we expect to be refractory to the drug and display the same brood size as pezo-1KO on its own.

6) "The defective ovulation is likely due to incomplete constriction of the sheath cells and improper gating of the distal spermathecal valve." – Incomplete constriction of the sheath is sufficient to explain the entry phenotype and it is not necessary to invoke improper gating of the distal spermathecal valve, which was not specifically tested. Moreover, improper sheath contraction is consistent with the absence of sperm, while distal valve defects are not.

7) "Of the oocytes that did successfully enter the spermatheca, many were frequently crushed when they exited through the sp-ut valve (Figure 3A'-E', Video 2-3). We observed that the sp-ut valve, labeled by DLG-1::GFP, did not completely open when the oocyte attempted to exit the spermatheca, which may lead to crushing the oocyte" – again, a defect in the sp-ut valve is not necessary to explain the phenotype because oocytes are not normally meant to pass through the sp-ut valve and the egg shell normally protects the exiting embryo.

8) Knocking down the positive calcium regulators itr-1, sca-1, and orai-1 by RNAi enhanced the pezo-1 null phenotype and knocking down the negative calcium regulator lfe-1 partially rescued the null phenotype. Based on these findings the authors write: "Therefore, these observations are consistent with the hypothesis that pezo-1 may regulate cytosolic and ER Ca^2+^ homeostasis, which is crucial for proper spermathecal contractility and dilation." – However, there is no direct evidence in the paper to support the hypothesis that pezo-1 is controlling calcium levels in the reproductive system. Furthermore, in the following section the authors observe and measure directly calcium signaling in the spermatheca and observe no defects. It appears that the authors approached this project with a strong preconception of what pezo-1 is doing (based on literature in other systems) and do not allow the observations to lead to the most plausible explanations.

9) "Surprisingly, GCaMP3 fluorescence in pezo-1 was not significantly different than wildtype" – to me this is not surprising, since the phenotype looks like a problem with sheath contraction due to absence of sperm. Perhaps if the authors imaged GCaMP3 in the sheath they would see a lower signal in the mutant, but that could also be because of the absence of sperm and not a direct role of pezo-1 in calcium release.

10) "It should be noted that we only imaged the GCaMP3 reporter during the very first three ovulations in young adult animals to avoid Ca^2+^ signaling interference from a distorted gonad morphology and mechanical pressure from a gravid uterus… our data does not exclude the possibility that Ca^2+^ signaling may be more severely disrupted as the animal goes through more ovulation cycles.” – despite the technical difficulties, the authors could image older worms in order to rule out or prove that Ca^2+^ signaling may be disrupted in the spermatheca.

11) I'm confused about the data regarding sperm navigation to the spermatheca. In subsection Sperm from matings rescues low brood size phenotype in pezo-1 mutants it is shown that pezo-1 mutant hermaphrodites resumed ovulation and fertilization upon mating once the male's sperm (from either wildtype, spe-9(hc52ts), or pezo-1 males) reached the spermatheca. From this we can conclude that pezo-1 spermatheca have no problem to attract sperm. (the same conclusion is also drawn from the AID experiment in which pezo-1 is degraded in the somatic tissues and rescued by mating with wt male sperm). However, in subsection Sperm guidance and navigation is disrupted in pezo-1 mutants it is shown that in pezo-1 hermaphrodites few sperm reach the spermatheca after mating and most sperm remained in zones furthest from the spermatheca. This was observed for both wildtype and pezo-1 mutant male sperm in mating with pezo-1 hermaphrodites. To me, these two results are contradictory, and the authors must clarify this point. Are pezo-1 hermaphrodites defective in attracting sperm to the spermatheca? And if so, how can mating rescue their phenotype?

12) Auxin Induced Degradation is performed by tagging pezo-1 with the degron sequence and crossing with tir-1::mRuby expressing worms. However, the authors don't show any data regarding the efficacy of the degradation. Since the protein is membranal, it is expected that ubiquitination of the degron might not lead immediately to degradation. If the authors labelled the fluorescently-tagged versions of pezo-1 they could quantify the degradation by measuring fluorescence. Another option would be to perform a Western blot, if they have an antibody that recognizes pezo-1. Without such controls it is hard to draw a conclusion from the AID experiments because we don't know how much of the protein was actually degraded in either the germline or somatic tissues. Is the reason for the weaker phenotype compared to the KO the tissue specificity or residual protein?

13) Figure 8 panels H-L – the text describing these results is not clear and the labeling of two of the images must be wrong because they are all labelled with Ppie-1::tir-1::mRuby but in the graph two bars are labelled with Peft-3. In the text it is hard to follow in which case auxin was added, what the promoter was and what the result was. From the text I understood that there was a sperm navigation defect when pezo-1 was degraded in the somatic tissues, but from the graph it appears the sperm navigation defect was only observed when pezo-1 was degraded in the germline.

14) The rationale behind the experiments with inx-14 and inx-22 is not clear and I disagree with the authors conclusion that "the enhancement of phenotypes with these innexins suggests that PEZO-1 may be involved in regulating inter-tissue signaling.". In my view this is a sick + sick = sicker scenario and without further experiments to directly tie pezo-1 to inter-tissue signaling the authors cannot make such claims.

15) Similarly, the observation of yolk accumulation in the pseudocoelomic cavity surrounding the gonad of the pezo-1 mutants is an intriguing observation that could be developed with further experiments (in a future manuscript), but on its own does not warrant a place in this paper. The suggesting that yolk endocytosis into the oocytes is defective and that the reduced endocytosis of yolk may disrupt prostaglandin synthesis in the oocyte, which may lead to a defect in the oocytes' ability to attract sperm towards the spermatheca is super speculative and does not belong in the Results section.

Reviewer #3:

General assessment: This paper describes a functional analysis of pezo-1, the sole PIEZO channel ortholog in *C. elegans*. PIEZO channels are involved in mechanotransduction in a variety of systems and contexts. Here, several mutations are created and analyzed including deletions, an early stop, and gain-of-function mutations associated with human disease. pezo-1 is widely expressed in different worm tissues, notably reproductive tissues including both somatic and germline, and disruption of pezo-1 leads to fecundity defects. Assays for reproductive processes implicate defects in ovulation and sperm targeting that likely lead to this decreased fertility. While calcium signaling appears normal, mutations in genes involved in calcium signaling enhance pezo-1 as does certain disruptions of gonadal signaling. Tissue-specific rescue experiments and auxin-induced degradation are used to analyze the focus of action of pezo-1 in different reproductive processes. This paper covers a lot of ground: analysis of several distinct phenotypes in pezo-1 mutants, calcium imaging, genetic interactions, interrogation of disease alleles, and use of the relatively new auxin-induced degradation system to assess tissue specific effects. Overall, the experiments fit together well and make a coherent story.

While I have extensive comments, my major specific questions and concerns are about the following, described in more detail below (*):

1) Description and interpretation of experiments involving spe-9

2) Controls for the experiment shown in Figure 6E

3) Compared to other experiments, the inx genetic interaction experiments are relatively uninformative, and I suggest toning down the interpretation of the results. I agree that pezo-1 appears to be required for "inter-tissue signaling" but it is not clear how many cases are direct.

4) Auxin experiment controls

5) Details of sperm navigation experiments

Comments:

Re: Widespread expression of PEZO-1:

Multiple reporter knock-ins are used to examine where PEZO-1 is expressed. Images in Figures 1 and Figure 1—figure supplement 1 clearly show that PEZO-1 is expressed in several tissues including germ line. However, whether different transgenes showed consistent expression patterns should be stated. A schematic of the *C. elegans* gonad would help non-experts interpret the images and some added labeling would be useful.

In particular:

In Figure 1B, label intestine

For Figure 1F, additional description and labeling is needed for context. Does the lower right of the panel show developing sperm, developing oocytes, or something else?

For 1G, label both oocytes and the central structure (intestine?).

For 1J, indicate the sheath cells with an arrow.

Re Figure 1—figure supplement 1D: Why are both "GFP" and mScarlet shown in this image; and what structures is GFP associated with?

Subsection “PEZO-1 is expressed in multiple tissues throughout development”. This section, especially the statement "GFP::PEZO-1 is expressed in both spermathecal valves until the valves open" might be read to indicate that PEZO-1 localization changes during valve opening or other events of ovulation. Was this observed? This should be clarified, and any data indicating change in localization would need to be pointed out.

Re: Deletion of pezo-1 reduces brood size

These data show several different deletion/ putative loss of function alleles of pezo-1 were generated and give similar – though different strength – phenotypes of reduced fecundity and embryonic inviability. The fecundity defect worsens with age of the hermaphrodite. Treatment with a PIEZO agonist causes a phenotype similar to that of deletion alleles. It is notable that both lof and gof cause similar phenotypes. How do the authors interpret this effect? Is it consistent with other studies of PIEZO activity in other systems and the known mechanisms of action of these channels?

Since treatment with either of two different concentrations of the agonist yields similar effects, it does not seem necessary to show both concentrations. If a wider range of concentrations was tested and gave dose-dependent effects, this could be informative. Some deletion mutants might have a different dose-response curve than the wild type.

It seems like more could be done to take advantage of this pharmacological tool, though this is not necessary.

Re: severe ovulation defects

Live observations of ovulation reveal defects in sheath contractions and valve opening that move the oocyte/embryo forward through the gonad.

These data are convincing, though it would make sense to group the observations about oocyte crushing, oocytic masses, etc. from the previous section of the results with this section – or to simply combine the two sections into one.

Re: PEZO-1 mutants genetically interact with cytosolic Ca++ regulators/ show normal calcium signaling during ovulation

While most of the ms is nicely written, this section is hard to follow.

*In this reviewer's opinion, the term "genetically interact" can be problematic since it is neither precise for geneticists nor very accessible to non-geneticists. Even though calcium imaging appears normal in the pezo mutant, these experiments establish a potential link between the mechanisms of PEZO-1 and canonical PIEZO channels. It would be clearer for most readers to frame them in terms of manipulating Ca and avoid terms like positive or negative genetic interactions or phrases like "enhancing the reduction of brood size". This is especially important because these are such interesting experiments.

Brood size data are presented as scatterplots with separate counts of early progeny and later progeny. For most of the analysis, it is unclear why splitting the data in this manner is necessary. In many cases, it makes it more difficult for a reader to compare the phenotypes of different strains, e.g. for comparing pezo-1 alleles and in the genetic interaction experiments.

In addition, the wild-type brood sizes are surprisingly variable. Does the variability decrease if full brood sizes are considered? One explanation for this could be variability in staging L4s. In turn, this raises the question of whether splitting the data is the most accurate way to show brood sizes (at least in cases where progeny are abundant at both time points – when essentially all offspring are produced early, splitting the data is indeed helpful).

Overall, it seems worth considering 1- if the data could be presented in a simpler format for some experiments and 2- if the format is not revised, if the full brood counts should be included in the supplemental data.

Re: Sperm from matings rescues pezo-1 low brood size

The experiments here show the result that pezo-1 does not appear to be required in sperm and sperm defects are not responsible for pezo-1 reproductive phenotypes.

-pezo-1 males are fully fertile in crosses to fem-1 females, indicating their sperm are functional for migration and fertilization. This is convincing.

-Crossing males to sperm-depleted pezo-1 hermaphrodites induces resumption of offspring production, indicating pezo-1 is not required for the sperm-to-ovulation signal. This is also convincing.

The authors show that signaling from oocyte to sperm, needed for navigation, is disrupted; both wild-type and pezo-1 sperm have localization defects within pezo-1 hermaphrodites. However, there do not appear to be defects in the signaling from sperm to oocytes/sheath cells that increases ovulation rate. When sperm are depleted, ovulation rate is low, but re-introduction of sperm is sufficient to induce ovulation. Furthermore, pezo-1 sperm introduced into females do fertilize oocytes, implying they are functional for migration, fertilization, and induction of ovulation.

*The way experiments in this section are presented is misleading. Based on the title, I expected to see results that pezo-1 reproductive defects are due to their sperm. The presentation of the spe-9 experiment, as a test of "whether sperm signaling was defective" as well as the description of the result, further imply a sperm-based defect.

Ultimately, the spe-9 result is not surprising. spe-9 sperm are known to signal ovulation in wild type, and it is expected that re-introduction of sperm into a depleted hermaphrodite would increase ovulation rate if oocytes can respond to MSP. The data here are fine but the text should be revised for clarity and accessibility. This also applies in the Abstract and spe-9 experiments done with disease alleles.

The authors do not comment on quantitative differences between crosses into wild-type hermaphrodites and into pezo-1 hermaphrodites, and statistics are only presented for non-mated herm vs various mated conditions. Have the authors considered these comparisons?

*To interpret 6E, additional controls are needed in which wild-type male sperm are supplied and ovulation rates are measured.

Direct measurement of ovulation rate – as shown in Figure 6E for spe-9 – would be a more direct assay for induction of ovulation by introduced sperm.

An n of 4-6 seems very low for these experiments.

*Re: Sperm navigation is disrupted

– Does zone 3 include only the spermatheca, as described in the text, or the spermatheca and adjacent region of the uterus, as shown in Figure 7 (and as it is usually defined in publications from the Miller lab and others)? Please clarify the text.

– Were sperm counted in a single focal plane, or throughout the gonad?

– It is unclear why 72 hr adult hermaphrodites were used for these experiments; younger animals are more typically assayed. What is the rationale for this?

– The images in Figure 7 show 72 hr adults; the quantification is with 60 hr adults. While it might seem unlikely, this difference could matter. At 60 hours, there is likely to be some hermaphrodite sperm remaining, at least in wild type, while at 72 hr sperm are more likely to be depleted, which would presumably alter the signaling environment. Related to this concern, there appear to be quite a few sperm in zones 2 and 3 in panel G. The data in 7H suggests fewer sperm should be visible in zone 3.

– There is a mismatch between the callouts and labels for the image panels in Figure 7. According to the text, 7B,D,F should show wild-type hermaphrodites; according to labels, this is B,D,E.

– The order of image panels in Figure 7 is confusing, in part because some crosses are not shown. One could add the two "missing" combinations (pezo-1 x WT and WT x pezo, zone 1-2). Alternatively, 7B and 7F could be moved to supplementary data or not shown.

– In 7H: Are the data from one replicate, so that the error bars reflect worm-to-worm variability, or are the data from the 3 repeats, so the error bars reflect differences among the average distribution in each experiment?

– For indicating p values in 7H, brackets should be used to show what is being compared. Presumably. the current comparisons are 1 vs 2, 3 vs 4; comparing the same male sperm in 2 different hermaphrodites. It would be interesting to add statistical comparisons for 1 vs 3, 2 vs 4, i.e. comparing different male sperm in the same hermaphrodites.

Re: Auxin experiments

– The use of auxin induced degradation to disrupt function is a nice way to try to examine tissue-specific effects.

*– Data presented in 8E,F,G show the different degron strains either untreated or treated with auxin. A control is needed for treatment with auxin in the absence of the degron transgene – especially since all strains undergo a similar reduction in brood size in the presence of auxin (8E,F)

– Figure 8C,D – The sperm expression is described as faint, and it is indeed hard to see in the images. It is not unexpected that expression might be low, but are the authors confident that the putative sperm expression is not autofluorescence?

– It should be made clear in the text that germline expression includes (or is likely to include) both sperm and oocytes, so that either tissue could be the primary course of phenotypes observed with the germline AID strains. I do agree that it is more likely to be due to oocyte/ attractant signaling defects.

– The text states that Figure 8H, J show the somatic-specific (Peft-3) AID strain with auxin, but the figure is labeled as the Ppie-1 strain without auxin. Which is it?

– The age of the hermaphrodites used for the mitotracker assays needs to be stated. This is relevant to whether or not self sperm could be contribute to targeting defects.

Re: Multiple roles in inter-tissue signaling

– *The data in Figure S5 demonstrate enhancement of the brood size defects in inx; pezo as compared to inx(RNAi) or pezo- alone. However, this could be interpreted in many different ways, and does not necessarily mean that the same process is being affected, especially when a relatively non-specific phenotype is the assay. Thus, this experiment does not add insight.

– The images in Figure S6 do show excess extracellular yolk in pezo-1 that is not present in WT. However, YP170 levels in pezo-1 oocytes appear higher than in wild type, if anything. Therefore, I am not convinced that these data provide evidence for defects in YP170 endocytosis. Instead, is it possible that there are defects within oocytes, in conversion of yolk to downstream signaling molecules? Without additional experiments, this also seems to be dispensable.

Overall, while the experiments in this section do not contradict the model of pezo-1's being involved in inter-tissue signaling, they do little to support it.

---

## [Author Response]

Essential revisions:1) Because the overall set of data do not directly support that PIEZO in *C. elegans* is a calcium regulator, the authors should revise the manuscript, particularly many of the conclusions reached, to less strongly advocate that their data supports a role for this *C. elegans* homolog in calcium signaling. The reviewers agree this, in fact, does not detract from the interest of the paper but instead reveals important information about the potential function of these proteins in different developmental contexts.

Thank you for this feedback. We have revised the manuscript extensively to tone down our conclusions about calcium signaling. You will see evidence of this throughout the manuscript.

2) The experiments and explanations of roles in sperm signaling (in particular comments from reviewer 3 on spe-9 experiments) and experiments and conclusions about sperm navigation were either incomplete or confusing. It is not clear why the defects may not arise as secondary effects from the oocyte production defects. The authors should address all reviewer comments in order to re-evaluate the conclusions that can be reached regarding these experiments in order to make claims about such roles.

We have made extensive changes to these sections and have added better controls and even a few new experiments. We made *pezo-1* females to determine if some of our phenotypes were dependent on the presence of self-sperm. We show in Figures 7 and Figure 6—figure supplement 1 (new figure) that hermaphrodites have a dramatic reduction in brood size with almost no self-sperm returning to the spermatheca, while cross-sperm are much more successful at siring progeny and navigating to the spermatheca where they can fertilize oocytes. We do still believe that there is also a signaling problem from the sheath cells to attract both self-sperm and cross-sperm to the spermatheca. We discuss these results in great detail in the text. All comments have been addressed.

3) There should be more controls (or better explanation that address the need for controls) as requested by reviewers for some of the experiments presented, particularly the AID, spe-9, and Yoda1 experiments.

We have addressed each of these concerns. The controls for the AID, spe-9, and Yoda1 experiments have been added to the appropriate sections of the manuscript. Please also check out Figures 2C, Figure 6, Figure 8E, and Figure 8—figure supplement 2 that were added.

4) Evidence for inter-tissue signaling is not well supported and thus, unless the authors have additional, stronger data, should be removed from the manuscript.

We have removed the discussion of yolk protein and have limited our discussion of inter-tissue signaling solely to the sperm migration defect. To address the apparent contradictory data that *pezo-1* self-sperm are defective in navigating back to the spermatheca after being expelled by each ovulation compared to male sperm mated in from *pezo-1* mutant males that navigate quite well, we have done the following:

We added more text to this section of the paper to explain these experiments.

In Figure 6—figure supplement 1, you will also notice that we are careful to distinguish between self-sperm and cross-sperm from male matings. We believe there may be a difference between the two populations of sperm. The data show that male sperm were also stuck in the uteri in *pezo-1 CΔ; fem-1(hc17)* after mating, however, 40-50% of the stained sperm able to navigate to the spermatheca. Therefore, we believe there may be a difference between the two populations of sperm. These are observations we plan to pursue in the future.

Overall, we would like authors to 1) revise the manuscript to be careful about claims that PEZO-1 functions as a calcium regulator and to consider other roles or functions that may be revealed by their data , 2) revise the presentation and provide controls (or potentially add new data if available to strengthen their points) about claims for PEZO-1 function in sperm signaling or navigation. 3) provide controls for other experiments requested in individual reviewer comments. Reviewers felt that these issues should not warrant significant new experimental analysis.

Thank you for this feedback. We have addressed all of these concerns as can be seen in the main text and figures.

Reviewer #1:The manuscript describes the characterization of the function of PEZO-1 in *C. elegans*. PEZO-1 is the homolog of PIEZO1 and PIEZO2 in humans, which are mechano-sensitive channel proteins. The authors generate GFP-tagged versions of PEZO-1 to show that it is expressed broadly in many cell types, including reproductive tissues. They examine knock-out mutants of pezo-1 and find severe defects in fertility. They find that loss of pezo-1 causes oocytes to be crushed during the processes of ovulation, fertilization, and expulsion to the uterus. They further show that defective sperm navigation may also contribute to fertility defects. The authors have evidence that PEZO-1 works through calcium signaling – mutation in pezo-1 shows synergistic defects when combined with reduction of calcium signaling regulators, but calcium signaling itself was not different in their assays. Both tissue-specific knock-down of *C. elegans* pezo-1 and introduction of PIEZO patient-specific alleles in *C. elegans* also lead to fertility defects.The work is interesting and novel. This is the first characterization of PIEZO function in *C. elegans*, where it is possible to use a variety of approaches to assess PIEZO function in a model organism. The high-quality data obtained through a combination of genetic knock-downs and AID degradation, in vivo imaging of tagged proteins, fertility assays, and calcium imaging is a strength of the work. Though PIEZO proteins are known to be important for mechanosensory transduction in several contexts in other systems, including humans and Drosophila, this is also the first demonstration that PEZO-1 is important for fertility, in particular, the production of germ cells that requires an orchestration of movements. In particular, the 'crushed oocyte' phenotype is quite striking. Their demonstration that PIEZO patient-specific alleles also result in fertility defects links the *C. elegans* work to understanding human disease and show that there is potential to use this system in the future to further dissect PIEZO function in developmental contexts in a tractable system. Because of these numerous strengths, I would recommend publication in eLife.There are a few revisions that are required. One modest concern which is that the role of PEZO-1 in mechanical signal transduction is still not described in a concrete way. Of course, this is partially due to the obviously complicated nature of the fertility process that is affected. There are multiple points at which the mechano-sensation and transduction may be affected that result in the major phenotype of oocyte crushing. This could be improved by revising Figure 10. Instead of highlighting 'PEZO-1 Dysfunction' it would be clearer for the authors to highlight where they think PEZO-1 IS functioning. This could be done by either by adding a clear point at each step that describes that function or retitling each step to better highlight PEZO-1. The model as is now (in the text and figure) is more a reiteration of results and not so much a model for how and where PEZO-1 may be 'sensing' or 'responding' to mechanical stress. It is actually interesting that the phenotype is complicated because PIEZO proteins in other organisms may also have likewise complex roles.

This is a great suggestion and we have totally revised this Figure to emphasize the many processes we think PEZO-1 may influence during fertilization. We no longer reiterate the defects we already described throughout the paper. We believe our discussion of our data and our working hypothesis is better articulated.

Reviewer #2:This paper explores for the first time the role of the piezo-1/2 ortholog in *C. elegans*, pezo-1, and it identifies a critical function in retaining sperm in the spermatheca. In the absence of pezo-1 sperm are quickly depleted from the spermatheca and as a consequence embryo production is halted and oocyte crushing begins on both ends of the spermatheca. The quality of the work is high and its results are novel and of broad interest. However, the paper suffers from several problems that must be rectified before publication. Likely because piezo-1/2 are known to be mechanosensing calcium channels, the authors probably started this project with the hypothesis that pezo-1 functions as a mechanosensitive calcium channel. However, nature being as it is, in the end they don't have any evidence for such a function. That in itself is not a problem. What is problematic in my view is that in writing the paper they nevertheless continue to push a narrative of pezo-1 regulating calcium in the reproductive system, despite there not being any evidence for this. The second problem I have as a reviewer with the current paper is that the interpretation of many of the experiments is taken too far and it is done in the Results section, which could lead a reader to confuse speculations (or wishful thinking) for actual results. The defect in pezo-1 mutants that can explain all of the observed phenotypes is the rapid depletion of sperm from the spermatheca because they fail to return after being washed out with embryos exiting into the uterus. The mating experiments and somatic or germline specific AID experiments were supposed to clarify whether the sperm navigation defect is due to lack of pezo-1 activity in the sperm or in the somatic gonad. Unfortunately, the data presented is contradictory (or perhaps only not explained well) because after reading and rereading the paper I still can't figure out if pezo-1 is required in the sperm or in the gonad or both. Once the authors clarify this point, remove experiments that don't lead to a conclusion (inx-14/22, YP170::tdimer2), remove overinterpretation statements from the restuls section, and add missing controls for some experiments (Yoda1, AID) as detailed in my comments below, I believe the paper will be worthy of publication in eLife.

Thank you for these comments. We have addressed the problematic parts of this manuscript, toned down our narrative about calcium signaling, and removed a number of uninformative experiments that do not add to the general observations of ovulation and sperm navigation. Below we address each concern.

1) In multiple places the authors jump to conclusions based on associative evidence such as localization or genetic interaction. For example: "Notably, PEZO-1 is strongly expressed in several tubular tissues, including the pharyngeal-intestinal and spermathecal-uterine valves, which is consistent with our hypothesis that pezo-1 may be responsible for mechanoperception in these tissues." And: "Consistent with the hypothesis that reproductive tissues are regulated by mechanosensitive stimuli in *C. elegans*, expression of PEZO-1 likely functions to sense physical strain or contractility during ovulation and fertilization." – such sentences push a narrative of pezo-1 function in mechanoperception without any support in the data. Speculations and interpretations belong in the discussion. The result section should present the results without biasing the reader to a single interpretation.2) "Partial co-localization of PEZO-1 with the ER marker SP12::GFP suggested that PEZO-1 may be processed in the ER and transported to the plasma membrane (Figure 1E)." – based on the image in Figure 1E the localization of SP12 and PEZO-1 is complementary and primarily non-overlapping. If the authors have reason to believe there is substantial co-localization they need to show and quantify it. In any case, it isn't clear why this is an important point to focus on. If the authors think the location of function of PEZO-1 is the plasma membrane then why not just focus on that localization?

We have omitted Figure 1E; it is not an important point at all. We agree that our focus should be on the plasma membrane.

3) "The genome-edited animals behaved normally, suggesting no functional disruption of tagging PEZO-1 with these fluorescent reporter genes." – This statement is questionable in light of what is visible in Figure 1F, where there is an accumulation of small oocyte pieces or vesicles or something else that looks abnormal between the -1 oocyte and spermatheca.

Thanks so much for pointing that out. What an oversight on our end not to have been more specific in the figure legend. Those “pieces” are only observed in adults just before the first ovulation event and are the few remaining spermatids that have not migrated into the spermatheca yet. The larger pieces are residual bodies that have yet to be engulfed by the sheath cells. We have added this to the legend and also cited the paper that originally made these observations. This figure actually highlights quite well that PEZO-1 is on sperm membranes.

4) "The fluorescent signal of GFP::PEZO-1 is observed in both spermathecal valves until the valves open, suggesting that PEZO-1 may function to sense the mechanical stimuli at the valves during ovulation" – since the fluorescent protein is at the membrane it is expected to give a higher signal when the tissue is contracted compared to when it is dilated. The change in fluorescence intensity observed in the GFP::PEZO-1 movie on its own is not an indication of any mechanosensing. If the authors wish to claim enrichment in the valve at a certain time point they would need to do a ratiometric comparison with a membrane marker such as PH::mCherry.

This is an excellent point and we have removed all comments suggesting quantitative differences in expression in the spermatheca.

5) "To mimic a gain-of-function phenotype in pezo-1, we fed wildtype animals with Yoda1, a PIEZO1 specific chemical agonist, which keeps the channel open" – No control was performed to show that Yoda1 doesn't have side effects in worms. There are plenty of chemicals that are supposedly specific but actually are not. A simple experiment that could at least show Yoda1's phenotype is PEZO-1-dependent is to perform the treatment on pezo-1KO, which we expect to be refractory to the drug and display the same brood size as pezo-1KO on its own.

We agreed with the reviewer’s comments and added the control experiments with the treatment of Yoda-1 on *pezo-1 CΔ* in Figure 2C.

6) "The defective ovulation is likely due to incomplete constriction of the sheath cells and improper gating of the distal spermathecal valve." – Incomplete constriction of the sheath is sufficient to explain the entry phenotype and it is not necessary to invoke improper gating of the distal spermathecal valve, which was not specifically tested. Moreover, improper sheath contraction is consistent with the absence of sperm, while distal valve defects are not.

We revised the sentence to “The defective ovulation is likely due to incomplete constriction of the sheath cells”. Thanks for that suggestion.

7) "Of the oocytes that did successfully enter the spermatheca, many were frequently crushed when they exited through the sp-ut valve (Figure 3A'-E', Video 2-3). We observed that the sp-ut valve, labeled by DLG-1::GFP, did not completely open when the oocyte attempted to exit the spermatheca, which may lead to crushing the oocyte" – again, a defect in the sp-ut valve is not necessary to explain the phenotype because oocytes are not normally meant to pass through the sp-ut valve and the egg shell normally protects the exiting embryo.

Though it is true that oocytes do not normally pass through the sp-ut valve in wild-type animals, oocytes from *spe* mutants, which are not fertilized, do survive their transit through the sp-ut valve and the vulva without being crushed. Many *spe* mutants are readily identifiable because of the number of undamaged oocytes on the plate.

8) Knocking down the positive calcium regulators itr-1, sca-1, and orai-1 by RNAi enhanced the pezo-1 null phenotype and knocking down the negative calcium regulator lfe-1 partially rescued the null phenotype. Based on these findings the authors write: "Therefore, these observations are consistent with the hypothesis that pezo-1 may regulate cytosolic and ER Ca^2+^ homeostasis, which is crucial for proper spermathecal contractility and dilation." – However, there is no direct evidence in the paper to support the hypothesis that pezo-1 is controlling calcium levels in the reproductive system. Furthermore, in the following section the authors observe and measure directly calcium signaling in the spermatheca and observe no defects. It appears that the authors approached this project with a strong preconception of what pezo-1 is doing (based on literature in other systems) and do not allow the observations to lead to the most plausible explanations.

Guilty as charged. We have toned down the entire emphasis on calcium signaling throughout the manuscript.

9) "Surprisingly, GCaMP3 fluorescence in pezo-1 was not significantly different than wildtype" – to me this is not surprising, since the phenotype looks like a problem with sheath contraction due to absence of sperm. Perhaps if the authors imaged GCaMP3 in the sheath they would see a lower signal in the mutant, but that could also be because of the absence of sperm and not a direct role of pezo-1 in calcium release.

We agree and hope to pursue calcium signaling in specific tissues involved in this process in the future. Sheath-specific imaging of calcium is definitely on our list for the future.

10) "It should be noted that we only imaged the GCaMP3 reporter during the very first three ovulations in young adult animals to avoid Ca^2+^ signaling interference from a distorted gonad morphology and mechanical pressure from a gravid uterus… our data does not exclude the possibility that Ca^2+^ signaling may be more severely disrupted as the animal goes through more ovulation cycles.” – despite the technical difficulties, the authors could image older worms in order to rule out or prove that Ca^2+^ signaling may be disrupted in the spermatheca.

We actually tried this and because of the broken oocytes both in the uterus and some in the oviduct, these images were far too confusing to make sound conclusions. Furthermore, the bigger problem that we have yet to resolve is that older animals do not ovulate under our imaging conditions, making it impossible to image calcium dynamics even if the animals did not have crushed oocytes in their uteri. The process of squeezing the animals between a coverslip and an agarose pad often pops older and larger wild-type animals, making movies of later embryo transits harder to image.

11) I'm confused about the data regarding sperm navigation to the spermatheca. In subsection Sperm from matings rescues low brood size phenotype in pezo-1 mutants it is shown that pezo-1 mutant hermaphrodites resumed ovulation and fertilization upon mating once the male's sperm (from either wildtype, spe-9(hc52ts), or pezo-1 males) reached the spermatheca. From this we can conclude that pezo-1 spermatheca have no problem to attract sperm. (the same conclusion is also drawn from the AID experiment in which pezo-1 is degraded in the somatic tissues and rescued by mating with wt male sperm). However, in subsection Sperm guidance and navigation is disrupted in pezo-1 mutants it is shown that in pezo-1 hermaphrodites few sperm reach the spermatheca after mating and most sperm remained in zones furthest from the spermatheca. This was observed for both wildtype and pezo-1 mutant male sperm in mating with pezo-1 hermaphrodites. To me, these two results are contradictory, and the authors must clarify this point. Are pezo-1 hermaphrodites defective in attracting sperm to the spermatheca? And if so, how can mating rescue their phenotype?

Our data is contradictory. We do show that mating into *pezo-1* hermaphrodites does rescue the ovulation and brood size. So mated sperm, whether from wild-type males or from *pezo-1* males, can navigate to the spermatheca to fertilize oocytes. We now suspect that there is a defect in hermaphrodite self-sperm and their ability to navigate back to the spermatheca after being expelled during ovulation. This remains an interesting contradiction and one that we are currently pursuing. We had hoped to report it as one of the complexities of the *pezo-1* phenotype. We have added a few sentences to this section of the Discussion to be open about this apparent contradiction in sperm phenotypes. We address this concern above with reviewer #1’s comment 2. Please refer to Figure 6—figure supplement 1. The sperm distribution was affected in *pezo-1 CΔ; fem-1(hc17)* after mating, however, there are still 40-50% stained sperm able to navigate to the spermatheca (Figure 6—figure supplement 1E). Additionally, the fertilization rate of the laid embryos (we used the total sperm number here since the sperm in the uteri are still able to crawl back to spermatheca) is lower in *pezo-1 CΔ; fem-1(hc17)* at permissive temperature (15° C) after mating with both wild type and *pezo-1 CΔ* when compared to *fem-1(hc17)* only.

12) Auxin Induced Degradation is performed by tagging pezo-1 with the degron sequence and crossing with tir-1::mRuby expressing worms. However, the authors don't show any data regarding the efficacy of the degradation. Since the protein is membranal, it is expected that ubiquitination of the degron might not lead immediately to degradation. If the authors labelled the fluorescently-tagged versions of pezo-1 they could quantify the degradation by measuring fluorescence. Another option would be to perform a Western blot, if they have an antibody that recognizes pezo-1. Without such controls it is hard to draw a conclusion from the AID experiments because we don't know how much of the protein was actually degraded in either the germline or somatic tissues. Is the reason for the weaker phenotype compared to the KO the tissue specificity or residual protein?

Thank you for this criticism. We have repeated these experiments with our GFP-tagged PEZO-1 to address the level of knockdown and have observed that at least 2-3 fold of fluorescent intensity of GFP::PEZO-1::Degron was reduced when the animal treated with auxin compared to non-auxin control. These results are now shown in the Figure 8—figure supplement 2.

13) Figure 8 panels H-L – the text describing these results is not clear and the labeling of two of the images must be wrong because they are all labelled with Ppie-1::tir-1::mRuby but in the graph two bars are labelled with Peft-3. In the text it is hard to follow in which case auxin was added, what the promoter was and what the result was. From the text I understood that there was a sperm navigation defect when pezo-1 was degraded in the somatic tissues, but from the graph it appears the sperm navigation defect was only observed when pezo-1 was degraded in the germline.

We have addressed these errors in the text and Figure 8. Panels H-L are now G-J and should be much easier to follow now. We do not show DIC with MitoTracker images for the Peft-3 strain because there was no significant difference from controls. We only show the bar graph for Peft-3 in panel 8K to demonstrate that it is not significantly different with or without auxin.

14) The rationale behind the experiments with inx-14 and inx-22 is not clear and I disagree with the authors conclusion that "the enhancement of phenotypes with these innexins suggests that PEZO-1 may be involved in regulating inter-tissue signaling.". In my view this is a sick + sick = sicker scenario and without further experiments to directly tie pezo-1 to inter-tissue signaling the authors cannot make such claims.

We agree with the reviewer’s comment and removed the data for future study.

15) Similarly, the observation of yolk accumulation in the pseudocoelomic cavity surrounding the gonad of the pezo-1 mutants is an intriguing observation that could be developed with further experiments (in a future manuscript), but on its own does not warrant a place in this paper. The suggesting that yolk endocytosis into the oocytes is defective and that the reduced endocytosis of yolk may disrupt prostaglandin synthesis in the oocyte, which may lead to a defect in the oocytes' ability to attract sperm towards the spermatheca is super speculative and does not belong in the Results section.

Thank you for this feedback. We agree that this was highly speculative and have omitted the figure. We do speculate in the Discussion that prostaglandin synthesis could be perturbed since that is a known attractant for sperm to migrate to the spermatheca, but we keep that short and make it clear that this is pure speculation.

Reviewer #3:General assessment: This paper describes a functional analysis of pezo-1, the sole PIEZO channel ortholog in *C. elegans*. PIEZO channels are involved in mechanotransduction in a variety of systems and contexts. Here, several mutations are created and analyzed including deletions, an early stop, and gain-of-function mutations associated with human disease. pezo-1 is widely expressed in different worm tissues, notably reproductive tissues including both somatic and germline, and disruption of pezo-1 leads to fecundity defects. Assays for reproductive processes implicate defects in ovulation and sperm targeting that likely lead to this decreased fertility. While calcium signaling appears normal, mutations in genes involved in calcium signaling enhance pezo-1 as does certain disruptions of gonadal signaling. Tissue-specific rescue experiments and auxin-induced degradation are used to analyze the focus of action of pezo-1 in different reproductive processes. This paper covers a lot of ground: analysis of several distinct phenotypes in pezo-1 mutants, calcium imaging, genetic interactions, interrogation of disease alleles, and use of the relatively new auxin-induced degradation system to assess tissue specific effects. Overall, the experiments fit together well and make a coherent story.While I have extensive comments, my major specific questions and concerns are about the following, described in more detail below (*):1) Description and interpretation of experiments involving spe-9

We revised the text and carefully interpret the spe-9 data in the manuscript.

2) Controls for the experiment shown in Figure 6E

We added new control data and revised the figure.

3) Compared to other experiments, the inx genetic interaction experiments are relatively uninformative, and I suggest toning down the interpretation of the results. I agree that pezo-1 appears to be required for "inter-tissue signaling" but it is not clear how many cases are direct.

We agree with the reviewer’s comment and remove the inx genetic interaction data for future study.

4) Auxin experiment controls

We added the AID control data in the Figure 8 and Figure 8—figure supplement 2.

5) Details of sperm navigation experiments

We revised the figure and text and added a few new experiments to clarify the experiment.

Comments:Re: Widespread expression of PEZO-1:Multiple reporter knock-ins are used to examine where PEZO-1 is expressed. Images in Figures 1 and Figure 1—figure supplement 1 clearly show that PEZO-1 is expressed in several tissues including germ line. However, whether different transgenes showed consistent expression patterns should be stated. A schematic of the *C. elegans* gonad would help non-experts interpret the images and some added labeling would be useful.

We appreciate the reviewer’s comment regarding the expression pattern of PEZO-1. Dr. Paul Sternberg’s lab reported the expression patterns of PEZO-1, consistent with our data. They had expressed GFP driven by the *pezo-1* promoter (Abstract 739C at 22^nd^ International *C. elegans* conference). We have added a schematic of the *C. elegans* gonad in Figure 1 F and better labeling throughout our figures.

In particular:In Figure 1B, label intestine

Done

For Figure 1F, additional description and labeling is needed for context. Does the lower right of the panel show developing sperm, developing oocytes, or something else?

We have clarified what these cells are in the legend.

For 1G, label both oocytes and the central structure (intestine?).

Done.

For 1J, indicate the sheath cells with an arrow.

Done.

Re Figure 1—figure supplement 1D: Why are both "GFP" and mScarlet shown in this image; and what structures is GFP associated with?

We have added more detail to the legend as to what structures are GFP+. It turns out that our GFP fusion lights up the male tail fan, a sensory structure needed for mating. Anterior to that is the cloaca/spicules also lighting up green. We made these trans-heterozygotes to show that the two fusion proteins overlap but also have some distinct localizations. Given that 8 of the 14 isoforms would contain the N-terminal GFP fusion, this image suggests that only the full length forms are expressed in the male tail fan.

Subsection “PEZO-1 is expressed in multiple tissues throughout development”. This section, especially the statement "GFP::PEZO-1 is expressed in both spermathecal valves until the valves open" might be read to indicate that PEZO-1 localization changes during valve opening or other events of ovulation. Was this observed? This should be clarified, and any data indicating change in localization would need to be pointed out.

We corrected the language here as not to suggest any change in localization.

Re: Deletion of pezo-1 reduces brood sizeThese data show several different deletion/ putative loss of function alleles of pezo-1 were generated and give similar – though different strength – phenotypes of reduced fecundity and embryonic inviability. The fecundity defect worsens with age of the hermaphrodite. Treatment with a PIEZO agonist causes a phenotype similar to that of deletion alleles. It is notable that both lof and gof cause similar phenotypes. How do the authors interpret this effect? Is it consistent with other studies of PIEZO activity in other systems and the known mechanisms of action of these channels?

We thank the reviewer for this comment. Dysfunctions of PIEZO1 and PIEZO2 caused a variety of physiological disorders, which were caused by both gain-of-function and loss-of-function. A dogmatic model is that osmoregulation is disturbed in either gof or lof mutants, which interfered with downstream cellular signaling pathways. However, the cellular and molecular mechanisms of PIEZO gain-of-function vs. PIEZO loss-of-function in these diseases are not well understood. The ovulation and fertility process are complicated and are regulated by a sophisticated signaling network. In our study, there are a few speculations about the role of PEZO-1 in reproductive signal transduction. We speculate that either loss-of-function or gain-of-function may spatiotemporally disturb the reproductive signaling pathways, which lead to a common read-out as reduced fecundity and embryonic inviability. We revised and highlighted our working model in Figure 10.

Since treatment with either of two different concentrations of the agonist yields similar effects, it does not seem necessary to show both concentrations. If a wider range of concentrations was tested and gave dose-dependent effects, this could be informative. Some deletion mutants might have a different dose-response curve than the wild type.It seems like more could be done to take advantage of this pharmacological tool, though this is not necessary.

These are good points and since we did not test a variety of doses, we now just show one dose of Yoda.

Re: severe ovulation defectsLive observations of ovulation reveal defects in sheath contractions and valve opening that move the oocyte/embryo forward through the gonad.These data are convincing, though it would make sense to group the observations about oocyte crushing, oocytic masses, etc. from the previous section of the results with this section – or to simply combine the two sections into one.

This is a good suggestion and we merged these two sections since they both highlight our studies of the ovulation defects.

Re: PEZO-1 mutants genetically interact with cytosolic Ca++ regulators/ show normal calcium signaling during ovulationWhile most of the ms is nicely written, this section is hard to follow.*In this reviewer's opinion, the term "genetically interact" can be problematic since it is neither precise for geneticists nor very accessible to non-geneticists. Even though calcium imaging appears normal in the pezo mutant, these experiments establish a potential link between the mechanisms of PEZO-1 and canonical PIEZO channels. It would be clearer for most readers to frame them in terms of manipulating Ca and avoid terms like positive or negative genetic interactions or phrases like "enhancing the reduction of brood size". This is especially important because these are such interesting experiments.

We did try to alter our language here to make it easier to read and omitted positive and negative genetic interactions. However, these genetic arguments do usually use such language as enhance or suppress when discussing phenotypes of double mutants. Since we were not directly measuring calcium, it is safer to conclude that the depletion of a given gene enhanced or suppressed the phenotypes of pezo-1 alone.

Brood size data are presented as scatterplots with separate counts of early progeny and later progeny. For most of the analysis, it is unclear why splitting the data in this manner is necessary. In many cases, it makes it more difficult for a reader to compare the phenotypes of different strains, e.g. for comparing pezo-1 alleles and in the genetic interaction experiments.In addition, the wild-type brood sizes are surprisingly variable. Does the variability decrease if full brood sizes are considered? One explanation for this could be variability in staging L4s. In turn, this raises the question of whether splitting the data is the most accurate way to show brood sizes (at least in cases where progeny are abundant at both time points – when essentially all offspring are produced early, splitting the data is indeed helpful).Overall, it seems worth considering 1- if the data could be presented in a simpler format for some experiments and 2- if the format is not revised, if the full brood counts should be included in the supplemental data.

We thank reviewer’s comments and adjusted a few figures (Figure 2A, 2C 8E, 8F) to show the total brood for the full time period. However, we did split the brood sizes into two time periods for some experiments (like Figure 4) to emphasize that the onset of the phenotypes was late. We had considered only showing brood sizes of Day2 and later adults to highlight the decrease in brood size, but thought it best to show that Day1 adults were not as affected.

Re: Sperm from matings rescues pezo-1 low brood sizeThe experiments here show the result that pezo-1 does not appear to be required in sperm and sperm defects are not responsible for pezo-1 reproductive phenotypes.-pezo-1 males are fully fertile in crosses to fem-1 females, indicating their sperm are functional for migration and fertilization. This is convincing.-Crossing males to sperm-depleted pezo-1 hermaphrodites induces resumption of offspring production, indicating pezo-1 is not required for the sperm-to-ovulation signal. This is also convincing.

Agree. Our data suggests that mutant male sperm are fully able to induce ovulation, to navigate to the spermatheca upon mating, and to fertilize oocytes.

The authors show that signaling from oocyte to sperm, needed for navigation, is disrupted; both wild-type and pezo-1 sperm have localization defects within pezo-1 hermaphrodites. However, there do not appear to be defects in the signaling from sperm to oocytes/sheath cells that increases ovulation rate. When sperm are depleted, ovulation rate is low, but re-introduction of sperm is sufficient to induce ovulation. Furthermore, pezo-1 sperm introduced into females do fertilize oocytes, implying they are functional for migration, fertilization, and induction of ovulation.

Also agree.

*The way experiments in this section are presented is misleading. Based on the title, I expected to see results that pezo-1 reproductive defects are due to their sperm. The presentation of the spe-9 experiment, as a test of "whether sperm signaling was defective" as well as the description of the result, further imply a sperm-based defect.Ultimately, the spe-9 result is not surprising. spe-9 sperm are known to signal ovulation in wild type, and it is expected that re-introduction of sperm into a depleted hermaphrodite would increase ovulation rate if oocytes can respond to MSP. The data here are fine but the text should be revised for clarity and accessibility. This also applies in the Abstract and spe-9 experiments done with disease alleles.

Thank you for these comments. We have altered the writing to make it clear that we think sperm navigation is disrupted but not because the sperm are defective but rather the attractant signal for the sperm to migrate from the uterus to the spermatheca is disrupted. By “sperm-signaling”, we were thinking the signals to the sperm, but now see how this phrase sounds like the sperm is defective in signaling. We modified the Abstract and spe-9 discussion as well. Our data however is consistent with the conclusion that the attractive signal for sperm to migrate back to the spermatheca is defective and that the mutant sperm themselves are fully capable of sensing the signal, crawling, repopulating the spermatheca, and in fem-1 females, even fertilizing oocytes. However, the feedback from the three reviewers has prompted us to ask whether there might be a defect in self-sperm versus cross sperm. Even though male cross sperm don’t navigate as well as WT, they do still crawl towards the spermatheca, whereas mutant self-sperm get washed out of the spermatheca and never make it back. We have addressed these issues by the new experiments with pezo-1 females in Figure 6—figure supplement 1.

The authors do not comment on quantitative differences between crosses into wild-type hermaphrodites and into pezo-1 hermaphrodites, and statistics are only presented for non-mated herm vs various mated conditions. Have the authors considered these comparisons?

This too is a good suggestion. The data in Figure 6B suggests that mutant sperm when mated with WT can migrate to the spermatheca and fertilize a large number of oocytes. However, when mated into the C∆ hermaphrodites, they do sire cross progeny but at greatly reduced levels. This we believe is further support that the attractive signal from the oocytes or sheath cells are somewhat defective and not that there is a problem with the sperm crawling and fertilizing oocytes. This data is also consistent with Figure 7J where we do quantitate where these cross sperm are located 60 minutes after removing the males. We have reworked this entire section to make it clearer that the signal of pezo-1 mutants to attract sperm to the spermatheca is what appears to be dysfunctional.

We have carried out an additional experiment to test whether one possible defect was in the ability of the sheath to respond to the sperm signal to trigger ovulation. Even though our data in Figure 6E suggests that just the presence of sperm can trigger ovulation, we went on to show that purified MSP can also trigger ovulation in older pezo-1 ∆ hermaphrodites that are depleted of sperm and are no longer ovulating. We added this data in Figure 6F, G, and H.

*To interpret 6E, additional controls are needed in which wild-type male sperm are supplied and ovulation rates are measured.

We added the control in Figure 6E.

Direct measurement of ovulation rate – as shown in Figure 6E for spe-9 – would be a more direct assay for induction of ovulation by introduced sperm.An n of 4-6 seems very low for these experiments.

We increased number of tested gonad arms to >10 in Figure 6E and other ovulation assays.

*Re: Sperm navigation is disrupted– Does zone 3 include only the spermatheca, as described in the text, or the spermatheca and adjacent region of the uterus, as shown in Figure 7 (and as it is usually defined in publications from the Miller lab and others)? Please clarify the text.

We defined the zone 3 as the region including the entire spermatheca and an adjacent embryo.

– Were sperm counted in a single focal plane, or throughout the gonad?

We quantified sperm throughout the gonad.

– It is unclear why 72 hr adult hermaphrodites were used for these experiments; younger animals are more typically assayed. What is the rationale for this?

We intentionally used older animals to make sure the animal was totally depleted of self-sperm.

– The images in Figure 7 show 72 hr adults; the quantification is with 60 hr adults. While it might seem unlikely, this difference could matter. At 60 hours, there is likely to be some hermaphrodite sperm remaining, at least in wild type, while at 72 hr sperm are more likely to be depleted, which would presumably alter the signaling environment. Related to this concern, there appear to be quite a few sperm in zones 2 and 3 in panel G. The data in 7H suggests fewer sperm should be visible in zone 3.

Figure 7J quantifies all of the data (n=8) that is represented by Figure 7H and 7I. Since 7J is the average of 8 animals, we had shown an animal with the most sperm in the zone 3. We have replaced 7H and 7I with an animal that more typical reflects 7J.

– There is a mismatch between the callouts and labels for the image panels in Figure 7. According to the text, 7B,D,F should show wild-type hermaphrodites; according to labels, this is B,D,E.– The order of image panels in Figure 7 is confusing, in part because some crosses are not shown. One could add the two "missing" combinations (pezo-1 x WT and WT x pezo, zone 1-2). Alternatively, 7B and 7F could be moved to supplementary data or not shown.

We also revised much of Figure 7 for clarifying the comments above. We think this figure is much easier to follow now.

– In 7H: Are the data from one replicate, so that the error bars reflect worm-to-worm variability, or are the data from the 3 repeats, so the error bars reflect differences among the average distribution in each experiment?

We repeated the experiments at least three times with 3-5 worms each to precisely control the mating time windows, and error bars reflect worm-to-worm variability.

– For indicating p values in 7H, brackets should be used to show what is being compared. Presumably. the current comparisons are 1 vs 2, 3 vs 4; comparing the same male sperm in 2 different hermaphrodites. It would be interesting to add statistical comparisons for 1 vs 3, 2 vs 4, i.e. comparing different male sperm in the same hermaphrodites.

These are all very good suggestions and we totally rearranged the data in Figure 7 and cleaned up the callouts, labels, and figure legend. We now show representative images of at least one zone for each mating combination, and quantified all in Figure 7J.

Re: Auxin experiments– The use of auxin induced degradation to disrupt function is a nice way to try to examine tissue-specific effects.*– Data presented in 8E,F,G show the different degron strains either untreated or treated with auxin. A control is needed for treatment with auxin in the absence of the degron transgene – especially since all strains undergo a similar reduction in brood size in the presence of auxin (8E,F)

This is a very good point and we have added such a control shown in Figure 8E in which the strains, including wild type (Bristol N_2_), PEZO-1::degron and each tir-1::mRuby transgenes alone driven by different promoters, were treated with and without auxin. Auxin alone did not reduce the brood size of these strains.

– Figure 8C,D – The sperm expression is described as faint, and it is indeed hard to see in the images. It is not unexpected that expression might be low, but are the authors confident that the putative sperm expression is not autofluorescence?

We added new data of the sperm autofluorescence in the Figure 8—figure supplement 1, with same imaging acquisition and exposure conditions. We were unable to detect sperm autofluorescence with wavelength of 561 nm, but we did observe that sperm cytosol has high autofluorescence with wavelength of 488 nm. Only germline specific tir-1::mRuby strains display red fluorescence in the sperm.

– It should be made clear in the text that germline expression includes (or is likely to include) both sperm and oocytes, so that either tissue could be the primary course of phenotypes observed with the germline AID strains. I do agree that it is more likely to be due to oocyte/ attractant signaling defects.

We revised the text to make the point clearer.

– The text states that Figure 8H, J show the somatic-specific (Peft-3) AID strain with auxin, but the figure is labeled as the Ppie-1 strain without auxin. Which is it?

We rearranged this whole figure to make it cleaner and corrected all the labels and callouts in the text and legend.

– The age of the hermaphrodites used for the mitotracker assays needs to be stated. This is relevant to whether or not self sperm could be contribute to targeting defects.

We state the stage of worms in the Figure 8K, and we added new data to address whether our phenotype is self-sperm dependent in Figure 6—figure supplement 1.

Re: Multiple roles in inter-tissue signaling– *The data in Figure S5 demonstrate enhancement of the brood size defects in inx; pezo as compared to inx(RNAi) or pezo- alone. However, this could be interpreted in many different ways, and does not necessarily mean that the same process is being affected, especially when a relatively non-specific phenotype is the assay. Thus, this experiment does not add insight.

We see your point since inx RNAi does significantly reduce the brood size in wild-type animals, such RNAi may further reduce the brood size in other genetic backgrounds whether in the same pathway or not. So, we have omitted this figure and the relevant text that accompanied it.

– The images in Figure S6 do show excess extracellular yolk in pezo-1 that is not present in WT. However, YP170 levels in pezo-1 oocytes appear higher than in wild type, if anything. Therefore, I am not convinced that these data provide evidence for defects in YP170 endocytosis. Instead, is it possible that there are defects within oocytes, in conversion of yolk to downstream signaling molecules? Without additional experiments, this also seems to be dispensable.Overall, while the experiments in this section do not contradict the model of pezo-1's being involved in inter-tissue signaling, they do little to support it.

Given the comments of all three reviewers, we agree that this figure can also be removed. This is an observation we are pursuing but agree that much more work would be required to make this an important part of the story.